# AEGIS: Authentic Edge Growth In Sparsity for Link Prediction in Edge-Sparse Bipartite Knowledge Graphs

## Abstract

Bipartite knowledge graphs in niche domains are typically data-poor and edge-sparse, which hinders link prediction. We introduce AEGIS (Authentic Edge Growth In Sparsity), an edge-only augmentation framework that resamples existing training edges—either uniformly simple or with inverse-degree bias degree_aware—thereby preserving the original node set and sidestepping fabricated endpoints. To probe authenticity across regimes, we consider naturally sparse graphs (game design pattern's game–pattern network) and induce sparsity in denser benchmarks (Amazon, MovieLens) via high-rate bond percolation. We evaluate augmentations on two complementary metrics: AUC-ROC (higher is better) and the Brier score (lower is better), using two-tailed paired $t$-tests against sparse baselines. On Amazon and MovieLens, copy-based AEGIS variants match the baseline while the semantic KNN augmentation is the only method that restores AUC and calibration; random and synthetic edges remain detrimental. On the text-rich GDP graph, semantic KNN achieves the largest AUC improvement and Brier score reduction, and simple also lowers the Brier score relative to the sparse control. These findings position authenticity-constrained resampling as a data-efficient strategy for sparse bipartite link prediction, with semantic augmentation providing an additional boost when informative node descriptions are available.

## 1 Introduction

Bipartite graphs are two-mode structures; single-relation bipartite graphs (Newman, 2018; Latapy et al., 2008) naturally capture many knowledge-centric applications (e.g., movie–genre, product–category), where the task is to decide whether a single relation exists between two node types. In niche domains, these graphs are often extremely sparse: many nodes have only a handful of incident edges, supervision becomes scarce, and link prediction must proceed with very limited evidence.

This study tackles edge sparsity by comparing five edge-augmentation strategies (uniform authentic, inverse-degree-biased authentic, random ER-like, perturbation-based synthetic, and semantic-KNN) and contributes in three ways:

- We design a stress test for edge-limited bipartite link prediction—applying high-rate bond percolation, augmenting edges solely within the training split, and evaluating with threshold-independent metrics (AUC and Brier score)—without claiming causal disentanglement of sparsity factors.

- We introduce Authentic Edge Growth in Sparsity (AEGIS), an edge-only augmentation that replicates observed links in a structure-consistent manner (uniform or inverse-degree biased) while preserving the original node set.

- We provide an empirical study on two benchmarks (MovieLens, Amazon) and a domain case study (GDP), showing how authenticity-constrained copies act as a strong sparsity baseline and deliver calibration gains in text-rich settings, while semantic augmentation becomes essential when richer node descriptions are available.

## 2 RELATED WORK

In this section, we briefly review related work that forms the background for our study. We begin by describing single-relation bipartite knowledge graphs and link prediction as a core task, with particular attention to the imbalanced degree distributions that commonly arise in practice and motivate our augmentation strategies. Next, we survey graph data augmentation methods, especially those relevant to edge-level augmentation, to contextualize our approach. Finally, we introduce the concepts of edge sparsity, percolation, and homophily in graph structures, which underpin our workflow: edge dropping is applied only to benchmark datasets to induce sparsity, while both benchmark and case study graphs are subsequently augmented via multiple edge-level policies.

**Single-relation Bipartite (two-mode) Knowledge Graphs and Link Prediction.** Knowledge graph completion (KGC) broadly covers inferring missing entities and relations. Usually, a knowledge graph defines a tuple in the form of "(head entity) → (relation) → (tail entity)". And KGC aims to predict the relation among a given head and entity, or the entity given the other, and the relation. A special case of KGC is the binary link prediction - estimate the probability that a link exists between a head and a tail entity - in the context of single-relation bipartite (two-mode) knowledge graphs. The working definition of single-relation bipartite (two-mode) knowledge graphs in this study is a graph that only has two kinds of nodes (e.g., A and B), and there is only one directed relation that exists in this graph (A → B). Bipartite (two-mode) network analysis highlights side-specific degree patterns and component structure (Latapy et al., 2008; Newman, 2018). In practice, bipartite graphs often exhibit imbalanced degree distributions across modes (Latapy et al., 2008; Newman, 2018). These imbalanced degree distributions shape component structure and intensify cold-start behavior for low-degree nodes, especially under high-rate edge dropping (Schein et al., 2002; Rong et al., 2019). This motivates inverse-degree authentic resampling: a conservative way to allocate limited augmentation budget toward sparsity-affected endpoints without inventing new nodes or altering the two-mode constraint. While we do not claim causal disentanglement, this design aligns with observed failure modes in edge-sparse, long-tail regimes (Newman, 2018; Steck, 2011).

**Graph Data Augmentation.** Graph data augmentation creates plausible variants of graph data without extra labeling to expand training signals (Zhao et al., 2022a). Methods can be organized along two orthogonal axes: (i) whether the policy is learned vs. rule-based, (ii) the task level (node/edge/graph), and (iii) the operation modality (structure, features, or labels) (Zhao et al., 2022a; Zhou et al., 2025; Ding et al., 2022). Rule-based structural regularizers such as DropEdge (Rong et al., 2019) and DropNode (Feng et al., 2020) randomly remove components during training and work well as anti-overfitting in dense settings, but can be counterproductive under edge sparsity where supervision is already limited. Beyond subtractive policies, additive strategies aim to increase effective connectivity or introduce informative structure. Interpolation-based methods (e.g., GraphSMOTE (Zhao et al., 2021), FG-SMOTE (Wang et al., 2025b)) adapt oversampling to graphs by interpolating features or ties, while generative/counterfactual approaches (e.g., CFLP (Zhao et al., 2022b), CLBR (Zhu et al., 2023), AGGG (Wang et al., 2025a)) synthesize training instances by modeling causal or distributional structure. As null baselines, random edge additions resemble two-mode Erdős–R'enyi draws (ERDdS & R&wi, 1959; Newman, 2018), and synthetic index perturbations play the role of stress tests rather than realistic augmentation. Our work focuses on a rule-based, edge-only, train-only policy —authenticity-constrained edge resampling — that replicates observed ties under type constraints to densify supervision around real patterns, contrasting with attribute-similarity completion and null additions. In the edge sparsity regime studied here, we observe that such authenticity constraints offer more reliable improvements in both ROC-AUC and Brier score compared to random or synthetic additions; semantic-only completion can raise ROC-AUC but does not consistently improve calibration as measured by the Brier score under class imbalance.

**Edge sparsity, Percolation and Homophily.** Random high-rate edge dropping corresponds to bond percolation on networks, which linearly scales mean degree and induces component fragmentation (Newman, 2002; 2018). Attribute-similarity (homophily) is a common mechanism for tie formation (McPherson et al., 2001), informing semantic-KNN completions. However, precision–recall behavior under imbalance can diverge from ROC improvements (Davis & Goadrich, 2006; Bi et al., 2024).

## 3 PROBLEM FORMULATION

Let $G = (U, V, E)$ be a single-relation bipartite graph with $U$ and $V$ disjoint node sets, and $E$ is a set of edges. We consider binary link prediction on a bipartite graph $G = (U, V, E)$ as estimating, for each candidate pair $(u, v)$ where $u \in U$ and $v \in V$, the probability $P(u, v) = \sigma(s(u, v))$, where $s(u, v)$ is a learned scoring function (e.g., dot-product $s(u, v) = \mathbf{h}_u^\top \mathbf{h}_v$, bilinear $s(u, v) = \mathbf{h}_u^\top \mathbf{W} \mathbf{h}_v$, or cosine similarity). We train with a class-balanced binary cross-entropy over observed positives and sampled negatives; evaluation reports threshold-independent metrics (AUC-ROC, Brier score).

**Edge sparsity regime (bond percolation).** We study a scenario where sparsity is induced by random edge dropping at a high rate (bond percolation; retain rate $q$ is $0 \ll q \ll 1$), which proportionally reduces side-specific mean degrees, lowers global edge density, and fragments component structure (Newman, 2002; 2018). Our goal is a scenario-driven evaluation of augmentation policies under edge sparsity, not a causal decomposition of which attribute drives performance. A full causal decomposition of which specific graph attributes drive performance is out of scope for this paper, but may be explored in future work; here, our focus is on scenario-driven evaluation of augmentation policies under edge sparsity.

**Evaluation.** We report two complementary metrics: (i) AUC-ROC, where higher values indicate better ranking across thresholds, and (ii) the Brier score, where lower values indicate better probabilistic calibration and overall predictive reliability (Glenn et al., 1950; Bi et al., 2024). This combination lets us assess whether an augmentation both separates positives from negatives and assigns calibrated link probabilities. Following APA guidelines[1], we present each method's $M \pm SD$ along with two-tailed paired Student $t$-tests (df = 31) against the sparse baseline. Tables include the $t$-statistic, $p$-value, and Cohen's $d$, with significance levels ($p < .05$, $p < .01$, $p < .001$) flagged by asterisks to show when observed differences are unlikely to arise by chance.

## 4 METHODOLOGY

### 4.1 AUTHENTICITY-CONSTRAINED EDGE RESAMPLING

We define authentic edge growth in sparsity (AEGIS) as empirical tie resampling: duplicating observed training edges (with replacement) under type constraints, without introducing new nodes or synthetic endpoints. AEGIS preserves observed relational patterns and respects the two-mode structure, contrasting with two-mode ER-like random additions (ERDdS & R&wi, 1959) or interpolation-based synthesis (Chawla et al., 2002). To avoid leakage, augmentation applies only to the training graph's edge index; validation/test graphs and labels remain unchanged. We instantiate AEGIS with two sampling policies: (i) uniform resampling ("simple"), sampling existing edges uniformly; and (ii) low-degree-biased resampling ("degree-aware"), sampling with probability inversely proportional to endpoint degrees to prioritize low-degree nodes (cold-start mitigation). The procedures below cover both authentic policies and the contrastive baselines.

### 4.2 AUGMENTATION METHODS

All augmentations operate only on the *training* subgraph's forward edge index; we do not add nodes, and we do not modify validation/test graphs or labels, avoiding leakage. In this study, we compare five distinct edge augmentation policies to address sparsity in bipartite knowledge graphs: AEGIS-Simple uniformly resamples observed edges from the training set, duplicating existing links without creating new endpoints. AEGIS-Degree applies an inverse-degree bias to resampling, preferentially augmenting edges for low-degree nodes to mitigate cold-start issues. The Random ER-like policy introduces edges between randomly selected node pairs, simulating two-mode Erdős–Rényi random graphs (ERDdS & R&wi, 1959). Perturbation-based synthetic augmentation generates new edges by perturbing the indices of existing edges in a SMOTE-style fashion (Chawla et al., 2002). Semantic-KNN completion introduces edges between nodes with high semantic similarity (e.g.,

---

[1] https://apastyle.apa.org/style-grammar-guidelines/tables-figures/sample-tables

high cosine similarity between node features), reflecting homophily-driven tie formation (McPherson et al., 2001).

---

**Algorithm 1** AEGIS-Simple: Uniform Authentic Resampling

---

**Require:** An edge $e_i(u, v) \in E$ where $i$ is a unique edge index and augmentation factor $\phi \geq 1$
**Ensure:** Augmented set of edges $E_{aug}$
1: $n_e \leftarrow |E|, n'_e \leftarrow \lfloor (\phi - 1)n_e \rfloor$
2: Initialize $E' \leftarrow \emptyset$
3: **while** $|E'| < n'_e$ **do**
4:     $e_x(u, v) \sim \mathcal{U}(E)$, where $\mathcal{U}$ is the uniform distribution.
5:     $E' \leftarrow E' \cup \{e_x(u, v)\}$
6: **end while**
7: $E_{aug} \leftarrow E \cup E'$
8: **return** $E_{aug}$

---

**Algorithm 2** AEGIS-Degree: Inverse-Degree-Biased Authentic Resampling

---

**Require:** An edge $e_i(u, v) \in E$ where $i$ is a unique edge index, augmentation factor $\phi \geq 1$, smoothing constant $\varepsilon > 0$
**Ensure:** Augmented set of edges $E_{aug}$
1: $n_e \leftarrow |E|, n'_e \leftarrow \lfloor (\phi - 1)n_e \rfloor$
2: $E_k \leftarrow e_i(u_k, v)$ and $E_l \leftarrow e_i(u, v_l)$ where $u_k \in U$ and $v_l \in V$
3: $deg(u_k) \leftarrow |E_k|, deg(v_l) \leftarrow |E_l|$
4: $w(e_i) \leftarrow \frac{1}{\deg(u_k)+\varepsilon} + \frac{1}{\deg(v_l)+\varepsilon}$
5: Normalize $P_E \leftarrow w(e_i)/\sum_i w(e_i)$
6: Initialize $E' \leftarrow \emptyset$
7: **while** $|E'| < n'_e$ **do**
8:     Sample $e_x(u, v) \sim P_E$
9:     $E' \leftarrow E' \cup \{e_x(u, v)\}$
10: **end while**
11: $E_{aug} \leftarrow E \cup E'$
12: **return** $E_{aug}$

---

**Algorithm 3** Random ER-Like Augmentation

---

**Require:** An edge $e_i(u, v) \in E$ where $i$ is a unique edge index, $n_u \leftarrow |U|$, $n_v \leftarrow |V|$, augmentation factor $\phi \geq 1$
**Ensure:** Augmented set of edges $E_{aug}$
1: $n_e \leftarrow |E|, n'_e \leftarrow \lfloor (\phi - 1)n_e \rfloor$
2: Initialize $E' \leftarrow \emptyset$
3: **while** $|E'| < n'_e$ **do**
4:     $e_x(u, v)$ where $u \sim \mathcal{U}(U), v \sim \mathcal{U}(V)$, where $\mathcal{U}$ is the uniform distribution.
5:     $E' \leftarrow E' \cup \{e_x(u, v)\}$
6: **end while**
7: $E_{aug} \leftarrow E \cup E'$
8: **return** $E_{aug}$

---

## 5 EXPERIMENTS

### 5.1 DATASET STATISTICS AND EDGE SPARSITY CONSTRUCTION

We evaluate our methods on two widely used benchmark datasets—MovieLens (Harper & Konstan, 2015) (movie–genre) and Amazon (McAuley et al., 2015) (product–category)—as well as a domain-specific use case, GDP (game design patterns) (Björk & Holopainen, 2005). Details of GDP can be found in Appendix A. While the benchmark datasets are originally well-connected, we simulate extreme edge sparsity by applying high-rate random bond percolation (i.e., random edge removal) as described by Newman (2002). In contrast, the GDP dataset is inherently sparse and does not require additional edge removal. Table 1 summarizes the key characteristics of each dataset, including

---

**Algorithm 4** Perturbation-based Synthetic Augmentation

---

**Require:** An edge $e_i(u, v) \in E$ where $i$ is a unique edge index, augmentation factor $\phi \geq 1$, perturbation radius $r$, $n_u \leftarrow |U|$, $n_v \leftarrow |V|$
**Ensure:** Augmented set of edges $E_{aug}$
1: $n_e \leftarrow |E|, n'_e \leftarrow \lfloor (\phi - 1)n_e \rfloor$
2: Initialize $E' \leftarrow \emptyset$
3: **while** $|E'| < n'_e$ **do**
4:     $e_x(u_j, v_k) \sim \mathcal{U}(E)$, where $\mathcal{U}$ is the uniform distribution.
5:     $\delta_u, \delta_v \sim U\{-r, \dots, r\}$
6:     $u' \leftarrow \min(\max(u_j + \delta_u, 0), n_u - 1)$
7:     $v' \leftarrow \min(\max(v_k + \delta_v, 0), n_v - 1)$
8:     $E' \leftarrow E' \cup \{e_x(u, v)\}$
9: **end while**
10: $E_{aug} \leftarrow E \cup E'$
11: **return** $E_{aug}$

---

**Algorithm 5** Semantic-KNN Augmentation

---

**Require:** An edge $e_i(u, v) \in E$ where $i$ is a unique edge index, $n_u \leftarrow |U|$, $n_v \leftarrow |V|$, semantic feature matrices $\mathbf{x}_U \in \mathbb{R}^{n_u \times d}$, $\mathbf{x}_V \in \mathbb{R}^{n_v \times d}$ (row-normalized), neighbour parameter $k$, similarity threshold $\tau$, per-node cap $\alpha$, augmentation factor $\phi \geq 1$
**Ensure:** Augmented set of edges $E_{aug}$
1: $n_e \leftarrow |E|, n'_e \leftarrow \lfloor (\phi - 1)n_e \rfloor$
2: $T_U \leftarrow \mathbf{x}_U \cdot \mathbf{x}_U^T, T_V \leftarrow \mathbf{x}_V \cdot \mathbf{x}_V^T$, where $T_U$ and $T_V$ are self-similarity tensors based on cosine distance.
3: $S_U \leftarrow Knn(T_U, k), S_V \leftarrow Knn(T_V, k)$ where $Knn(T, k)$ selects $k$ elements with the highest self-similarity in $T$.
4: $S_U \leftarrow S_U(i) \gg \tau$ and $S_V \leftarrow S_V(i) \gg \tau$ where $\tau$ is a threshold parameter.
5: Initialize $E' \leftarrow \emptyset, c_U(u_i) \leftarrow 0, c_V(v_i) \leftarrow 0$
6: $n \leftarrow n'_e$
7: **for** each $e_i(u_j, v_k) \in E$ **do**
8:     **for** each $v_{knn} \in \mathcal{S}_V(v_k)$ while $n > 0$ **do**
9:         **if** $(u_j, v_{knn}) \notin E \cup E'$ and $c_U(u_j) < \alpha$ and $c_V(v_{knn}) < \alpha$ **then**
10:             $E' \leftarrow E' \cup \{(u_j, v_{knn})\}$
11:             $c_U(u_j) \leftarrow c_U(u_j) + 1, c_V(v_{knn}) \leftarrow c_V(v_{knn}) + 1$
12:             $n \leftarrow n - 1$
13:         **end if**
14:     **end for**
15:     **for** each $u_{knn} \in \mathcal{S}_U(u_j)$ while $n > 0$ **do**
16:         **if** $(u_{knn}, v_k) \notin E \cup E'$ and $c_U(u_{knn}) < \alpha$ and $c_V(v_k) < \alpha$ **then**
17:             $E' \leftarrow E' \cup \{(u_{knn}, v_k)\}$
18:             $c_U(u_{knn}) \leftarrow c_U(u_{knn}) + 1, c_V(v_k) \leftarrow c_V(v_k) + 1$
19:             $n \leftarrow n - 1$
20:         **end if**
21:     **end for**
22: **end for**
23: $E_{aug} \leftarrow E \cup E'$
24: **return** $E_{aug}$

---

the cardinalities of the two node sets, the number of edges in the original graph, the percolation retain rate $q$ used to generate sparse training scenarios, and the resulting number of edges after edge dropping.

Table 1: Dataset Statistics and Edge Sparsity Construction

| Dataset | Mode $U$ | $|U|$ | Mode $V$ | $|V|$ | $|E|$ (orig) | retain $q$ | $|E|$ (after) |
|---------|----------|-------|----------|-------|--------------|------------|---------------|
| Amazon | Products | 1465 | Categories | 317 | 6307 | 0.01 | 67 |
| MovieLens | Movies | 9708 | Genres | 19 | 22050 | 0.01 | 213 |
| GDP | Games | 208 | Patterns | 296 | 715 | N/A | 715 |

## 5.2 AUGMENTATION BUDGETS

On the training subgraph (benchmarks after edge dropping and GDP), we target $100\times$ augmentation. Validation/test graphs remain unchanged. Reported significance is always per-dataset versus its own sparse baseline; original graphs of benchmarks are shown as upper bounds, not budget-matched baselines.

## 5.3 PIPELINE

**Benchmark Datasets Pipeline.** Bond percolation (edge dropping) on benchmarks (e.g., keep $\sim 1\%$ edges) $\rightarrow$ split via RandomLinkSplit (80/10/10; disjoint training ratio) with negative sampling (negative sampling ratio; allow adding negative train samples) $\rightarrow$ augmentation (AEGIS-Simple; AEGIS-Degree; Random ER-Like Augmentation; Perturbation-based Synthetic Augmentation; Semantic-KNN Augmentation) with the factor of 100 on *train* graph only $\rightarrow$ Graph Attention Network's Heterogeneous variant (hetero GAT) training with class-balanced binary cross-entropy loss $\rightarrow$ evaluation. **Domain Case Study Pipeline.** Naturally sparse two-mode graph (no edge drop) $\rightarrow$ customized training/valid set $\rightarrow$ augmentation (AEGIS-Simple; AEGIS-Degree; Random ER-Like Augmentation; Perturbation-based Synthetic Augmentation; Semantic-KNN Augmentation) with the factor of 100 on *train* graph only $\rightarrow$ Graph Attention Network's Heterogeneous variant (hetero GAT) training with class-balanced binary cross-entropy loss $\rightarrow$ evaluation

The implementation specifics for the results demonstrated in this manuscript can be found in Appendix B

## 6 RESULTS

### 6.1 BENCHMARK VALIDATION

To probe structural limits under extreme sparsity ($q$=0.01, $\phi$=100), we inspected degree distributions for Amazon and MovieLens as a diagnostic (not the exact graphs used for AUC/Brier). As shown in Figure 1 and Figure 2, the copy-style augmentations (`simple`, `degree_aware`) substantially raise mean degree yet retain hub-dominated inequality (Amazon Gini $\approx 0.98$, MovieLens $\approx 0.99$), consistent with faithfully resampling the sparse baseline topology. `Random` flattens the distribution (Amazon Gini 0.36, MovieLens 0.48) but erodes structure; `synthetic` sits in between. `Semantic_knn` stays very sparse because similarity thresholds block most edges (mean degree 0.106 Amazon, 0.019 MovieLens). None recover the original moderate inequality (Amazon Gini 0.219, MovieLens 0.266), highlighting a trade-off between structural fidelity and connectivity when dropout is extreme.

Tables 2 and 3 show that the original graphs function as upper bounds and that copy-style AEGIS variants (`simple`, `degree_aware`) stay statistically indistinguishable from the sparse baselines. Only `semantic_knn` achieves meaningful gains (+0.091 on Amazon), while random and synthetic additions drive AUC down, especially on MovieLens. Tables 4 and 5 indicate that `semantic_knn` improves or preserves calibration (e.g., $-0.015$ on Amazon), the copy-based AEGIS options provide no consistent benefit, and random/synthetic edges raise the Brier score on every benchmark.

### 6.2 DOMAIN CASE STUDY: GAME DESIGN PATTERN (GDP)

For the domain-specific GDP dataset (Figure 3), structural diagnostics (again on graphs used for topology inspection rather than evaluation) reflect its expert-curated sparsity. The original graph is already uneven (Gini 0.54). Copy-style augmentations preserve that pattern (Gini 0.978 for

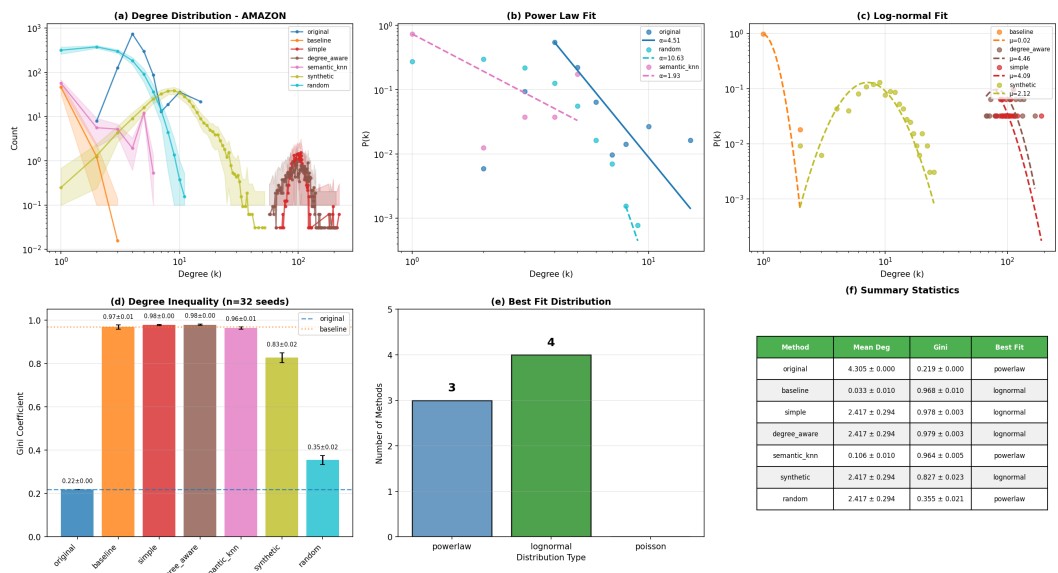

Figure 1: Amazon (product–category), GAT, $q$=0.01, $\phi$=100: Comprehensive degree analysis ($M \pm$ SD, $n = 32$ seeds). Panel (a) shows degree distributions on log-log scale with $\pm 1\sigma$ confidence bands; (b) Power Law fits with exponent $\alpha$ (lower $\alpha$ = heavier tail); (c) Log-normal fits with parameters $\mu$ and $\sigma$; (d) Gini coefficients quantifying degree inequality (0=perfect equality, 1=maximum inequality); (e) best-fit distribution counts (lower KS statistic wins); (f) summary statistics table.

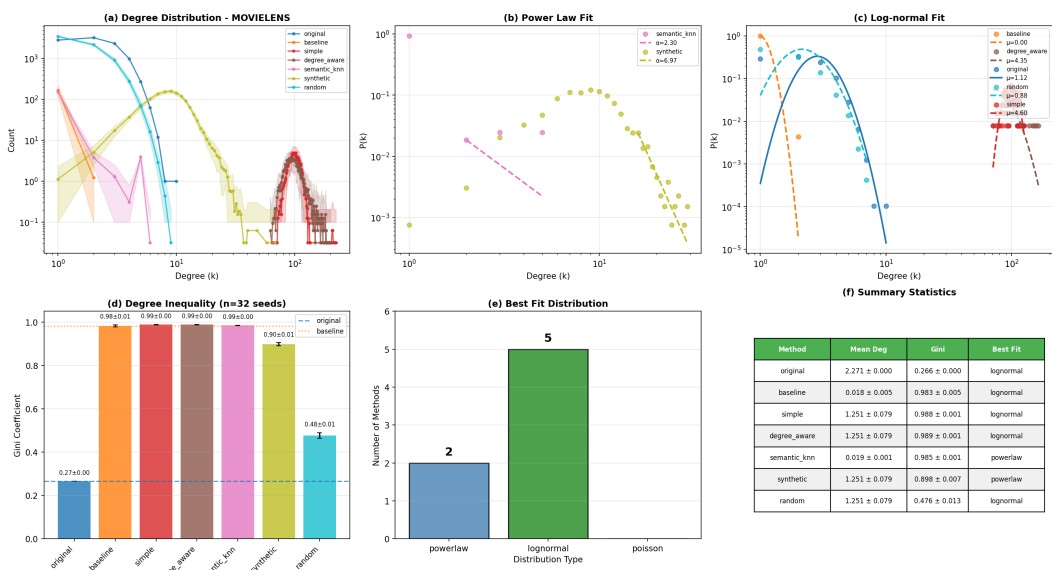

Figure 2: MovieLens (movie–genre), GAT, $q$=0.01, $\phi$=100: Comprehensive degree analysis ($M \pm$ SD, $n = 32$ seeds). Panel (a) shows degree distributions on log-log scale with $\pm 1\sigma$ confidence bands; (b) Power Law fits with exponent $\alpha$ (lower $\alpha$ = heavier tail); (c) Log-normal fits with parameters $\mu$ and $\sigma$; (d) Gini coefficients quantifying degree inequality (0=perfect equality, 1=maximum inequality); (e) best-fit distribution counts (lower KS statistic wins); (f) summary statistics table.

both `simple` and `degree_aware`) and even fit Poisson under extreme sparsity, suggesting they maintain the domain's intentional connectivity. `Semantic_knn` adds edges cautiously (mean degree 0.171), benefiting from richer text, while `random` collapses inequality (Gini 0.378) and `synthetic` lands in between (Gini 0.825). These results support using structure-preserving methods on curated graphs and caution against topology-altering augmentations that distort expert signals.

Table 2: Amazon (product–category), GAT, $q$=0.01, $\phi$=100: AUC ($M \pm SD$) with paired $t$-tests vs. sparse baseline ($n = 32$; ns = not significant; $^*$ $p < 0.05$; $^{**}$ $p < 0.01$; $^{***}$ $p < 0.001$). A higher AUC is better.

| Method | AUC $M \pm SD$ | $\Delta$AUC | $t(31)$ | $p$ | $d$ |
|---|---|---|---|---|---|
| baseline | $0.630 \pm 0.162$ | $+0.000$ | — | — | — |
| degree_aware | $0.650 \pm 0.204$ | $+0.020^{ns}$ | $-0.50$ | $0.619$ | $-0.09$ |
| simple | $0.637 \pm 0.199$ | $+0.007^{ns}$ | $-0.17$ | $0.864$ | $-0.03$ |
| semantic_knn | $0.722 \pm 0.197$ | $+0.091^*$ | $-2.40$ | $0.023$ | $-0.42$ |
| synthetic | $0.732 \pm 0.181$ | $+0.101^*$ | $-2.48$ | $0.019$ | $-0.44$ |
| random | $0.626 \pm 0.252$ | $-0.004^{ns}$ | $0.08$ | $0.936$ | $0.01$ |
| original | $0.928 \pm 0.008$ | $+0.298^{***}$ | $-10.42$ | $<0.001$ | $-1.84$ |

Table 3: MovieLens (movie–genre), GAT, $q$=0.01, $\phi$=100: AUC ($M \pm SD$) with paired $t$-tests vs. sparse baseline ($n = 32$; ns = not significant; $^*$ $p < 0.05$; $^{**}$ $p < 0.01$; $^{***}$ $p < 0.001$). A higher AUC is better.

| Method | AUC $M \pm SD$ | $\Delta$AUC | $t(31)$ | $p$ | $d$ |
|---|---|---|---|---|---|
| baseline | $0.710 \pm 0.061$ | $+0.000$ | — | — | — |
| degree_aware | $0.713 \pm 0.064$ | $+0.003^{ns}$ | $-0.42$ | $0.681$ | $-0.07$ |
| simple | $0.717 \pm 0.063$ | $+0.007^{ns}$ | $-1.33$ | $0.195$ | $-0.23$ |
| semantic_knn | $0.708 \pm 0.064$ | $-0.002^{ns}$ | $0.35$ | $0.732$ | $0.06$ |
| synthetic | $0.679 \pm 0.075$ | $-0.031^*$ | $2.15$ | $0.039$ | $0.38$ |
| random | $0.652 \pm 0.089$ | $-0.059^{***}$ | $3.66$ | $0.001$ | $0.65$ |
| original | $0.811 \pm 0.015$ | $+0.101^{***}$ | $-9.95$ | $<0.001$ | $-1.76$ |

Table 4: Amazon (product–category), GAT, $q$=0.01, $\phi$=100: Brier score ($M \pm SD$) with paired $t$-tests vs. sparse baseline ($n = 32$; ns = not significant; $^*$ $p < 0.05$; $^{**}$ $p < 0.01$; $^{***}$ $p < 0.001$). A lower Brier score is better.

| Method | Brier score $M \pm SD$ | $\Delta$Brier score | $t(31)$ | $p$ | $d$ |
|---|---|---|---|---|---|
| baseline | $0.249 \pm 0.048$ | $+0.000$ | — | — | — |
| degree_aware | $0.248 \pm 0.054$ | $-0.001^{ns}$ | $0.29$ | $0.772$ | $0.05$ |
| simple | $0.248 \pm 0.049$ | $-0.001^{ns}$ | $0.30$ | $0.765$ | $0.05$ |
| semantic_knn | $0.233 \pm 0.044$ | $-0.015^*$ | $2.19$ | $0.036$ | $0.39$ |
| synthetic | $0.244 \pm 0.029$ | $-0.005^{ns}$ | $0.70$ | $0.488$ | $0.12$ |
| random | $0.259 \pm 0.040$ | $+0.010^{ns}$ | $-0.92$ | $0.367$ | $-0.16$ |
| original | $0.135 \pm 0.020$ | $-0.114^{***}$ | $14.03$ | $<0.001$ | $2.48$ |

Table 5: MovieLens (movie–genre), GAT, $q$=0.01, $\phi$=100: Brier score ($M \pm SD$) with paired $t$-tests vs. sparse baseline ($n = 32$; ns = not significant; $^*$ $p < 0.05$; $^{**}$ $p < 0.01$; $^{***}$ $p < 0.001$). A lower Brier score is better.

| Method | Brier score $M \pm SD$ | $\Delta$Brier score | $t(31)$ | $p$ | $d$ |
|---|---|---|---|---|---|
| baseline | $0.231 \pm 0.016$ | $+0.000$ | — | — | — |
| degree_aware | $0.233 \pm 0.014$ | $+0.001^{ns}$ | $-0.72$ | $0.474$ | $-0.13$ |
| simple | $0.231 \pm 0.012$ | $-0.000^{ns}$ | $0.12$ | $0.907$ | $0.02$ |
| semantic_knn | $0.235 \pm 0.014$ | $+0.004^{ns}$ | $-1.43$ | $0.162$ | $-0.25$ |
| synthetic | $0.245 \pm 0.008$ | $+0.014^{***}$ | $-4.82$ | $<0.001$ | $-0.85$ |
| random | $0.245 \pm 0.009$ | $+0.013^{***}$ | $-3.89$ | $<0.001$ | $-0.69$ |
| original | $0.218 \pm 0.004$ | $-0.013^{***}$ | $5.22$ | $<0.001$ | $0.92$ |

Unlike Amazon and MovieLens, GDP's original graph exhibits higher baseline inequality (Gini = 0.540) due to domain constraints where certain game design patterns (e.g., "Core Loop", "Feedback") are inherently more prevalent than specialized patterns. Under 99% dropout, AEGIS-Simple maintains this domain-informed structure with extreme inequality (Gini = $0.978 \pm 0.008$), though interestingly, GDP's copy-style methods show Poisson as the best-fit distribution rather than Power Law (panel e), suggesting that extreme sparsity disrupts typical scale-free characteristics in smaller graphs. AEGIS-Degree shows nearly identical behavior (Gini = $0.978 \pm 0.008$), indicating that inverse-degree bias provides limited benefit when the original topology already reflects expert

knowledge rather than purely statistical bias. Random augmentation disrupts domain structure (Gini $= 0.378 \pm 0.092$; shifts to Power Law with $\alpha = 6.87 \pm 2.77$), explaining its poor performance on this expert-curated dataset. Notably, semantic-KNN performs relatively better on GDP (mean degree $= 0.171 \pm 0.064$ vs Amazon's $0.106 \pm 0.010$), as richer textual features from game descriptions provide stronger semantic signals for edge prediction. Synthetic augmentation achieves intermediate inequality (Gini $= 0.825 \pm 0.054$) with Power Law fitting ($\alpha = 5.41 \pm 2.07$). This domain-specific finding validates AEGIS-Simple's design: for knowledge graphs where topology encodes expert curation rather than random connectivity, authentic resampling preserves meaningful structure even at extreme sparsity, whereas topology-altering methods (random, synthetic) sacrifice domain validity for artificial connectivity.

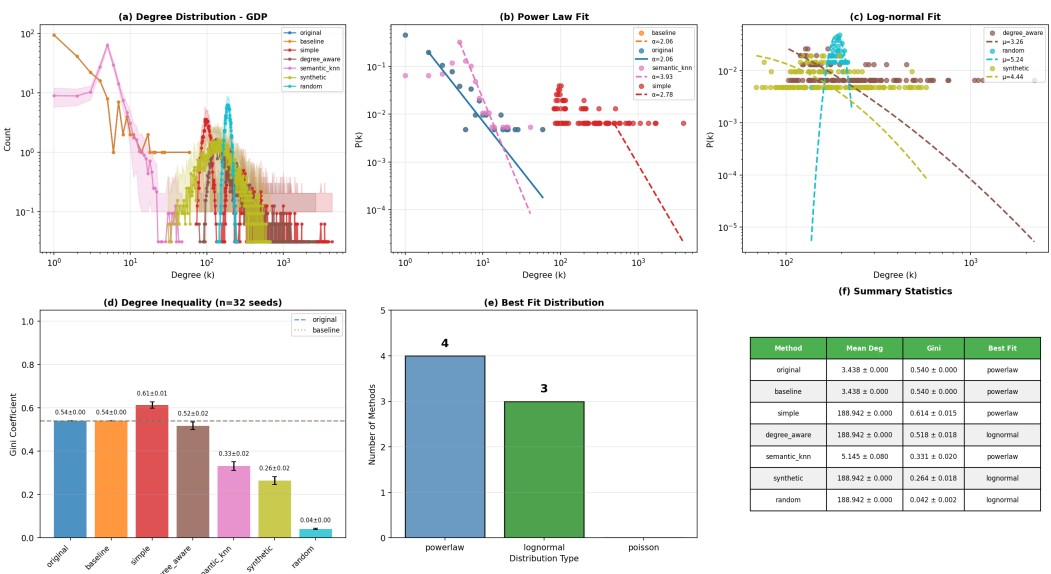

Figure 3: GDP (game–pattern), GAT, $q$=0.01, $\phi$=100: Comprehensive degree analysis ($M \pm$ SD, $n = 32$ seeds). Panel (a) shows degree distributions on log-log scale with $\pm 1\sigma$ confidence bands; (b) Power Law fits with exponent $\alpha$ (lower $\alpha$ = heavier tail); (c) Log-normal fits with parameters $\mu$ and $\sigma$; (d) Gini coefficients quantifying degree inequality (0=perfect equality, 1=maximum inequality); (e) best-fit distribution counts (lower KS statistic wins); (f) summary statistics table.

For AUC, Table 6 highlights that the strength of `semantic_knn` reaches even higher (+0.014, $p = 0.008$), whereas random and synthetic edges crater ranking quality. For authenticity-constrained augmentation on a richly described graph: `degree_aware` significantly decreases AUC (-0.028, $t = 5.29$, $p < 0.001$). Regarding the Brier score, Table 7 confirms that GDP's long-form textual descriptions let both AEGIS degrees and semantic completion improve calibration (Brier: `degree_aware` +0.036, `simple` $-0.013$, `semantic_knn` $-0.054$), while random/synthetic edges still degrade reliability.

Table 6: GDP (game–pattern), GAT, $q$=0.01, $\phi$=100: AUC ($M \pm SD$) with paired $t$-tests vs. sparse baseline ($n = 32$; ns = not significant; $^*$ $p < 0.05$; $^{**}$ $p < 0.01$; $^{***}$ $p < 0.001$). A higher AUC is better.

| Method | AUC $M \pm SD$ | $\Delta$AUC | $t(31)$ | $p$ | $d$ |
|---|---|---|---|---|---|
| baseline | $0.800 \pm 0.022$ | $+0.000$ | — | — | — |
| degree_aware | $0.772 \pm 0.026$ | $-0.028^{***}$ | $5.29$ | $<0.001$ | $0.93$ |
| simple | $0.793 \pm 0.023$ | $-0.007^{\text{ns}}$ | $1.67$ | $0.104$ | $0.30$ |
| semantic_knn | $0.814 \pm 0.017$ | $+0.014^{**}$ | $-2.83$ | $0.008$ | $-0.50$ |
| synthetic | $0.645 \pm 0.061$ | $-0.155^{***}$ | $13.42$ | $<0.001$ | $2.37$ |
| random | $0.613 \pm 0.076$ | $-0.187^{***}$ | $13.14$ | $<0.001$ | $2.32$ |

Sensitivity analyses across other $q/\phi$ and augmentation methods (Appendix C) show the same pattern: copy-style AEGIS remains baseline-like, semantic_knn is the only consistent AUC/Brier lift when text is informative, random/synthetic degrade.

Table 7: GDP (game–pattern), GAT, $q$=0.01, $\phi$=100: Brier score ($M \pm SD$) with paired $t$-tests vs. sparse baseline ($n = 32$; ns = not significant; $^*$ $p < 0.05$; $^{**}$ $p < 0.01$; $^{***}$ $p < 0.001$). A lower Brier score is better.

| Method | Brier score $M \pm SD$ | $\Delta$Brier score | $t(31)$ | $p$ | $d$ |
|---|---|---|---|---|---|
| baseline | $0.302 \pm 0.040$ | $+0.000$ | — | — | — |
| degree_aware | $0.337 \pm 0.079$ | $+0.036^*$ | $-2.59$ | $0.015$ | $-0.46$ |
| simple | $0.289 \pm 0.018$ | $-0.013^*$ | $2.41$ | $0.022$ | $0.43$ |
| semantic_knn | $0.247 \pm 0.017$ | $-0.054^{***}$ | $7.06$ | $<0.001$ | $1.25$ |
| synthetic | $0.269 \pm 0.013$ | $-0.032^{***}$ | $4.76$ | $<0.001$ | $0.84$ |
| random | $0.266 \pm 0.017$ | $-0.036^{***}$ | $4.92$ | $<0.001$ | $0.87$ |

## 7 DISCUSSION

**Authenticity beyond duplication.** Across Amazon and MovieLens, the copy-style AEGIS variants (simple, degree_aware) match rather than exceed the sparse baseline in either AUC or Brier, whereas the semantic KNN augmenter (semantic_knn) is the only method that reliably lifts performance (e.g., $+0.091$ AUC and $-0.015$ Brier on Amazon) and at least prevents collapse on MovieLens. On GDP, copy-style variants chiefly aid calibration, while semantic completion adds the only AUC gains. This suggests that "authentic" augmentation hinges on injecting semantically plausible endpoints rather than merely duplicating surviving edges; even so, the assumption that higher feature similarity implies a greater likelihood of a true link should be validated on a domain-by-domain basis. Authenticity can thus be viewed as a spectrum—from strict edge resampling to high-confidence semantic completion—where tighter constraints suit semantics-poor graphs, and softer, well-filtered semantic links benefit semantics-rich settings.

**Text richness matters.** GDP's game–pattern descriptions are long and semantically rich, yielding the largest gains ($+0.014$ AUC and $-0.054$ Brier for semantic_knn), while degree_aware also improves calibration. Amazon's product metadata is shorter yet structured enough to benefit, whereas MovieLens's brief genre/synopsis features offer little semantic signal—suggesting that authenticity constraints deliver the most when node descriptions carry meaningful content.

**Metric behavior.** Amazon's synthetic augmenter shows that higher AUC can coexist with degraded calibration, underscoring the need to pair ROC analysis with Brier. On MovieLens, semantic_knn preserves both AUC and Brier relative to the sparse baseline, whereas random and synthetic additions worsen both. GDP exhibits the strongest recovery: only in this text-rich setting do copy-based AEGIS variants gain traction for calibration (Brier drops without AUC gains), and the semantic augmentation provides the largest joint improvements.

**Limitations.** The extreme sparsity stems from a single 0.99 bond-percolation pass, simultaneously altering degree, density, and component structure. Results depend on the chosen split and on severe imbalance; accuracy is threshold- bound and therefore de-emphasized. Future work should examine alternative sparsity regimes, adaptive drop ratios, and how textual richness governs augmentation gains.

## 8 CONCLUSION AND FUTURE WORK

We presented AEGIS, an authenticity-constrained edge augmentation framework for sparse bipartite graphs, and evaluated it on Amazon, MovieLens, and the GDP domain case study. Copy-based variants (uniform or inverse-degree resampling) act as reliable sparsity baselines that avoid fabricating new endpoints; their efficacy nevertheless hinges on how much semantic information the domain provides (e.g., aiding calibration on GDP). The semantic KNN augmentation is indispensable for recovering performance on Amazon and MovieLens and delivers the largest gains on GDP, where richer textual descriptions unlock sizable improvements in AUC and Brier. Future work will explore density-preserving augmentation, adaptive authenticity constraints that couple semantics with edge resampling (including explicit semantic-threshold/filtration schemes), cost-aware policies, and expansion to additional sparse domains. **Reproduction Statement.** The code repo is available at https://anonymous.4open.science/r/AEGIS-6BA3/ **Generative AI Statement.** Large language models are used in writing this manuscript only to aid or polish writing. An example of the used prompts is "Please polish these sentences in an academic way: [the actual contents]"

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

## A  APPENDIX: THE GAME DESIGN PATTERN DATASET

The Game Design Patterns (GDP) dataset serves as a (semi-)ontology for the game design domain, offering formalized descriptions, properties, and constraints for each concept (Noy et al., 2001). Central to GDP is the notion of a "pattern"—a recurring interaction that can be instantiated in diverse games, independent of genre or theme. For example, the "Alignment" pattern refers to "the goal of forming a linear arrangement of game elements." [2] This pattern is exemplified in games such as Tic-Tac-Toe, Candy Crush Saga, and Tetris.

Patterns in GDP not only describe game mechanics but also exhibit structural relationships with other patterns: they can enable ("instantiate"), modify, or potentially conflict with the deployment of other patterns. As such, GDP provides a shared vocabulary for game designers to communicate, analyze, and create games.

---

[2] http://virt10.itu.chalmers.se/index.php/Alignment

However, the identification and verification of game design patterns is a highly specialized and expert-driven process. Despite the vast number of games, only a limited number of patterns have been formally proposed, and even fewer game–pattern relationships have been verified by experts. This results in an inherently sparse bipartite graph, making GDP an ideal testbed for evaluating augmentation strategies in edge-sparse domains.

# B  APPENDIX: IMPLEMENTATION SPECIFICS

## B.1  EXPERIMENTAL SETUP

**Hardware & Software:** CUDA 11.8, NVIDIA A40 GPU, PyTorch 2.4.0+cu118, PyTorch Geometric 2.7.0, Torch Sparse 0.6.18+pt24cu118.

Table 8: Hyperparameter configuration. TF-IDF dimensions are dataset-specific: Amazon (512), MovieLens (1024), GDP (128). Sensitivity analysis varies GNN type, drop ratio, and augmentation factor.

| Parameter | Value |
|---|---|
| *Data Split & Loading* | |
| Val / Test / Disjoint train ratio | 0.1 / 0.1 / 0.3154 |
| Negative sampling ratio | 1.4875 |
| Add negative train samples | True |
| Train / Val Neighbor sampling | [20, 10] / [20, 10] |
| Train / Val batch size | 128 / 64 |
| Train / Val shuffle | True / False |
| *Architecture & Training* | |
| Layers / Hidden dim / Link predictor | 3 / 768 / Dot |
| LR / Max epochs / Eval freq | $4.5 \times 10^{-4}$ / 779 / 15 |
| Early stopping (patience / $\delta$) | $10 / 10^{-3}$ |
| *Sensitivity Analysis* | |
| GNN architecture | GAT, GraphSAGE, GCN (GraphConv) |
| Edge drop ratio | 0.99, 0.95 |
| Augmentation factor | $1\times, 5\times$ |
| *Semantic-KNN* | |
| $k_g$, $k_p$ / sim_thresh / cap | 1, 1 / 0.6 / 4 |

# C  APPENDIX: SENSITIVITY ANALYSIS

## C.1  ANALYSIS DESIGN

### C.1.1  RETAIN RATES, AUGMENTATION FACTORS AND GNN ARCHITECTURES

The sensitivity analysis systematically evaluates edge augmentation methods across varying retain rates $q \in \{0.01, 0.05, 0.10\}$, augmentation factors $\phi \in \{1, 5, 100\}$, and GNN architectures (GAT, GraphSAGE, GCN). For the GCN baseline, we use `GraphConv` (Morris et al., 2019) [3] rather than the original spectral GCN (Kipf, 2016), as the latter's symmetric normalization is incompatible with PyTorch Geometric's heterogeneous graph conversion (`to_hetero()`). `GraphConv` performs mean aggregation and is functionally similar, as recommended by the PyG documentation for bipartite message passing[4].

Given the limited time and resource, we prioritize experiments that balance coverage across architectural variations and sparsity conditions. Table 9 summarizes the 36 completed experimental configurations across datasets.

---

[3] https://github.com/pyg-team/pytorch_geometric
[4] https://pytorch-geometric.readthedocs.io/en/latest/notes/heterogeneous.html

Table 9: Experimental configuration coverage across datasets, GNN architectures, retain rates ($q$), and augmentation factors ($\phi$). Checkmarks indicate completed experiments. Amazon and MovieLens use artificial sparsity via edge dropout; GDP is naturally sparse (no dropout, indicated by –). Factor $\phi = 1$ (baseline-only) experiments exist for all configurations but are excluded from appendix tables per design.

| Dataset | Architecture | $q$ | $\phi = 100$ | $\phi = 5$ | $\phi = 2$ |
|---|---|---|---|---|---|
| Amazon | GAT | 0.01 | ✓[†] | ✓ | ✓ |
| | GAT | 0.05 | – | ✓ | ✓ |
| | GAT | 0.10 | – | ✓ | ✓ |
| | GraphSAGE | 0.01 | – | ✓ | ✓ |
| | GraphSAGE | 0.05 | – | ✓ | ✓ |
| | GraphSAGE | 0.10 | – | ✓ | ✓ |
| | GCN | 0.01 | – | ✓ | ✓ |
| | GCN | 0.05 | – | ✓ | ✓ |
| | GCN | 0.10 | – | ✓ | ✓ |
| MovieLens | GAT | 0.01 | ✓[†] | ✓ | ✓ |
| | GAT | 0.05 | – | ✓ | ✓ |
| | GAT | 0.10 | – | ✓ | – |
| | GraphSAGE | 0.01 | – | ✓ | ✓ |
| | GCN | 0.01 | – | ✓ | ✓ |
| GDP | GAT | – | ✓[†] | ✓ | ✓ |
| | GraphSAGE | – | – | ✓ | ✓ |
| | GCN | – | – | ✓ | ✓ |

[†]Legacy $100\times$ experiments (Sept 2025) - no runtime statistics.

### C.1.2 DEGREE DISTRIBUTION ANALYSIS: METRICS AND RATIONALE

Beyond the primary metrics (AUC-ROC and Brier score), we report degree distribution analysis and runtime profiling to characterize how augmentation transforms graph structure and computational cost.

Edge dropping and augmentation fundamentally alters the graph's degree distribution, potentially affecting both model performance and the structural assumptions underlying GNN message passing. We quantify these changes using three complementary approaches: (i) summary statistics describing central tendency and dispersion, (ii) the Gini coefficient capturing degree inequality, and (iii) distribution fitting to distinguish scale-free from random graph structure.

**Summary Statistics.** For each original, sparsified (in the case of benchmark datasets) and augmented graph, we compute the mean degree $\bar{d} = \frac{1}{|V|} \sum_{v \in V} d_v$, standard deviation $\sigma_d$, median, minimum, and maximum degree, as well as the number of isolated nodes ($d_v = 0$). These statistics characterize how augmentation affects both the average connectivity and the presence of extreme values.

**Gini Coefficient.** The Gini coefficient (Ceriani & Verme, 2012), originally developed for income inequality measurement, has been adapted to quantify heterogeneity in network degree distributions (Hu & Wang, 2005; Badham, 2013). For a sorted degree sequence $(d_1, d_2, \ldots, d_n)$, the Gini coefficient is computed as:

$$G = \frac{2 \sum_{i=1}^{n} i \cdot d_i - (n+1) \sum_{i=1}^{n} d_i}{n \sum_{i=1}^{n} d_i} \tag{1}$$

where $G = 0$ indicates perfect equality (all nodes have identical degree) and $G = 1$ indicates maximal inequality (one node holds all edges). In the context of bipartite link prediction, a very high Gini coefficient ($G > 0.9$) indicates that most nodes have near-zero degree after percolation, concentrating supervision on a few hub nodes. AEGIS-Degree's inverse-degree bias explicitly targets this scenario by preferentially augmenting low-degree nodes, which should manifest as a reduction in $G$ relative to the baseline. Conversely, if $G$ becomes too low, over-smoothing may reduce discriminative structure.

**Distribution Fitting.** Many real-world networks exhibit heavy-tailed degree distributions, often claimed to follow power laws (Barabási & Albert, 1999). Following the rigorous statistical framework of Clauset et al. (2009), we fit three candidate distributions to the empirical degree sequence:

- **Power Law** (scale-free model): $P(d) \propto d^{-\alpha}$ for $d \geq d_{\min}$. The exponent $\alpha$ is estimated via maximum likelihood with automatic $d_{\min}$ selection. Scale-free networks exhibit hub nodes and preferential attachment dynamics.

- **Log-normal**: $P(d) \propto \frac{1}{d} \exp\left(-\frac{(\ln d - \mu)^2}{2\sigma^2}\right)$. Log-normal distributions arise from multiplicative growth processes and can fit many "scale-free" networks equally well or better (Broido & Clauset, 2019; Stumpf & Porter, 2012).

- **Poisson** (Erdős–Rényi model): $P(d) = \frac{\lambda^d e^{-\lambda}}{d!}$. The Poisson distribution characterizes classical random graphs $G(n, p)$ where $\lambda = np$ (ERDdS & R&wi, 1959). A good Poisson fit indicates augmentation produces ER-like random connectivity.

We compare fits using the Kolmogorov–Smirnov (KS) statistic, where lower values indicate better fit.

**Aggregation Protocol.** Since each of the 32 seeds produces a different sparse graph (due to distinct bond percolation realizations), we compute degree statistics independently for each seed and report aggregated results as $M \pm \mathrm{SD}$, consistent with our treatment of AUC-ROC and Brier scores. Distribution parameters ($\alpha$, $\mu$, $\sigma$, KS statistics) are likewise fitted per-seed and aggregated.

## C.2 OVERARCHING FINDINGS

Across datasets and augmentation methods, augmentation helps selectively and the best method depends on the metric: semantic_knn and synthetic drive AUC gains under extreme sparsity for Amazon (GAT/GCN), but semantic_knn is often neutral or negative for MovieLens and mixed for GDP; simple/degree_aware offer small, stable lifts in several settings (MovieLens GCN at $q{=}0.01$, Amazon GAT/GCN at higher $q$), while random rarely wins on AUC. Brier leaders differ: random/synthetic improve calibration for Amazon GraphSAGE and MovieLens GCN, semantic_knn leads GDP GAT at high $\phi$, and degree_aware/simple give modest Brier gains in multiple cases—highlighting a trade-off between ranking and calibration. Degree distributions show that most augmentations densify graphs and reduce isolation, with random/synthetic most effective at lowering Gini, whereas semantic_knn keeps graphs sparsest. Runtime overhead is negligible for augmentation (sub-0.16s) and training deltas stay small relative to dense-orig baselines, so performance differences are governed by graph quality rather than compute cost; practitioners should pick methods by desired metric (AUC vs. calibration) and acceptable shifts in degree structure.

## C.3 BENCHMARK DATASET - AMAZON

### C.3.1 SUMMARY

Across Amazon, graph augmentation helps when coupled with the right encoder: for GAT, semantic_knn and synthetic yield the largest AUC gains at extreme sparsity ($q{=}0.01$, $\phi{=}100$), and simple/degree_aware offer modest, more stable lifts at $q{=}0.10$, $\phi{=}2$. For GraphSAGE, AUC gains center on synthetic at $q{=}0.01$, while Brier is dominated by random (and synthetic at high dropout), revealing a trade-off between ranking quality and calibration. For GCN, semantic_knn is the clear AUC winner at $q{=}0.05$, whereas degree_aware and simple/random/synthetic drive the best Brier at higher $q$. Degree distributions remain sparse for most methods, with random/synthetic most effective at reducing inequality and isolated nodes, and semantic_knn keeping graphs sparsest. Runtime overheads are negligible across all augmentation methods (sub-0.07s augmentation; training within a few seconds of baseline), so observed gains or losses are driven by graph quality rather than compute cost.

### C.3.2 GAT

**Summary Analysis** Across Amazon GAT, AUC gains concentrate at high sparsity: `synthetic` and `semantic_knn` lead at $q$=0.01, $\phi$=100 (+0.101 and +0.091), while `simple`/`degree_aware` give smaller but significant lifts at $q$=0.10, $\phi$=2 (+0.017/+0.004); `random` is consistently weakest. Brier improvements are largest for `semantic_knn` at $q$=0.01, $\phi$=100 ($-0.015$) and for `simple` at $q$=0.10, $\phi$=2 ($-0.011$), with mild gains for `degree_aware`/`simple` at $q$=0.10, $\phi$=5 ($-0.009/-0.007$); `synthetic`/`random` often worsen Brier. Degree-wise, augmentations raise mean degree from the ultra-sparse baseline (0.033 at $q$=0.01, $\phi$=100) to $\approx$ 2.4 at $\phi$=100 (Gini 0.827–0.979) or to $\approx$ 0.6/0.48 at $\phi$=5/2 (Gini remains high), while `semantic_knn` keeps graphs sparsest (mean degree 0.106 at $\phi$=100) but slightly lowers inequality. Runtime overheads are negligible: augmentation $<$ 0.064s and training within a few seconds of baseline (versus $\sim$ 180s for the dense original), so accuracy differences stem from graph quality rather than cost.

**AUC and Brier Score**

Table 10: Amazon (product–category) GAT: AUC-ROC ($M \pm SD$) with paired $t$-tests vs. sparse baseline ($n = 32$ seeds). A higher AUC is better.

| $q$ | $\phi$ | Method | AUC $M \pm SD$ | $\Delta$AUC | $t(31)$ | $p$ | $d$ |
|---|---|---|---|---|---|---|---|
| 0.01 | 100× | baseline | $0.630 \pm 0.162$ | +0.000 | — | — | — |
| 0.01 | 100× | degree_aware | $0.650 \pm 0.204$ | +0.020$^{\text{ns}}$ | $-0.50$ | 0.619 | $-0.09$ |
| 0.01 | 100× | simple | $0.637 \pm 0.199$ | +0.007$^{\text{ns}}$ | $-0.17$ | 0.864 | $-0.03$ |
| 0.01 | 100× | semantic_knn | $0.722 \pm 0.197$ | +0.091$^{*}$ | $-2.40$ | 0.023 | $-0.42$ |
| 0.01 | 100× | **synthetic** | $0.732 \pm 0.181$ | +0.101$^{*}$ | $-2.48$ | 0.019 | $-0.44$ |
| 0.01 | 100× | random | $0.626 \pm 0.252$ | -0.004$^{\text{ns}}$ | 0.08 | 0.936 | +0.01 |
| 0.01 | 100× | original | $0.928 \pm 0.008$ | +0.298$^{***}$ | $-10.42$ | $<$0.001 | $-1.84$ |
| 0.01 | 5× | baseline | $0.648 \pm 0.188$ | +0.000 | — | — | — |
| 0.01 | 5× | degree_aware | $0.635 \pm 0.226$ | -0.013$^{\text{ns}}$ | 0.44 | 0.665 | +0.08 |
| 0.01 | 5× | simple | $0.625 \pm 0.221$ | -0.023$^{\text{ns}}$ | 0.79 | 0.435 | +0.14 |
| 0.01 | 5× | **semantic_knn** | $0.726 \pm 0.185$ | +0.078$^{\text{ns}}$ | $-1.80$ | 0.081 | $-0.32$ |
| 0.01 | 5× | synthetic | $0.613 \pm 0.248$ | -0.035$^{\text{ns}}$ | 0.83 | 0.410 | +0.15 |
| 0.01 | 5× | random | $0.611 \pm 0.199$ | -0.037$^{\text{ns}}$ | 0.91 | 0.368 | +0.16 |
| 0.01 | 5× | original | $0.929 \pm 0.009$ | +0.281$^{***}$ | $-8.40$ | $<$0.001 | $-1.49$ |
| 0.01 | 2× | **baseline** | $0.648 \pm 0.188$ | +0.000 | — | — | — |
| 0.01 | 2× | degree_aware | $0.628 \pm 0.237$ | -0.019$^{\text{ns}}$ | 0.70 | 0.488 | +0.12 |
| 0.01 | 2× | simple | $0.611 \pm 0.231$ | -0.036$^{\text{ns}}$ | 1.39 | 0.174 | +0.25 |
| 0.01 | 2× | semantic_knn | $0.617 \pm 0.237$ | -0.031$^{\text{ns}}$ | 0.95 | 0.347 | +0.17 |
| 0.01 | 2× | synthetic | $0.585 \pm 0.232$ | -0.063$^{\text{ns}}$ | 1.53 | 0.136 | +0.27 |
| 0.01 | 2× | random | $0.565 \pm 0.232$ | -0.083$^{*}$ | 2.38 | 0.023 | +0.42 |
| 0.01 | 2× | original | $0.929 \pm 0.009$ | +0.281$^{***}$ | $-8.40$ | $<$0.001 | $-1.49$ |
| 0.05 | 5× | baseline | $0.752 \pm 0.064$ | +0.000 | — | — | — |
| 0.05 | 5× | degree_aware | $0.755 \pm 0.063$ | +0.003$^{\text{ns}}$ | $-0.44$ | 0.665 | $-0.08$ |
| 0.05 | 5× | **simple** | $0.757 \pm 0.064$ | +0.005$^{\text{ns}}$ | $-0.79$ | 0.437 | $-0.14$ |
| 0.05 | 5× | semantic_knn | $0.730 \pm 0.071$ | -0.022$^{**}$ | 3.11 | 0.004 | +0.55 |
| 0.05 | 5× | synthetic | $0.646 \pm 0.074$ | -0.106$^{***}$ | 9.10 | $<$0.001 | +1.61 |
| 0.05 | 5× | random | $0.608 \pm 0.086$ | -0.144$^{***}$ | 9.68 | $<$0.001 | +1.71 |
| 0.05 | 5× | original | $0.929 \pm 0.009$ | +0.177$^{***}$ | $-14.93$ | $<$0.001 | $-2.64$ |
| 0.05 | 2× | **baseline** | $0.752 \pm 0.064$ | +0.000 | — | — | — |
| 0.05 | 2× | degree_aware | $0.752 \pm 0.067$ | -0.000$^{\text{ns}}$ | 0.06 | 0.953 | +0.01 |
| 0.05 | 2× | simple | $0.749 \pm 0.070$ | -0.003$^{\text{ns}}$ | 0.57 | 0.573 | +0.10 |
| 0.05 | 2× | semantic_knn | $0.735 \pm 0.070$ | -0.017$^{*}$ | 2.61 | 0.014 | +0.46 |
| 0.05 | 2× | synthetic | $0.712 \pm 0.067$ | -0.040$^{***}$ | 4.91 | $<$0.001 | +0.87 |
| 0.05 | 2× | random | $0.695 \pm 0.085$ | -0.057$^{***}$ | 4.78 | $<$0.001 | +0.84 |
| 0.05 | 2× | original | $0.929 \pm 0.009$ | +0.177$^{***}$ | $-14.93$ | $<$0.001 | $-2.64$ |

| $q$ | $\phi$ | Method | AUC $M \pm SD$ | $\Delta$AUC | $t(31)$ | $p$ | $d$ |
|---|---|---|---|---|---|---|---|
| 0.10 | 5× | baseline | $0.730 \pm 0.047$ | +0.000 | — | — | — |
| 0.10 | 5× | **degree_aware** | $0.745 \pm 0.051$ | +0.015** | −3.49 | 0.001 | −0.62 |
| 0.10 | 5× | simple | $0.745 \pm 0.036$ | +0.014** | −2.85 | 0.008 | −0.50 |
| 0.10 | 5× | semantic_knn | $0.732 \pm 0.043$ | +0.002$^{\text{ns}}$ | −0.21 | 0.837 | −0.04 |
| 0.10 | 5× | synthetic | $0.673 \pm 0.058$ | -0.057*** | 4.84 | <0.001 | +0.85 |
| 0.10 | 5× | random | $0.661 \pm 0.057$ | -0.070*** | 6.42 | <0.001 | +1.13 |
| 0.10 | 5× | original | $0.929 \pm 0.009$ | +0.199*** | −22.72 | <0.001 | −4.02 |
| 0.10 | 2× | baseline | $0.730 \pm 0.047$ | +0.000 | — | — | — |
| 0.10 | 2× | degree_aware | $0.734 \pm 0.040$ | +0.004$^{\text{ns}}$ | −0.74 | 0.463 | −0.13 |
| 0.10 | 2× | **simple** | $0.747 \pm 0.047$ | +0.017** | −3.15 | 0.004 | −0.56 |
| 0.10 | 2× | semantic_knn | $0.732 \pm 0.043$ | +0.002$^{\text{ns}}$ | −0.21 | 0.837 | −0.04 |
| 0.10 | 2× | synthetic | $0.708 \pm 0.048$ | -0.022** | 3.19 | 0.003 | +0.56 |
| 0.10 | 2× | random | $0.692 \pm 0.057$ | -0.038*** | 5.69 | <0.001 | +1.01 |
| 0.10 | 2× | original | $0.929 \pm 0.009$ | +0.199*** | −22.72 | <0.001 | −4.02 |

Table 11: Amazon (product–category) GAT: Brier Score ($M \pm SD$) with paired $t$-tests vs. sparse baseline ($n = 32$ seeds, lower is better).

| $q$ | $\phi$ | Method | Brier $M \pm SD$ | $\Delta$Brier | $t(31)$ | $p$ | $d$ |
|---|---|---|---|---|---|---|---|
| 0.01 | 100× | baseline | $0.249 \pm 0.048$ | +0.000 | — | — | — |
| 0.01 | 100× | degree_aware | $0.248 \pm 0.054$ | -0.001$^{\text{ns}}$ | 0.29 | 0.772 | +0.05 |
| 0.01 | 100× | simple | $0.248 \pm 0.049$ | -0.001$^{\text{ns}}$ | 0.30 | 0.765 | +0.05 |
| 0.01 | 100× | **semantic_knn** | $0.233 \pm 0.044$ | -0.015* | 2.19 | 0.036 | +0.39 |
| 0.01 | 100× | synthetic | $0.244 \pm 0.029$ | -0.005$^{\text{ns}}$ | 0.70 | 0.488 | +0.13 |
| 0.01 | 100× | random | $0.259 \pm 0.040$ | +0.010$^{\text{ns}}$ | −0.92 | 0.367 | −0.16 |
| 0.01 | 100× | original | $0.135 \pm 0.020$ | -0.114*** | 14.03 | <0.001 | +2.48 |
| 0.01 | 5× | baseline | $0.251 \pm 0.062$ | +0.000 | — | — | — |
| 0.01 | 5× | degree_aware | $0.245 \pm 0.053$ | -0.006$^{\text{ns}}$ | 1.85 | 0.074 | +0.33 |
| 0.01 | 5× | simple | $0.243 \pm 0.058$ | -0.008$^{\text{ns}}$ | 1.55 | 0.132 | +0.28 |
| 0.01 | 5× | **semantic_knn** | $0.236 \pm 0.048$ | -0.015$^{\text{ns}}$ | 1.61 | 0.117 | +0.29 |
| 0.01 | 5× | synthetic | $0.256 \pm 0.041$ | +0.005$^{\text{ns}}$ | −0.28 | 0.778 | −0.05 |
| 0.01 | 5× | random | $0.268 \pm 0.063$ | +0.017$^{\text{ns}}$ | −1.87 | 0.071 | −0.34 |
| 0.01 | 5× | original | $0.141 \pm 0.016$ | -0.110*** | 10.52 | <0.001 | +1.86 |
| 0.01 | 2× | baseline | $0.251 \pm 0.062$ | +0.000 | — | — | — |
| 0.01 | 2× | degree_aware | $0.254 \pm 0.066$ | +0.003$^{\text{ns}}$ | −0.35 | 0.730 | −0.06 |
| 0.01 | 2× | simple | $0.257 \pm 0.064$ | +0.006$^{\text{ns}}$ | −1.35 | 0.188 | −0.24 |
| 0.01 | 2× | **semantic_knn** | $0.247 \pm 0.047$ | -0.004$^{\text{ns}}$ | 0.94 | 0.353 | +0.17 |
| 0.01 | 2× | synthetic | $0.256 \pm 0.055$ | +0.005$^{\text{ns}}$ | −0.26 | 0.798 | −0.05 |
| 0.01 | 2× | random | $0.280 \pm 0.079$ | +0.029* | −2.38 | 0.024 | −0.43 |
| 0.01 | 2× | original | $0.141 \pm 0.016$ | -0.110*** | 10.52 | <0.001 | +1.86 |
| 0.05 | 5× | baseline | $0.238 \pm 0.022$ | +0.000 | — | — | — |
| 0.05 | 5× | degree_aware | $0.235 \pm 0.021$ | -0.003$^{\text{ns}}$ | 0.71 | 0.481 | +0.13 |
| 0.05 | 5× | **simple** | $0.234 \pm 0.027$ | -0.004$^{\text{ns}}$ | 1.09 | 0.285 | +0.19 |
| 0.05 | 5× | semantic_knn | $0.243 \pm 0.029$ | +0.006$^{\text{ns}}$ | −1.52 | 0.139 | −0.27 |
| 0.05 | 5× | synthetic | $0.256 \pm 0.024$ | +0.018*** | −4.52 | <0.001 | −0.80 |
| 0.05 | 5× | random | $0.272 \pm 0.028$ | +0.034*** | −8.69 | <0.001 | −1.54 |
| 0.05 | 5× | original | $0.141 \pm 0.016$ | -0.097*** | 20.57 | <0.001 | +3.64 |
| 0.05 | 2× | **baseline** | $0.238 \pm 0.022$ | +0.000 | — | — | — |
| 0.05 | 2× | degree_aware | $0.238 \pm 0.028$ | -0.000$^{\text{ns}}$ | 0.08 | 0.939 | +0.01 |
| 0.05 | 2× | simple | $0.238 \pm 0.031$ | -0.000$^{\text{ns}}$ | 0.02 | 0.984 | +0.00 |
| 0.05 | 2× | semantic_knn | $0.244 \pm 0.028$ | +0.006$^{\text{ns}}$ | −1.90 | 0.067 | −0.34 |
| 0.05 | 2× | synthetic | $0.242 \pm 0.028$ | +0.004$^{\text{ns}}$ | −1.02 | 0.317 | −0.18 |

| $q$ | $\phi$ | Method | Brier $M \pm SD$ | $\Delta$Brier | $t(31)$ | $p$ | $d$ |
|---|---|---|---|---|---|---|---|
| 0.05 | 2× | random | $0.245 \pm 0.031$ | $+0.007^{\text{ns}}$ | $-1.50$ | $0.145$ | $-0.26$ |
| 0.05 | 2× | original | $0.141 \pm 0.016$ | $-0.097^{***}$ | $20.57$ | $<0.001$ | $+3.64$ |
| 0.10 | 5× | baseline | $0.235 \pm 0.024$ | $+0.000$ | — | — | — |
| 0.10 | 5× | **degree_aware** | $0.226 \pm 0.023$ | $-0.009^{*}$ | $2.29$ | $0.029$ | $+0.41$ |
| 0.10 | 5× | simple | $0.228 \pm 0.023$ | $-0.007^{*}$ | $2.25$ | $0.032$ | $+0.40$ |
| 0.10 | 5× | semantic_knn | $0.230 \pm 0.022$ | $-0.005^{\text{ns}}$ | $1.16$ | $0.255$ | $+0.20$ |
| 0.10 | 5× | synthetic | $0.253 \pm 0.024$ | $+0.018^{**}$ | $-3.56$ | $0.001$ | $-0.63$ |
| 0.10 | 5× | random | $0.260 \pm 0.022$ | $+0.025^{***}$ | $-4.96$ | $<0.001$ | $-0.88$ |
| 0.10 | 5× | original | $0.141 \pm 0.016$ | $-0.094^{***}$ | $17.66$ | $<0.001$ | $+3.12$ |
| 0.10 | 2× | baseline | $0.235 \pm 0.024$ | $+0.000$ | — | — | — |
| 0.10 | 2× | degree_aware | $0.232 \pm 0.024$ | $-0.003^{\text{ns}}$ | $0.83$ | $0.411$ | $+0.15$ |
| 0.10 | 2× | **simple** | $0.224 \pm 0.027$ | $-0.011^{***}$ | $3.72$ | $<0.001$ | $+0.66$ |
| 0.10 | 2× | semantic_knn | $0.230 \pm 0.022$ | $-0.005^{\text{ns}}$ | $1.16$ | $0.255$ | $+0.20$ |
| 0.10 | 2× | synthetic | $0.240 \pm 0.025$ | $+0.005^{\text{ns}}$ | $-1.33$ | $0.193$ | $-0.24$ |
| 0.10 | 2× | random | $0.243 \pm 0.021$ | $+0.008^{*}$ | $-2.10$ | $0.044$ | $-0.37$ |
| 0.10 | 2× | original | $0.141 \pm 0.016$ | $-0.094^{***}$ | $17.66$ | $<0.001$ | $+3.12$ |

## Degree Distribution Analysis

Table 12: Amazon (product–category) GAT: Degree Distribution Statistics ($M \pm SD$, $n = 32$ seeds). Lower Gini coefficient indicates more uniform degree distribution.

| $q$ | $\phi$ | Method | Mean Degree | Gini Coeff. | Num. Isolated | Best Fit |
|---|---|---|---|---|---|---|
| 0.01 | 100× | baseline | $0.0334 \pm 0.0102$ | $0.968 \pm 0.010$ | $1417.3 \pm 14.4$ | lognormal |
| 0.01 | 100× | degree_aware | $2.4168 \pm 0.2940$ | $0.979 \pm 0.003$ | $1430.3 \pm 4.4$ | lognormal |
| 0.01 | 100× | simple | $2.4168 \pm 0.2940$ | $0.978 \pm 0.003$ | $1430.3 \pm 4.4$ | lognormal |
| 0.01 | 100× | semantic_knn | $0.1063 \pm 0.0100$ | $0.964 \pm 0.005$ | $1382.2 \pm 10.3$ | powerlaw |
| 0.01 | 100× | synthetic | $2.4168 \pm 0.2940$ | $0.827 \pm 0.023$ | $1130.6 \pm 41.5$ | lognormal |
| 0.01 | 100× | **random** | $2.4168 \pm 0.2940$ | $0.355 \pm 0.021$ | $136.8 \pm 40.9$ | powerlaw |
| 0.01 | 100× | original | $4.3051 \pm 0.0000$ | $0.219 \pm 0.000$ | $114.0 \pm 0.0$ | powerlaw |
| 0.01 | 5× | baseline | $0.0426 \pm 0.0054$ | $0.960 \pm 0.005$ | $1404.4 \pm 7.7$ | lognormal |
| 0.01 | 5× | degree_aware | $0.1208 \pm 0.0147$ | $0.982 \pm 0.002$ | $1430.3 \pm 4.4$ | powerlaw |
| 0.01 | 5× | simple | $0.1208 \pm 0.0147$ | $0.982 \pm 0.002$ | $1430.3 \pm 4.4$ | powerlaw |
| 0.01 | 5× | semantic_knn | $0.1060 \pm 0.0096$ | $0.963 \pm 0.005$ | $1381.6 \pm 10.1$ | powerlaw |
| 0.01 | 5× | synthetic | $0.1208 \pm 0.0147$ | $0.925 \pm 0.010$ | $1329.9 \pm 17.3$ | powerlaw |
| 0.01 | 5× | **random** | $0.1208 \pm 0.0147$ | $0.893 \pm 0.012$ | $1298.8 \pm 19.1$ | lognormal |
| 0.01 | 5× | original | $4.3051 \pm 0.0000$ | $0.219 \pm 0.000$ | $114.0 \pm 0.0$ | powerlaw |
| 0.01 | 2× | baseline | $0.0426 \pm 0.0054$ | $0.960 \pm 0.005$ | $1404.4 \pm 7.7$ | lognormal |
| 0.01 | 2× | degree_aware | $0.0483 \pm 0.0059$ | $0.982 \pm 0.002$ | $1430.3 \pm 4.4$ | powerlaw |
| 0.01 | 2× | simple | $0.0483 \pm 0.0059$ | $0.982 \pm 0.002$ | $1430.3 \pm 4.4$ | powerlaw |
| 0.01 | 2× | semantic_knn | $0.0483 \pm 0.0059$ | $0.978 \pm 0.003$ | $1419.5 \pm 6.0$ | powerlaw |
| 0.01 | 2× | synthetic | $0.0483 \pm 0.0059$ | $0.960 \pm 0.005$ | $1401.4 \pm 8.1$ | lognormal |
| 0.01 | 2× | **random** | $0.0483 \pm 0.0059$ | $0.954 \pm 0.006$ | $1396.2 \pm 8.4$ | lognormal |
| 0.01 | 2× | original | $4.3051 \pm 0.0000$ | $0.219 \pm 0.000$ | $114.0 \pm 0.0$ | powerlaw |
| 0.05 | 5× | baseline | $0.2153 \pm 0.0133$ | $0.824 \pm 0.011$ | $1181.3 \pm 16.7$ | lognormal |
| 0.05 | 5× | degree_aware | $0.5930 \pm 0.0365$ | $0.916 \pm 0.005$ | $1301.9 \pm 9.3$ | powerlaw |
| 0.05 | 5× | simple | $0.5930 \pm 0.0365$ | $0.915 \pm 0.005$ | $1301.9 \pm 9.3$ | powerlaw |
| 0.05 | 5× | semantic_knn | $0.2682 \pm 0.0140$ | $0.858 \pm 0.009$ | $1188.1 \pm 16.1$ | powerlaw |
| 0.05 | 5× | synthetic | $0.5930 \pm 0.0365$ | $0.718 \pm 0.016$ | $910.9 \pm 31.0$ | powerlaw |
| 0.05 | 5× | **random** | $0.5930 \pm 0.0365$ | $0.639 \pm 0.018$ | $811.7 \pm 33.7$ | powerlaw |
| 0.05 | 5× | original | $4.3051 \pm 0.0000$ | $0.219 \pm 0.000$ | $114.0 \pm 0.0$ | powerlaw |
| 0.05 | 2× | baseline | $0.2153 \pm 0.0133$ | $0.824 \pm 0.011$ | $1181.3 \pm 16.7$ | lognormal |

| $q$ | $\phi$ | Method | Mean Degree | Gini Coeff. | Num. Isolated | Best Fit |
|---|---|---|---|---|---|---|
| 0.05 | 2× | degree_aware | $0.2372 \pm 0.0146$ | $0.919 \pm 0.005$ | $1301.9 \pm 9.3$ | powerlaw |
| 0.05 | 2× | simple | $0.2372 \pm 0.0146$ | $0.920 \pm 0.004$ | $1301.9 \pm 9.3$ | powerlaw |
| 0.05 | 2× | semantic_knn | $0.2372 \pm 0.0146$ | $0.876 \pm 0.010$ | $1221.0 \pm 17.0$ | powerlaw |
| 0.05 | 2× | synthetic | $0.2372 \pm 0.0146$ | $0.829 \pm 0.011$ | $1174.2 \pm 17.8$ | powerlaw |
| 0.05 | 2× | **random** | $0.2372 \pm 0.0146$ | $0.810 \pm 0.012$ | $1156.3 \pm 18.2$ | lognormal |
| 0.05 | 2× | original | $4.3051 \pm 0.0000$ | $0.219 \pm 0.000$ | $114.0 \pm 0.0$ | powerlaw |
| 0.10 | 5× | baseline | $0.4323 \pm 0.0163$ | $0.703 \pm 0.012$ | $950.6 \pm 19.3$ | lognormal |
| 0.10 | 5× | degree_aware | $1.1873 \pm 0.0446$ | $0.841 \pm 0.006$ | $1154.6 \pm 11.2$ | powerlaw |
| 0.10 | 5× | simple | $1.1873 \pm 0.0446$ | $0.843 \pm 0.006$ | $1154.6 \pm 11.2$ | powerlaw |
| 0.10 | 5× | semantic_knn | $0.4346 \pm 0.0171$ | $0.760 \pm 0.010$ | $1007.2 \pm 17.0$ | powerlaw |
| 0.10 | 5× | synthetic | $1.1873 \pm 0.0446$ | $0.576 \pm 0.015$ | $572.2 \pm 27.0$ | powerlaw |
| 0.10 | 5× | **random** | $1.1873 \pm 0.0446$ | $0.489 \pm 0.013$ | $450.4 \pm 26.1$ | powerlaw |
| 0.10 | 5× | original | $4.3051 \pm 0.0000$ | $0.219 \pm 0.000$ | $114.0 \pm 0.0$ | powerlaw |
| 0.10 | 2× | baseline | $0.4323 \pm 0.0163$ | $0.703 \pm 0.012$ | $950.6 \pm 19.3$ | lognormal |
| 0.10 | 2× | degree_aware | $0.4749 \pm 0.0179$ | $0.848 \pm 0.006$ | $1154.6 \pm 11.2$ | powerlaw |
| 0.10 | 2× | simple | $0.4749 \pm 0.0179$ | $0.850 \pm 0.006$ | $1154.6 \pm 11.2$ | powerlaw |
| 0.10 | 2× | semantic_knn | $0.4346 \pm 0.0171$ | $0.760 \pm 0.010$ | $1007.2 \pm 17.0$ | powerlaw |
| 0.10 | 2× | synthetic | $0.4749 \pm 0.0179$ | $0.712 \pm 0.012$ | $941.3 \pm 17.4$ | powerlaw |
| 0.10 | 2× | **random** | $0.4749 \pm 0.0179$ | $0.685 \pm 0.014$ | $911.9 \pm 21.6$ | powerlaw |
| 0.10 | 2× | original | $4.3051 \pm 0.0000$ | $0.219 \pm 0.000$ | $114.0 \pm 0.0$ | powerlaw |

## Runtime Analysis

Table 13: Amazon (product–category) GAT: Runtime Statistics ($M \pm SD$, seconds, $n = 32$ seeds). Lower times are better.

| $q$ | $\phi$ | Method | Aug. Time (s) | Train Time (s) |
|---|---|---|---|---|
| 0.01 | 5× | baseline | $0.0000 \pm 0.0000$ | $3.95 \pm 0.41$ |
| 0.01 | 5× | degree_aware | $0.0013 \pm 0.0010$ | $3.98 \pm 1.26$ |
| 0.01 | 5× | simple | $0.0010 \pm 0.0001$ | $3.83 \pm 0.95$ |
| 0.01 | 5× | semantic_knn | $0.0116 \pm 0.0056$ | $3.85 \pm 0.87$ |
| 0.01 | 5× | synthetic | $0.0011 \pm 0.0004$ | $3.74 \pm 0.74$ |
| 0.01 | 5× | **random** | $0.0009 \pm 0.0001$ | $4.00 \pm 0.96$ |
| 0.01 | 5× | original | $0.0000 \pm 0.0000$ | $178.54 \pm 20.74$ |
| 0.01 | 2× | baseline | $0.0000 \pm 0.0000$ | $4.04 \pm 0.43$ |
| 0.01 | 2× | degree_aware | $0.0013 \pm 0.0005$ | $3.93 \pm 0.91$ |
| 0.01 | 2× | simple | $0.0011 \pm 0.0001$ | $3.94 \pm 0.90$ |
| 0.01 | 2× | semantic_knn | $0.0066 \pm 0.0084$ | $3.80 \pm 0.70$ |
| 0.01 | 2× | synthetic | $0.0013 \pm 0.0013$ | $3.81 \pm 0.71$ |
| 0.01 | 2× | **random** | $0.0010 \pm 0.0001$ | $3.89 \pm 0.78$ |
| 0.01 | 2× | original | $0.0000 \pm 0.0000$ | $180.52 \pm 20.72$ |
| 0.05 | 5× | baseline | $0.0000 \pm 0.0000$ | $9.81 \pm 4.34$ |
| 0.05 | 5× | degree_aware | $0.0014 \pm 0.0007$ | $9.93 \pm 3.32$ |
| 0.05 | 5× | simple | $0.0011 \pm 0.0001$ | $10.23 \pm 4.17$ |
| 0.05 | 5× | semantic_knn | $0.0358 \pm 0.0058$ | $9.66 \pm 3.27$ |
| 0.05 | 5× | synthetic | $0.0012 \pm 0.0003$ | $12.38 \pm 4.65$ |
| 0.05 | 5× | **random** | $0.0010 \pm 0.0001$ | $15.04 \pm 5.33$ |
| 0.05 | 5× | original | $0.0000 \pm 0.0000$ | $181.93 \pm 20.91$ |
| 0.05 | 2× | baseline | $0.0000 \pm 0.0000$ | $9.47 \pm 4.17$ |
| 0.05 | 2× | degree_aware | $0.0014 \pm 0.0012$ | $8.61 \pm 2.24$ |
| 0.05 | 2× | **simple** | $0.0010 \pm 0.0000$ | $9.77 \pm 5.23$ |
| 0.05 | 2× | semantic_knn | $0.0219 \pm 0.0068$ | $9.74 \pm 3.93$ |

| $q$ | $\phi$ | Method | Aug. Time (s) | Train Time (s) |
|---|---|---|---|---|
| 0.05 | 2× | synthetic | $0.0011 \pm 0.0006$ | $11.43 \pm 3.88$ |
| 0.05 | 2× | random | $0.0010 \pm 0.0000$ | $11.23 \pm 3.91$ |
| 0.05 | 2× | original | $0.0000 \pm 0.0000$ | $178.24 \pm 20.43$ |
| 0.10 | 5× | baseline | $0.0000 \pm 0.0000$ | $20.88 \pm 6.92$ |
| 0.10 | 5× | degree_aware | $0.0013 \pm 0.0001$ | $24.83 \pm 8.14$ |
| 0.10 | 5× | **simple** | $0.0011 \pm 0.0001$ | $24.59 \pm 8.47$ |
| 0.10 | 5× | semantic_knn | $0.0615 \pm 0.0026$ | $22.58 \pm 7.62$ |
| 0.10 | 5× | synthetic | $0.0011 \pm 0.0001$ | $24.20 \pm 7.16$ |
| 0.10 | 5× | random | $0.0012 \pm 0.0008$ | $24.25 \pm 6.23$ |
| 0.10 | 5× | original | $0.0000 \pm 0.0000$ | $178.61 \pm 20.40$ |
| 0.10 | 2× | baseline | $0.0000 \pm 0.0000$ | $21.17 \pm 7.02$ |
| 0.10 | 2× | degree_aware | $0.0012 \pm 0.0001$ | $23.30 \pm 8.60$ |
| 0.10 | 2× | **simple** | $0.0010 \pm 0.0001$ | $23.62 \pm 8.30$ |
| 0.10 | 2× | semantic_knn | $0.0636 \pm 0.0026$ | $22.85 \pm 7.70$ |
| 0.10 | 2× | synthetic | $0.0011 \pm 0.0001$ | $23.27 \pm 8.54$ |
| 0.10 | 2× | random | $0.0010 \pm 0.0001$ | $24.57 \pm 9.45$ |
| 0.10 | 2× | original | $0.0000 \pm 0.0000$ | $179.84 \pm 20.52$ |

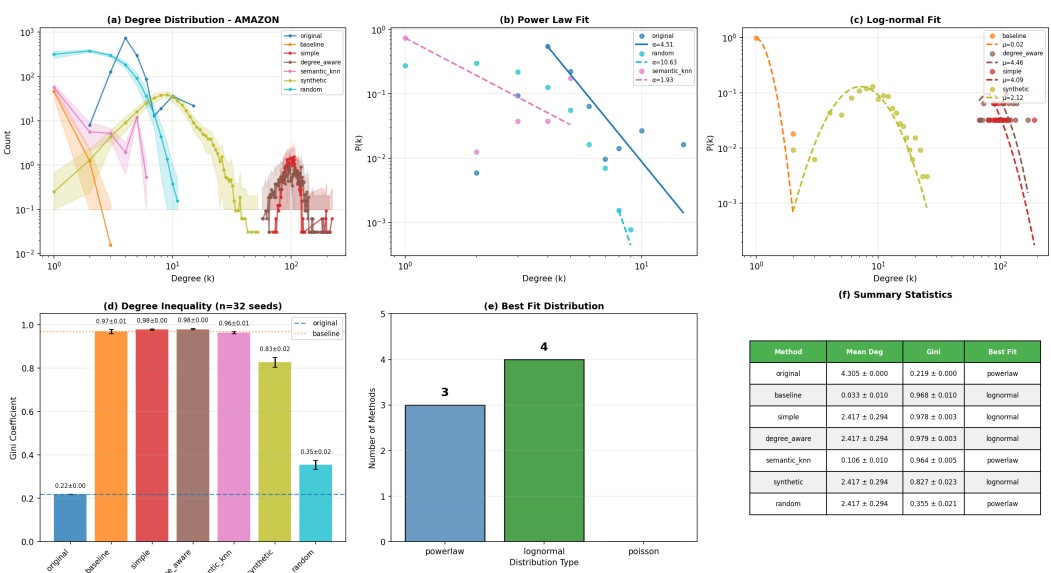

Figure 4: Amazon (product–category), GAT, $q$=0.01, $\phi$=100: Comprehensive analysis ($M \pm$ SD, $n = 32$ seeds) comparing baseline, augmentation methods, and original graph. Panel (a) shows degree distributions on log-log scale with confidence bands; (b) Power Law fits with exponent $\alpha$; (c) Log-normal fits with parameters $\mu$ and $\sigma$; (d) Gini coefficients quantifying degree inequality (lower = more uniform); (e) runtime comparison showing training time (left axis) and augmentation time (right axis, log scale); (f) best-fit distribution counts across methods.

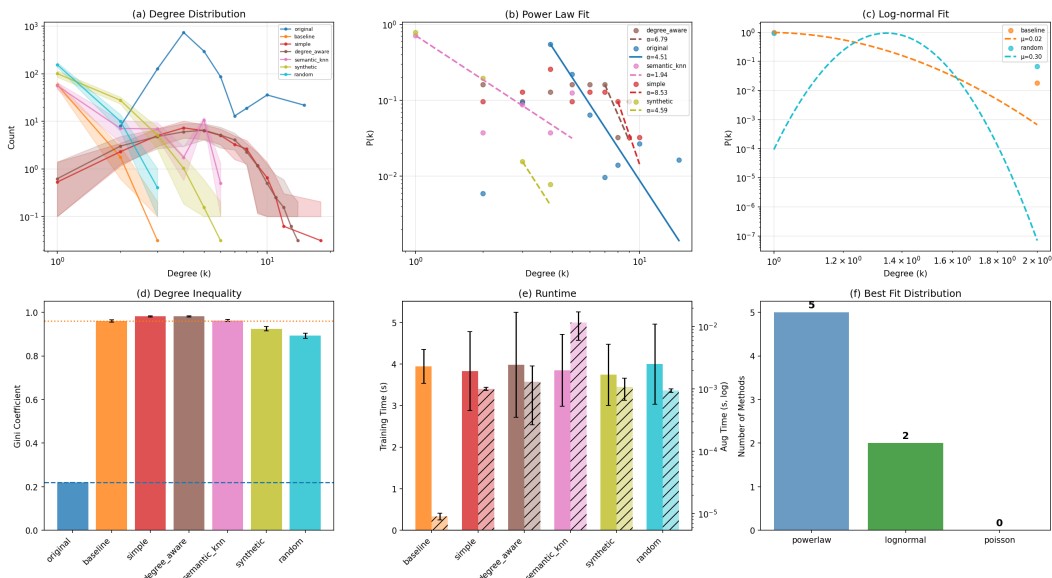

Figure 5: Amazon (product–category), GAT, $q$=0.01, $\phi$=5: Comprehensive analysis ($M \pm \mathrm{SD}$, $n = 32$ seeds) comparing baseline, augmentation methods, and original graph. Panel (a) shows degree distributions on log-log scale with confidence bands; (b) Power Law fits with exponent $\alpha$; (c) Log-normal fits with parameters $\mu$ and $\sigma$; (d) Gini coefficients quantifying degree inequality (lower = more uniform); (e) runtime comparison showing training time (left axis) and augmentation time (right axis, log scale); (f) best-fit distribution counts across methods.

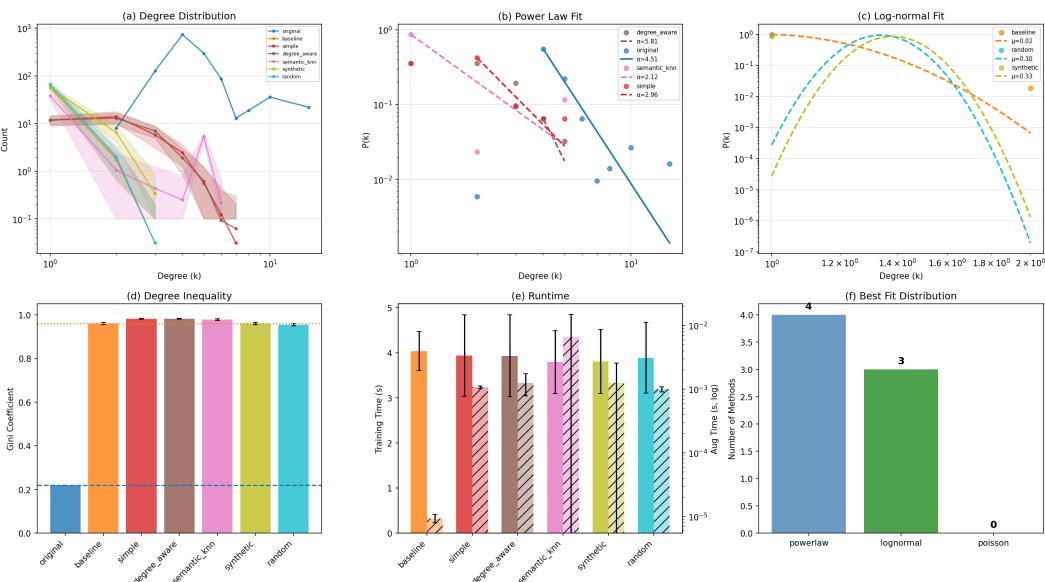

Figure 6: Amazon (product–category), GAT, $q$=0.01, $\phi$=2: Comprehensive analysis ($M \pm \mathrm{SD}$, $n = 32$ seeds) comparing baseline, augmentation methods, and original graph. Panel (a) shows degree distributions on log-log scale with confidence bands; (b) Power Law fits with exponent $\alpha$; (c) Log-normal fits with parameters $\mu$ and $\sigma$; (d) Gini coefficients quantifying degree inequality (lower = more uniform); (e) runtime comparison showing training time (left axis) and augmentation time (right axis, log scale); (f) best-fit distribution counts across methods.

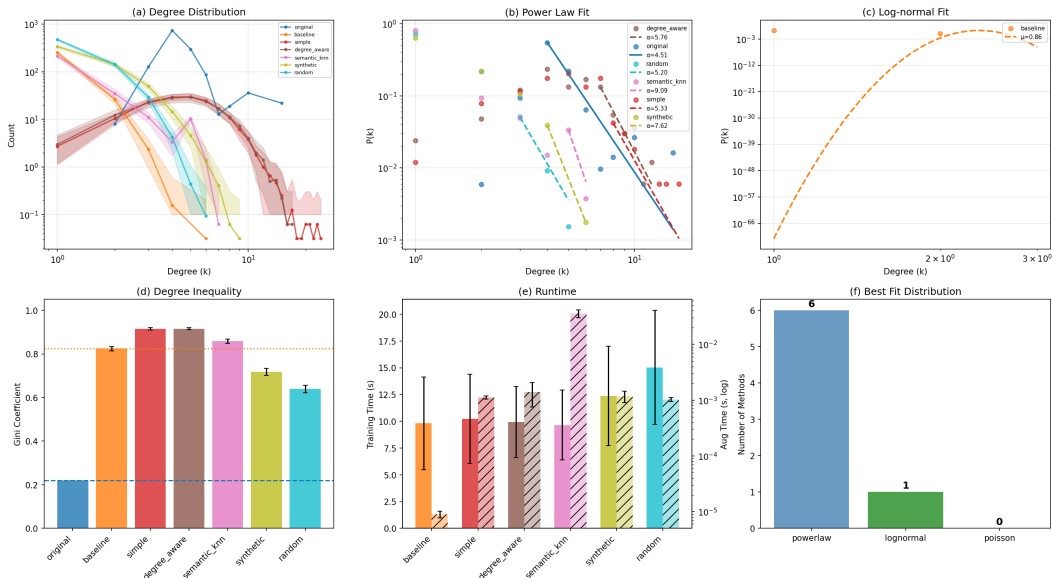

Figure 7: Amazon (product–category), GAT, $q$=0.05, $\phi$=5: Comprehensive analysis ($M \pm \mathrm{SD}$, $n = 32$ seeds) comparing baseline, augmentation methods, and original graph. Panel (a) shows degree distributions on log-log scale with confidence bands; (b) Power Law fits with exponent $\alpha$; (c) Log-normal fits with parameters $\mu$ and $\sigma$; (d) Gini coefficients quantifying degree inequality (lower = more uniform); (e) runtime comparison showing training time (left axis) and augmentation time (right axis, log scale); (f) best-fit distribution counts across methods.

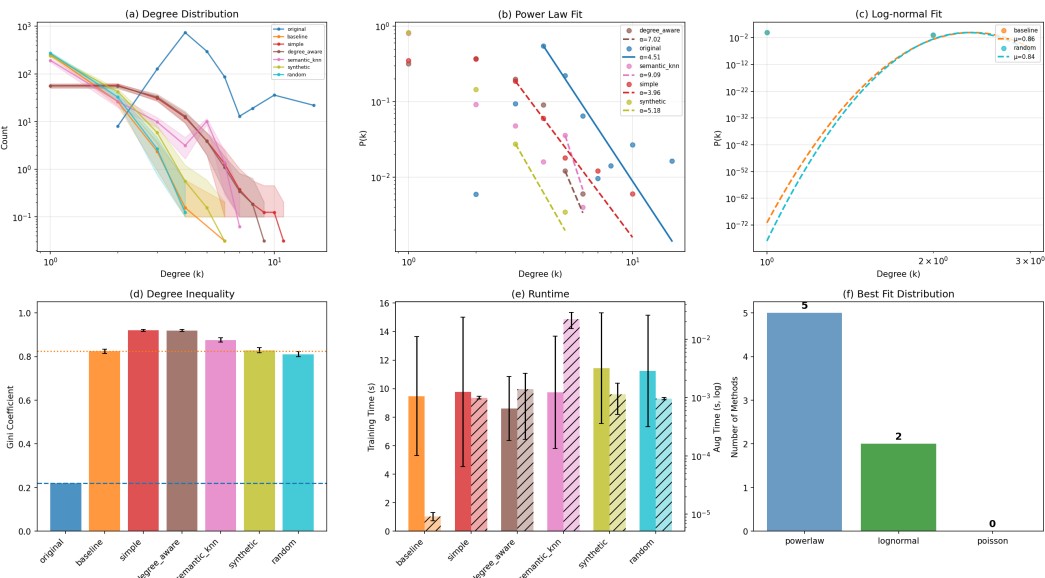

Figure 8: Amazon (product–category), GAT, $q$=0.05, $\phi$=2: Comprehensive analysis ($M \pm \mathrm{SD}$, $n = 32$ seeds) comparing baseline, augmentation methods, and original graph. Panel (a) shows degree distributions on log-log scale with confidence bands; (b) Power Law fits with exponent $\alpha$; (c) Log-normal fits with parameters $\mu$ and $\sigma$; (d) Gini coefficients quantifying degree inequality (lower = more uniform); (e) runtime comparison showing training time (left axis) and augmentation time (right axis, log scale); (f) best-fit distribution counts across methods.

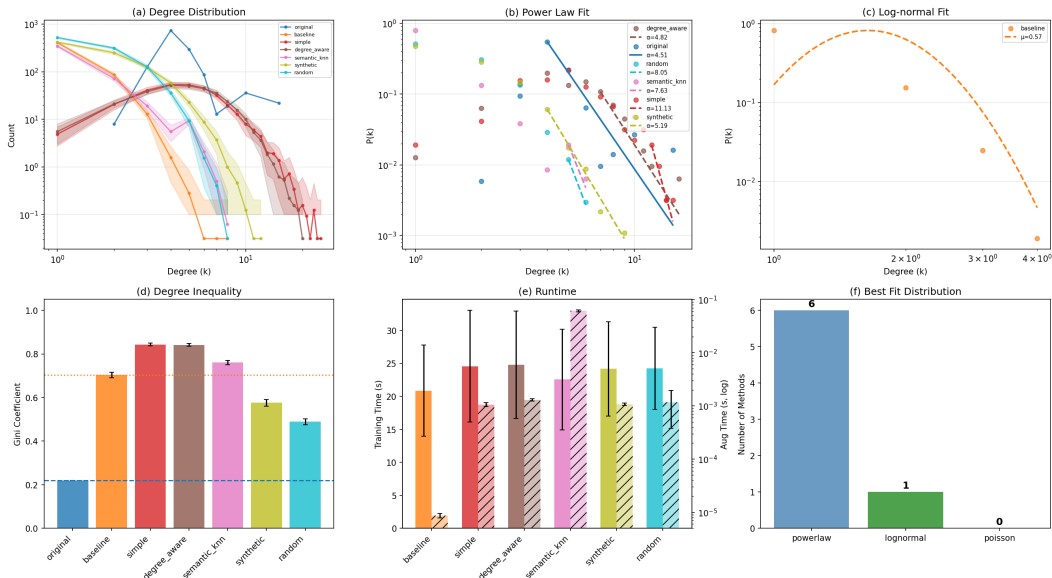

Figure 9: Amazon (product–category), GAT, $q$=0.10, $\phi$=5: Comprehensive analysis ($M \pm$ SD, $n = 32$ seeds) comparing baseline, augmentation methods, and original graph. Panel (a) shows degree distributions on log-log scale with confidence bands; (b) Power Law fits with exponent $\alpha$; (c) Log-normal fits with parameters $\mu$ and $\sigma$; (d) Gini coefficients quantifying degree inequality (lower = more uniform); (e) runtime comparison showing training time (left axis) and augmentation time (right axis, log scale); (f) best-fit distribution counts across methods.

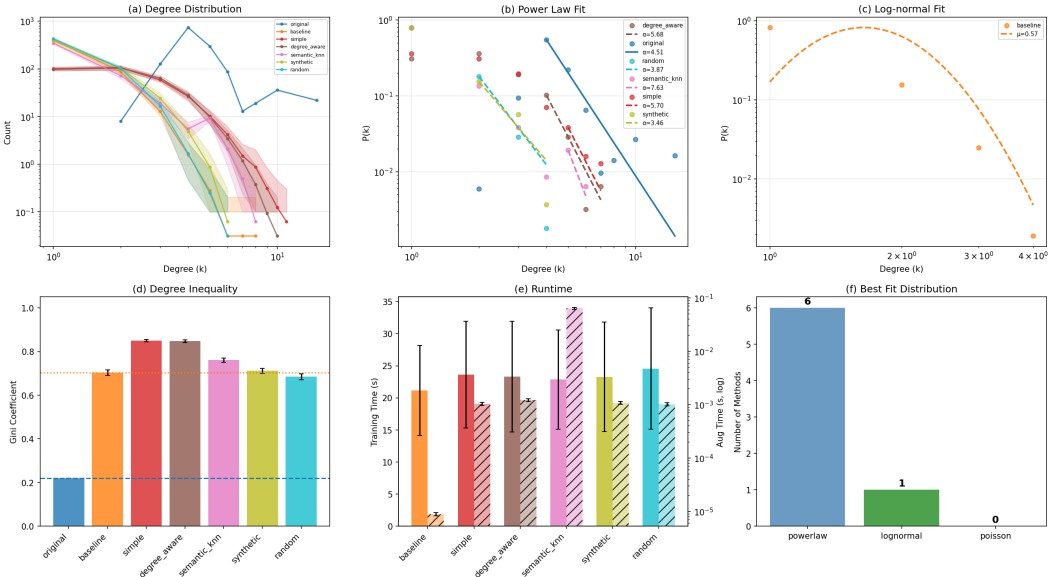

Figure 10: Amazon (product–category), GAT, $q$=0.10, $\phi$=2: Comprehensive analysis ($M \pm$ SD, $n = 32$ seeds) comparing baseline, augmentation methods, and original graph. Panel (a) shows degree distributions on log-log scale with confidence bands; (b) Power Law fits with exponent $\alpha$; (c) Log-normal fits with parameters $\mu$ and $\sigma$; (d) Gini coefficients quantifying degree inequality (lower = more uniform); (e) runtime comparison showing training time (left axis) and augmentation time (right axis, log scale); (f) best-fit distribution counts across methods.

### C.3.3 GRAPHSAGE

**Summary Analysis**  For Amazon GraphSAGE, `synthetic` leads AUC at $q$=0.01, $\phi$=5 (+0.077) and stays positive at $\phi$=2 (+0.036), while most other augmentations hover near baseline and

random is the weakest. Brier tells the opposite story: random delivers the largest reductions (e.g., $q$=0.10, $\phi$=5, $\Delta$Brier= $-0.092$; $q$=0.05, $\phi$=5, $-0.079$), with synthetic also improving at $q$=0.10, $\phi$=5 ($-0.072$); semantic_knn and degree_aware typically give modest or no Brier gains. Degree statistics show augmentations raising mean degree from the ultra-sparse baseline (0.043) to $\sim 0.12$ at $\phi$=5 and up to $\sim 1.19$ at $q$=0.10, $\phi$=5, while random/synthetic sharply lower Gini and isolated counts. Runtime overheads remain negligible: augmentation under 0.012s and training within a few seconds of baseline (vs. $\sim 134$s for the dense original), so differences are driven by graph quality rather than cost.

**AUC and Brier Score**

Table 14: Amazon (product–category) GraphSAGE: AUC-ROC ($M \pm SD$) with paired $t$-tests vs. sparse baseline ($n = 32$ seeds). A higher AUC is better.

| $q$ | $\phi$ | Method | AUC $M \pm SD$ | $\Delta$AUC | $t(31)$ | $p$ | $d$ |
|---|---|---|---|---|---|---|---|
| 0.01 | 5× | baseline | $0.507 \pm 0.153$ | +0.000 | — | — | — |
| 0.01 | 5× | degree_aware | $0.520 \pm 0.192$ | +0.012$^{\text{ns}}$ | $-0.40$ | 0.689 | $-0.07$ |
| 0.01 | 5× | simple | $0.494 \pm 0.194$ | -0.014$^{\text{ns}}$ | 0.42 | 0.675 | +0.07 |
| 0.01 | 5× | semantic_knn | $0.526 \pm 0.177$ | +0.019$^{\text{ns}}$ | $-0.65$ | 0.521 | $-0.11$ |
| 0.01 | 5× | **synthetic** | $0.584 \pm 0.170$ | +0.077* | $-2.34$ | 0.026 | $-0.41$ |
| 0.01 | 5× | random | $0.507 \pm 0.183$ | -0.000$^{\text{ns}}$ | 0.01 | 0.990 | +0.00 |
| 0.01 | 5× | original | $0.921 \pm 0.009$ | +0.413*** | $-15.34$ | <0.001 | $-2.71$ |
| 0.01 | 2× | baseline | $0.507 \pm 0.153$ | +0.000 | — | — | — |
| 0.01 | 2× | degree_aware | $0.516 \pm 0.177$ | +0.009$^{\text{ns}}$ | $-0.30$ | 0.766 | $-0.05$ |
| 0.01 | 2× | simple | $0.471 \pm 0.176$ | -0.036$^{\text{ns}}$ | 1.38 | 0.177 | +0.24 |
| 0.01 | 2× | semantic_knn | $0.522 \pm 0.163$ | +0.014$^{\text{ns}}$ | $-0.46$ | 0.646 | $-0.08$ |
| 0.01 | 2× | **synthetic** | $0.543 \pm 0.186$ | +0.036$^{\text{ns}}$ | $-1.10$ | 0.279 | $-0.19$ |
| 0.01 | 2× | random | $0.528 \pm 0.177$ | +0.020$^{\text{ns}}$ | $-0.54$ | 0.593 | $-0.10$ |
| 0.01 | 2× | original | $0.921 \pm 0.009$ | +0.413*** | $-15.34$ | <0.001 | $-2.71$ |
| 0.05 | 5× | baseline | $0.663 \pm 0.086$ | +0.000 | — | — | — |
| 0.05 | 5× | degree_aware | $0.649 \pm 0.082$ | -0.014$^{\text{ns}}$ | 1.18 | 0.247 | +0.21 |
| 0.05 | 5× | simple | $0.658 \pm 0.069$ | -0.005$^{\text{ns}}$ | 0.45 | 0.655 | +0.08 |
| 0.05 | 5× | **semantic_knn** | $0.677 \pm 0.079$ | +0.014$^{\text{ns}}$ | $-1.31$ | 0.198 | $-0.23$ |
| 0.05 | 5× | synthetic | $0.677 \pm 0.075$ | +0.014$^{\text{ns}}$ | $-1.32$ | 0.195 | $-0.23$ |
| 0.05 | 5× | random | $0.611 \pm 0.113$ | -0.051** | 2.98 | 0.006 | +0.53 |
| 0.05 | 5× | original | $0.921 \pm 0.009$ | +0.258*** | $-17.13$ | <0.001 | $-3.03$ |
| 0.05 | 2× | baseline | $0.663 \pm 0.086$ | +0.000 | — | — | — |
| 0.05 | 2× | degree_aware | $0.646 \pm 0.079$ | -0.016$^{\text{ns}}$ | 1.46 | 0.154 | +0.26 |
| 0.05 | 2× | simple | $0.632 \pm 0.074$ | -0.031** | 3.04 | 0.005 | +0.54 |
| 0.05 | 2× | **semantic_knn** | $0.668 \pm 0.085$ | +0.005$^{\text{ns}}$ | $-0.54$ | 0.591 | $-0.10$ |
| 0.05 | 2× | synthetic | $0.655 \pm 0.096$ | -0.008$^{\text{ns}}$ | 0.69 | 0.497 | +0.12 |
| 0.05 | 2× | random | $0.648 \pm 0.084$ | -0.014$^{\text{ns}}$ | 1.05 | 0.301 | +0.19 |
| 0.05 | 2× | original | $0.921 \pm 0.009$ | +0.258*** | $-17.13$ | <0.001 | $-3.03$ |
| 0.10 | 5× | **baseline** | $0.743 \pm 0.037$ | +0.000 | — | — | — |
| 0.10 | 5× | degree_aware | $0.730 \pm 0.038$ | -0.013$^{\text{ns}}$ | 1.95 | 0.061 | +0.34 |
| 0.10 | 5× | simple | $0.737 \pm 0.043$ | -0.005$^{\text{ns}}$ | 0.73 | 0.469 | +0.13 |
| 0.10 | 5× | semantic_knn | $0.733 \pm 0.038$ | -0.010$^{\text{ns}}$ | 1.49 | 0.147 | +0.26 |
| 0.10 | 5× | synthetic | $0.737 \pm 0.057$ | -0.006$^{\text{ns}}$ | 0.51 | 0.615 | +0.09 |
| 0.10 | 5× | random | $0.698 \pm 0.065$ | -0.045** | 3.54 | 0.001 | +0.63 |
| 0.10 | 5× | original | $0.921 \pm 0.009$ | +0.178*** | $-28.27$ | <0.001 | $-5.00$ |
| 0.10 | 2× | **baseline** | $0.743 \pm 0.037$ | +0.000 | — | — | — |
| 0.10 | 2× | degree_aware | $0.737 \pm 0.045$ | -0.006$^{\text{ns}}$ | 0.83 | 0.411 | +0.15 |
| 0.10 | 2× | simple | $0.740 \pm 0.049$ | -0.003$^{\text{ns}}$ | 0.32 | 0.752 | +0.06 |
| 0.10 | 2× | semantic_knn | $0.733 \pm 0.038$ | -0.010$^{\text{ns}}$ | 1.49 | 0.147 | +0.26 |
| 0.10 | 2× | synthetic | $0.738 \pm 0.038$ | -0.005$^{\text{ns}}$ | 0.65 | 0.521 | +0.11 |
| 0.10 | 2× | random | $0.730 \pm 0.050$ | -0.013$^{\text{ns}}$ | 1.45 | 0.158 | +0.26 |

| $q$ | $\phi$ | Method | AUC $M \pm SD$ | $\Delta$AUC | $t(31)$ | $p$ | $d$ |
|---|---|---|---|---|---|---|---|
| 0.10 | 2× | original | $0.921 \pm 0.009$ | $+0.178^{***}$ | $-28.27$ | $<0.001$ | $-5.00$ |

Table 15: Amazon (product–category) GraphSAGE: Brier Score ($M \pm SD$) with paired $t$-tests vs. sparse baseline ($n = 32$ seeds, lower is better).

| $q$ | $\phi$ | Method | Brier $M \pm SD$ | $\Delta$Brier | $t(31)$ | $p$ | $d$ |
|---|---|---|---|---|---|---|---|
| 0.01 | 5× | baseline | $0.397 \pm 0.041$ | $+0.000$ | — | — | — |
| 0.01 | 5× | degree_aware | $0.395 \pm 0.050$ | $-0.001^{\text{ns}}$ | $-0.01$ | $0.993$ | $-0.00$ |
| 0.01 | 5× | **simple** | $0.391 \pm 0.046$ | $-0.006^{\text{ns}}$ | $0.46$ | $0.646$ | $+0.08$ |
| 0.01 | 5× | semantic_knn | $0.399 \pm 0.059$ | $+0.002^{\text{ns}}$ | $-0.29$ | $0.777$ | $-0.05$ |
| 0.01 | 5× | synthetic | $0.398 \pm 0.039$ | $+0.001^{\text{ns}}$ | $-0.35$ | $0.729$ | $-0.06$ |
| 0.01 | 5× | random | $0.401 \pm 0.033$ | $+0.004^{\text{ns}}$ | $-0.68$ | $0.499$ | $-0.12$ |
| 0.01 | 5× | original | $0.105 \pm 0.011$ | $-0.292^{***}$ | $37.24$ | $<0.001$ | $+6.58$ |
| 0.01 | 2× | baseline | $0.397 \pm 0.041$ | $+0.000$ | — | — | — |
| 0.01 | 2× | degree_aware | $0.394 \pm 0.057$ | $-0.003^{\text{ns}}$ | $0.20$ | $0.847$ | $+0.04$ |
| 0.01 | 2× | simple | $0.408 \pm 0.036$ | $+0.012^{\text{ns}}$ | $-1.72$ | $0.095$ | $-0.31$ |
| 0.01 | 2× | semantic_knn | $0.402 \pm 0.044$ | $+0.005^{\text{ns}}$ | $-0.96$ | $0.344$ | $-0.17$ |
| 0.01 | 2× | **synthetic** | $0.393 \pm 0.047$ | $-0.004^{\text{ns}}$ | $0.31$ | $0.758$ | $+0.06$ |
| 0.01 | 2× | random | $0.398 \pm 0.046$ | $+0.001^{\text{ns}}$ | $-0.22$ | $0.829$ | $-0.04$ |
| 0.01 | 2× | original | $0.105 \pm 0.011$ | $-0.292^{***}$ | $37.24$ | $<0.001$ | $+6.58$ |
| 0.05 | 5× | baseline | $0.361 \pm 0.032$ | $+0.000$ | — | — | — |
| 0.05 | 5× | degree_aware | $0.367 \pm 0.035$ | $+0.005^{\text{ns}}$ | $-0.82$ | $0.419$ | $-0.14$ |
| 0.05 | 5× | simple | $0.366 \pm 0.031$ | $+0.005^{\text{ns}}$ | $-0.74$ | $0.468$ | $-0.13$ |
| 0.05 | 5× | semantic_knn | $0.361 \pm 0.031$ | $-0.000^{\text{ns}}$ | $0.03$ | $0.976$ | $+0.01$ |
| 0.05 | 5× | synthetic | $0.330 \pm 0.073$ | $-0.031^{*}$ | $2.56$ | $0.015$ | $+0.45$ |
| 0.05 | 5× | **random** | $0.282 \pm 0.072$ | $-0.079^{***}$ | $6.62$ | $<0.001$ | $+1.17$ |
| 0.05 | 5× | original | $0.105 \pm 0.011$ | $-0.256^{***}$ | $41.94$ | $<0.001$ | $+7.41$ |
| 0.05 | 2× | **baseline** | $0.361 \pm 0.032$ | $+0.000$ | — | — | — |
| 0.05 | 2× | degree_aware | $0.367 \pm 0.027$ | $+0.006^{\text{ns}}$ | $-1.01$ | $0.322$ | $-0.18$ |
| 0.05 | 2× | simple | $0.365 \pm 0.024$ | $+0.004^{\text{ns}}$ | $-0.76$ | $0.453$ | $-0.13$ |
| 0.05 | 2× | semantic_knn | $0.372 \pm 0.029$ | $+0.011^{*}$ | $-2.16$ | $0.038$ | $-0.38$ |
| 0.05 | 2× | synthetic | $0.375 \pm 0.028$ | $+0.014^{*}$ | $-2.43$ | $0.021$ | $-0.43$ |
| 0.05 | 2× | random | $0.369 \pm 0.036$ | $+0.008^{\text{ns}}$ | $-1.33$ | $0.194$ | $-0.23$ |
| 0.05 | 2× | original | $0.105 \pm 0.011$ | $-0.256^{***}$ | $41.94$ | $<0.001$ | $+7.41$ |
| 0.10 | 5× | baseline | $0.315 \pm 0.050$ | $+0.000$ | — | — | — |
| 0.10 | 5× | degree_aware | $0.315 \pm 0.056$ | $+0.000^{\text{ns}}$ | $-0.02$ | $0.986$ | $-0.00$ |
| 0.10 | 5× | simple | $0.319 \pm 0.041$ | $+0.004^{\text{ns}}$ | $-0.49$ | $0.627$ | $-0.09$ |
| 0.10 | 5× | semantic_knn | $0.305 \pm 0.042$ | $-0.009^{\text{ns}}$ | $1.07$ | $0.293$ | $+0.19$ |
| 0.10 | 5× | synthetic | $0.243 \pm 0.056$ | $-0.072^{***}$ | $6.29$ | $<0.001$ | $+1.11$ |
| 0.10 | 5× | **random** | $0.223 \pm 0.028$ | $-0.092^{***}$ | $9.04$ | $<0.001$ | $+1.60$ |
| 0.10 | 5× | original | $0.105 \pm 0.011$ | $-0.210^{***}$ | $24.31$ | $<0.001$ | $+4.30$ |
| 0.10 | 2× | baseline | $0.315 \pm 0.050$ | $+0.000$ | — | — | — |
| 0.10 | 2× | degree_aware | $0.313 \pm 0.045$ | $-0.001^{\text{ns}}$ | $0.13$ | $0.898$ | $+0.02$ |
| 0.10 | 2× | simple | $0.325 \pm 0.045$ | $+0.010^{\text{ns}}$ | $-0.87$ | $0.390$ | $-0.15$ |
| 0.10 | 2× | semantic_knn | $0.305 \pm 0.042$ | $-0.009^{\text{ns}}$ | $1.07$ | $0.293$ | $+0.19$ |
| 0.10 | 2× | synthetic | $0.286 \pm 0.047$ | $-0.029^{*}$ | $2.38$ | $0.023$ | $+0.42$ |
| 0.10 | 2× | **random** | $0.261 \pm 0.051$ | $-0.053^{***}$ | $3.95$ | $<0.001$ | $+0.70$ |
| 0.10 | 2× | original | $0.105 \pm 0.011$ | $-0.210^{***}$ | $24.31$ | $<0.001$ | $+4.30$ |

**Degree Distribution Analysis**

Table 16: Amazon (product–category) GraphSAGE: Degree Distribution Statistics ($M \pm SD$, $n = 32$ seeds). Lower Gini coefficient indicates more uniform degree distribution.

| $q$ | $\phi$ | Method | Mean Degree | Gini Coeff. | Num. Isolated | Best Fit |
|---|---|---|---|---|---|---|
| 0.01 | 5× | baseline | $0.0426 \pm 0.0054$ | $0.960 \pm 0.005$ | $1404.4 \pm 7.7$ | lognormal |
| 0.01 | 5× | degree_aware | $0.1208 \pm 0.0147$ | $0.982 \pm 0.002$ | $1430.3 \pm 4.4$ | powerlaw |
| 0.01 | 5× | simple | $0.1208 \pm 0.0147$ | $0.982 \pm 0.002$ | $1430.3 \pm 4.4$ | powerlaw |
| 0.01 | 5× | semantic_knn | $0.1060 \pm 0.0096$ | $0.963 \pm 0.005$ | $1381.6 \pm 10.1$ | powerlaw |
| 0.01 | 5× | synthetic | $0.1208 \pm 0.0147$ | $0.925 \pm 0.010$ | $1329.9 \pm 17.3$ | powerlaw |
| 0.01 | 5× | **random** | $0.1208 \pm 0.0147$ | $0.893 \pm 0.012$ | $1298.8 \pm 19.1$ | lognormal |
| 0.01 | 5× | original | $4.3051 \pm 0.0000$ | $0.219 \pm 0.000$ | $114.0 \pm 0.0$ | powerlaw |
| 0.01 | 2× | baseline | $0.0426 \pm 0.0054$ | $0.960 \pm 0.005$ | $1404.4 \pm 7.7$ | lognormal |
| 0.01 | 2× | degree_aware | $0.0483 \pm 0.0059$ | $0.982 \pm 0.002$ | $1430.3 \pm 4.4$ | powerlaw |
| 0.01 | 2× | simple | $0.0483 \pm 0.0059$ | $0.982 \pm 0.002$ | $1430.3 \pm 4.4$ | powerlaw |
| 0.01 | 2× | semantic_knn | $0.0483 \pm 0.0059$ | $0.978 \pm 0.003$ | $1419.5 \pm 6.0$ | powerlaw |
| 0.01 | 2× | synthetic | $0.0483 \pm 0.0059$ | $0.960 \pm 0.005$ | $1401.4 \pm 8.1$ | lognormal |
| 0.01 | 2× | **random** | $0.0483 \pm 0.0059$ | $0.954 \pm 0.006$ | $1396.2 \pm 8.4$ | lognormal |
| 0.01 | 2× | original | $4.3051 \pm 0.0000$ | $0.219 \pm 0.000$ | $114.0 \pm 0.0$ | powerlaw |
| 0.05 | 5× | baseline | $0.2153 \pm 0.0133$ | $0.824 \pm 0.011$ | $1181.3 \pm 16.7$ | lognormal |
| 0.05 | 5× | degree_aware | $0.5930 \pm 0.0365$ | $0.916 \pm 0.005$ | $1301.9 \pm 9.3$ | powerlaw |
| 0.05 | 5× | simple | $0.5930 \pm 0.0365$ | $0.915 \pm 0.005$ | $1301.9 \pm 9.3$ | powerlaw |
| 0.05 | 5× | semantic_knn | $0.2682 \pm 0.0140$ | $0.858 \pm 0.009$ | $1188.1 \pm 16.1$ | powerlaw |
| 0.05 | 5× | synthetic | $0.5930 \pm 0.0365$ | $0.718 \pm 0.016$ | $910.9 \pm 31.0$ | powerlaw |
| 0.05 | 5× | **random** | $0.5930 \pm 0.0365$ | $0.639 \pm 0.018$ | $811.7 \pm 33.7$ | powerlaw |
| 0.05 | 5× | original | $4.3051 \pm 0.0000$ | $0.219 \pm 0.000$ | $114.0 \pm 0.0$ | powerlaw |
| 0.05 | 2× | baseline | $0.2153 \pm 0.0133$ | $0.824 \pm 0.011$ | $1181.3 \pm 16.7$ | lognormal |
| 0.05 | 2× | degree_aware | $0.2372 \pm 0.0146$ | $0.919 \pm 0.005$ | $1301.9 \pm 9.3$ | powerlaw |
| 0.05 | 2× | simple | $0.2372 \pm 0.0146$ | $0.920 \pm 0.004$ | $1301.9 \pm 9.3$ | powerlaw |
| 0.05 | 2× | semantic_knn | $0.2372 \pm 0.0146$ | $0.876 \pm 0.010$ | $1221.0 \pm 17.0$ | powerlaw |
| 0.05 | 2× | synthetic | $0.2372 \pm 0.0146$ | $0.829 \pm 0.011$ | $1174.2 \pm 17.8$ | powerlaw |
| 0.05 | 2× | **random** | $0.2372 \pm 0.0146$ | $0.810 \pm 0.012$ | $1156.3 \pm 18.2$ | lognormal |
| 0.05 | 2× | original | $4.3051 \pm 0.0000$ | $0.219 \pm 0.000$ | $114.0 \pm 0.0$ | powerlaw |
| 0.10 | 5× | baseline | $0.4323 \pm 0.0163$ | $0.703 \pm 0.012$ | $950.6 \pm 19.3$ | lognormal |
| 0.10 | 5× | degree_aware | $1.1873 \pm 0.0446$ | $0.841 \pm 0.006$ | $1154.6 \pm 11.2$ | powerlaw |
| 0.10 | 5× | simple | $1.1873 \pm 0.0446$ | $0.843 \pm 0.006$ | $1154.6 \pm 11.2$ | powerlaw |
| 0.10 | 5× | semantic_knn | $0.4346 \pm 0.0171$ | $0.760 \pm 0.010$ | $1007.2 \pm 17.0$ | powerlaw |
| 0.10 | 5× | synthetic | $1.1873 \pm 0.0446$ | $0.576 \pm 0.015$ | $572.2 \pm 27.0$ | powerlaw |
| 0.10 | 5× | **random** | $1.1873 \pm 0.0446$ | $0.489 \pm 0.013$ | $450.4 \pm 26.1$ | powerlaw |
| 0.10 | 5× | original | $4.3051 \pm 0.0000$ | $0.219 \pm 0.000$ | $114.0 \pm 0.0$ | powerlaw |
| 0.10 | 2× | baseline | $0.4323 \pm 0.0163$ | $0.703 \pm 0.012$ | $950.6 \pm 19.3$ | lognormal |
| 0.10 | 2× | degree_aware | $0.4749 \pm 0.0179$ | $0.848 \pm 0.006$ | $1154.6 \pm 11.2$ | powerlaw |
| 0.10 | 2× | simple | $0.4749 \pm 0.0179$ | $0.850 \pm 0.006$ | $1154.6 \pm 11.2$ | powerlaw |
| 0.10 | 2× | semantic_knn | $0.4346 \pm 0.0171$ | $0.760 \pm 0.010$ | $1007.2 \pm 17.0$ | powerlaw |
| 0.10 | 2× | synthetic | $0.4749 \pm 0.0179$ | $0.712 \pm 0.012$ | $941.3 \pm 17.4$ | powerlaw |
| 0.10 | 2× | **random** | $0.4749 \pm 0.0179$ | $0.685 \pm 0.014$ | $911.9 \pm 21.6$ | powerlaw |
| 0.10 | 2× | original | $4.3051 \pm 0.0000$ | $0.219 \pm 0.000$ | $114.0 \pm 0.0$ | powerlaw |

**Runtime Analysis**

Table 17: Amazon (product–category) GraphSAGE: Runtime Statistics ($M \pm SD$, seconds, $n = 32$ seeds). Lower times are better.

| $q$ | $\phi$ | Method | Aug. Time (s) | Train Time (s) |
|---|---|---|---|---|
| 0.01 | 5× | baseline | $0.0000 \pm 0.0000$ | $4.08 \pm 1.40$ |
| 0.01 | 5× | degree_aware | $0.0013 \pm 0.0007$ | $3.97 \pm 1.62$ |
| 0.01 | 5× | simple | $0.0012 \pm 0.0009$ | $3.65 \pm 1.41$ |
| 0.01 | 5× | semantic_knn | $0.0117 \pm 0.0067$ | $4.11 \pm 1.91$ |
| 0.01 | 5× | synthetic | $0.0011 \pm 0.0004$ | $4.55 \pm 1.44$ |
| 0.01 | 5× | **random** | $0.0010 \pm 0.0001$ | $3.94 \pm 1.23$ |
| 0.01 | 5× | original | $0.0000 \pm 0.0000$ | $133.71 \pm 11.00$ |
| 0.01 | 2× | baseline | $0.0000 \pm 0.0000$ | $4.02 \pm 1.36$ |
| 0.01 | 2× | degree_aware | $0.0013 \pm 0.0008$ | $3.64 \pm 1.54$ |
| 0.01 | 2× | simple | $0.0010 \pm 0.0000$ | $3.83 \pm 1.72$ |
| 0.01 | 2× | semantic_knn | $0.0050 \pm 0.0011$ | $3.68 \pm 1.29$ |
| 0.01 | 2× | synthetic | $0.0011 \pm 0.0004$ | $3.70 \pm 1.51$ |
| 0.01 | 2× | **random** | $0.0009 \pm 0.0000$ | $4.00 \pm 1.84$ |
| 0.01 | 2× | original | $0.0000 \pm 0.0000$ | $132.37 \pm 11.42$ |
| 0.05 | 5× | baseline | $0.0000 \pm 0.0000$ | $6.55 \pm 1.25$ |
| 0.05 | 5× | degree_aware | $0.0012 \pm 0.0001$ | $7.16 \pm 3.75$ |
| 0.05 | 5× | **simple** | $0.0010 \pm 0.0000$ | $7.62 \pm 2.02$ |
| 0.05 | 5× | semantic_knn | $0.0343 \pm 0.0023$ | $7.68 \pm 2.54$ |
| 0.05 | 5× | synthetic | $0.0010 \pm 0.0000$ | $7.71 \pm 2.24$ |
| 0.05 | 5× | random | $0.0010 \pm 0.0000$ | $6.29 \pm 1.14$ |
| 0.05 | 5× | original | $0.0000 \pm 0.0000$ | $133.17 \pm 11.53$ |
| 0.05 | 2× | baseline | $0.0000 \pm 0.0000$ | $6.72 \pm 1.28$ |
| 0.05 | 2× | degree_aware | $0.0014 \pm 0.0008$ | $7.56 \pm 3.10$ |
| 0.05 | 2× | simple | $0.0011 \pm 0.0001$ | $7.42 \pm 2.51$ |
| 0.05 | 2× | semantic_knn | $0.0224 \pm 0.0082$ | $8.51 \pm 3.91$ |
| 0.05 | 2× | synthetic | $0.0014 \pm 0.0016$ | $8.11 \pm 2.11$ |
| 0.05 | 2× | **random** | $0.0010 \pm 0.0001$ | $8.21 \pm 2.37$ |
| 0.05 | 2× | original | $0.0000 \pm 0.0000$ | $134.67 \pm 11.76$ |
| 0.10 | 5× | baseline | $0.0000 \pm 0.0000$ | $15.43 \pm 5.57$ |
| 0.10 | 5× | degree_aware | $0.0013 \pm 0.0001$ | $16.61 \pm 6.89$ |
| 0.10 | 5× | simple | $0.0011 \pm 0.0001$ | $15.24 \pm 3.81$ |
| 0.10 | 5× | semantic_knn | $0.0624 \pm 0.0026$ | $13.80 \pm 5.13$ |
| 0.10 | 5× | synthetic | $0.0011 \pm 0.0001$ | $13.90 \pm 2.43$ |
| 0.10 | 5× | **random** | $0.0010 \pm 0.0001$ | $13.15 \pm 1.35$ |
| 0.10 | 5× | original | $0.0000 \pm 0.0000$ | $133.62 \pm 11.59$ |
| 0.10 | 2× | baseline | $0.0000 \pm 0.0000$ | $15.21 \pm 5.53$ |
| 0.10 | 2× | degree_aware | $0.0014 \pm 0.0016$ | $14.29 \pm 6.33$ |
| 0.10 | 2× | **simple** | $0.0010 \pm 0.0001$ | $15.29 \pm 4.83$ |
| 0.10 | 2× | semantic_knn | $0.0626 \pm 0.0052$ | $13.57 \pm 5.02$ |
| 0.10 | 2× | synthetic | $0.0011 \pm 0.0005$ | $12.98 \pm 4.72$ |
| 0.10 | 2× | random | $0.0010 \pm 0.0000$ | $11.62 \pm 3.01$ |
| 0.10 | 2× | original | $0.0000 \pm 0.0000$ | $132.70 \pm 11.33$ |

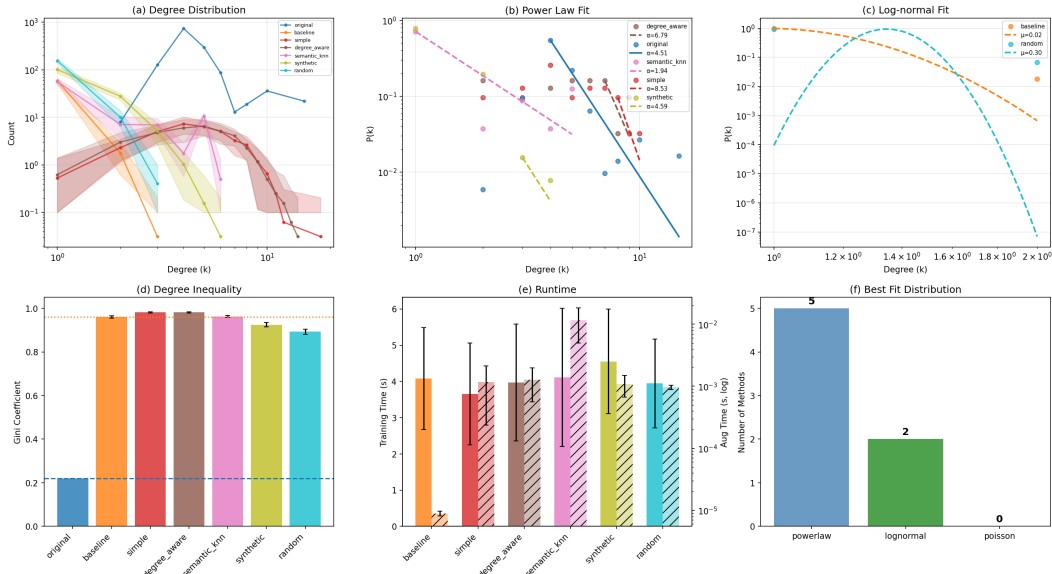

Figure 11: Amazon (product–category), GraphSAGE, $q$=0.01, $\phi$=5: Comprehensive analysis ($M \pm$ SD, $n = 32$ seeds) comparing baseline, augmentation methods, and original graph. Panel (a) shows degree distributions on log-log scale with confidence bands; (b) Power Law fits with exponent $\alpha$; (c) Log-normal fits with parameters $\mu$ and $\sigma$; (d) Gini coefficients quantifying degree inequality (lower = more uniform); (e) runtime comparison showing training time (left axis) and augmentation time (right axis, log scale); (f) best-fit distribution counts across methods.

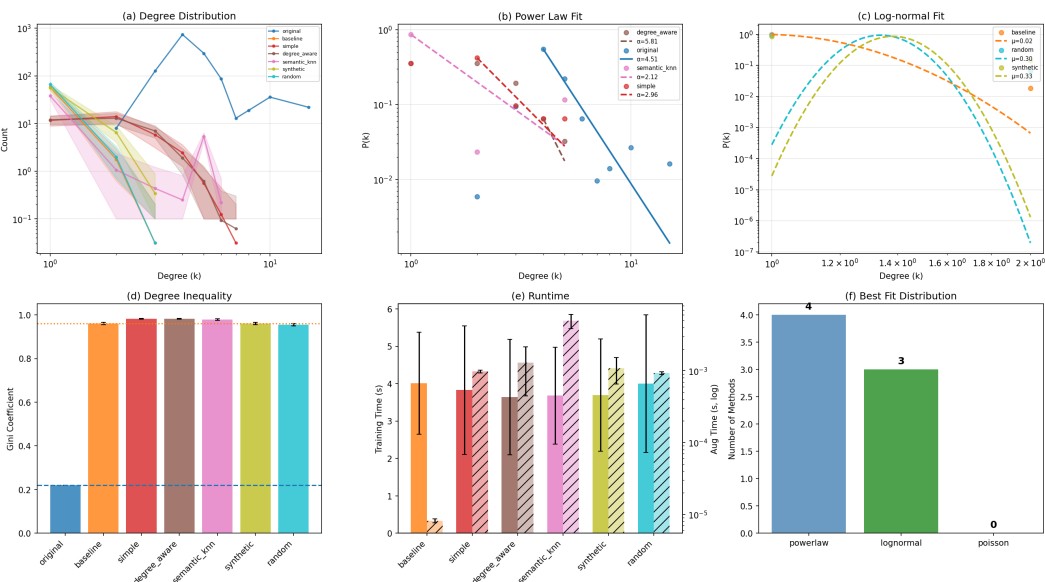

Figure 12: Amazon (product–category), GraphSAGE, $q$=0.01, $\phi$=2: Comprehensive analysis ($M \pm$ SD, $n = 32$ seeds) comparing baseline, augmentation methods, and original graph. Panel (a) shows degree distributions on log-log scale with confidence bands; (b) Power Law fits with exponent $\alpha$; (c) Log-normal fits with parameters $\mu$ and $\sigma$; (d) Gini coefficients quantifying degree inequality (lower = more uniform); (e) runtime comparison showing training time (left axis) and augmentation time (right axis, log scale); (f) best-fit distribution counts across methods.

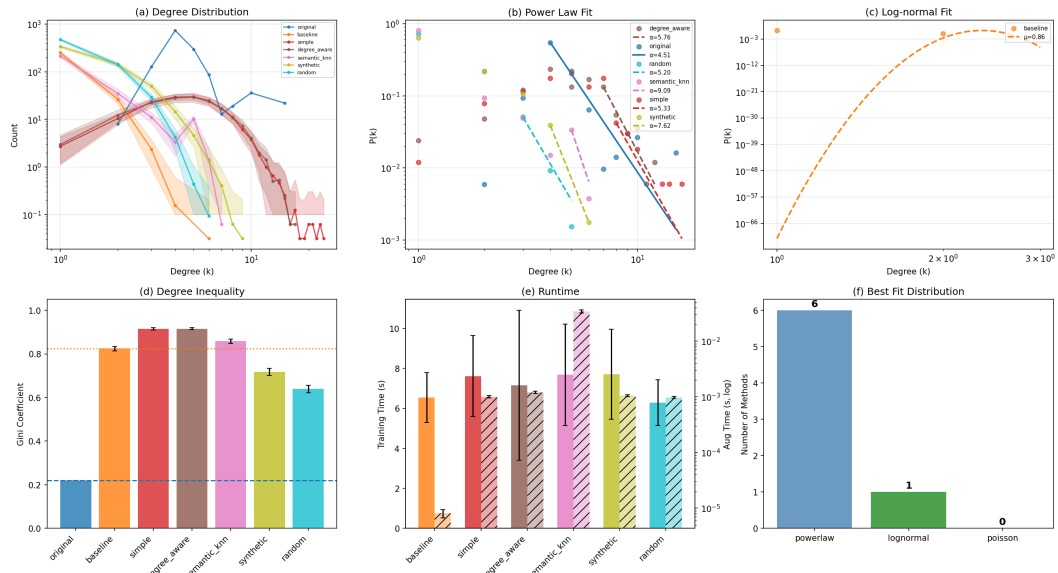

Figure 13: Amazon (product–category), GraphSAGE, $q$=0.05, $\phi$=5: Comprehensive analysis ($M \pm$ SD, $n = 32$ seeds) comparing baseline, augmentation methods, and original graph. Panel (a) shows degree distributions on log-log scale with confidence bands; (b) Power Law fits with exponent $\alpha$; (c) Log-normal fits with parameters $\mu$ and $\sigma$; (d) Gini coefficients quantifying degree inequality (lower = more uniform); (e) runtime comparison showing training time (left axis) and augmentation time (right axis, log scale); (f) best-fit distribution counts across methods.

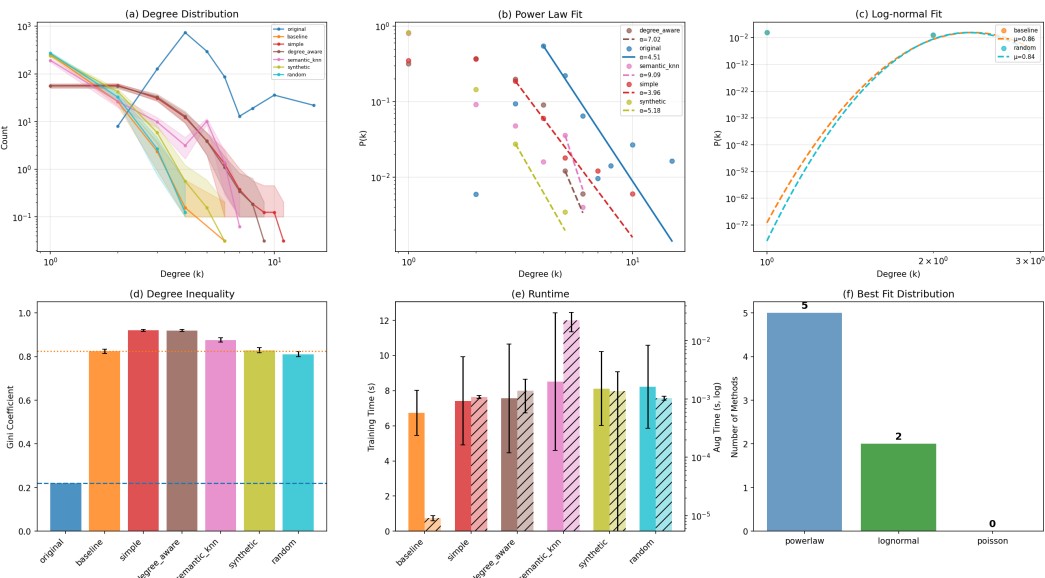

Figure 14: Amazon (product–category), GraphSAGE, $q$=0.05, $\phi$=2: Comprehensive analysis ($M \pm$ SD, $n = 32$ seeds) comparing baseline, augmentation methods, and original graph. Panel (a) shows degree distributions on log-log scale with confidence bands; (b) Power Law fits with exponent $\alpha$; (c) Log-normal fits with parameters $\mu$ and $\sigma$; (d) Gini coefficients quantifying degree inequality (lower = more uniform); (e) runtime comparison showing training time (left axis) and augmentation time (right axis, log scale); (f) best-fit distribution counts across methods.

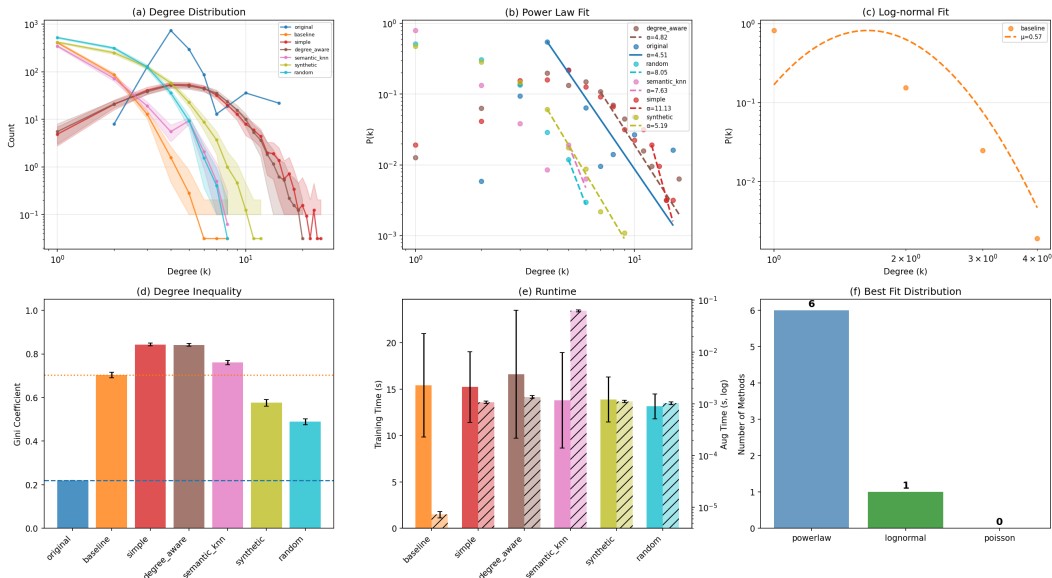

Figure 15: Amazon (product–category), GraphSAGE, $q$=0.10, $\phi$=5: Comprehensive analysis ($M \pm$ SD, $n = 32$ seeds) comparing baseline, augmentation methods, and original graph. Panel (a) shows degree distributions on log-log scale with confidence bands; (b) Power Law fits with exponent $\alpha$; (c) Log-normal fits with parameters $\mu$ and $\sigma$; (d) Gini coefficients quantifying degree inequality (lower = more uniform); (e) runtime comparison showing training time (left axis) and augmentation time (right axis, log scale); (f) best-fit distribution counts across methods.

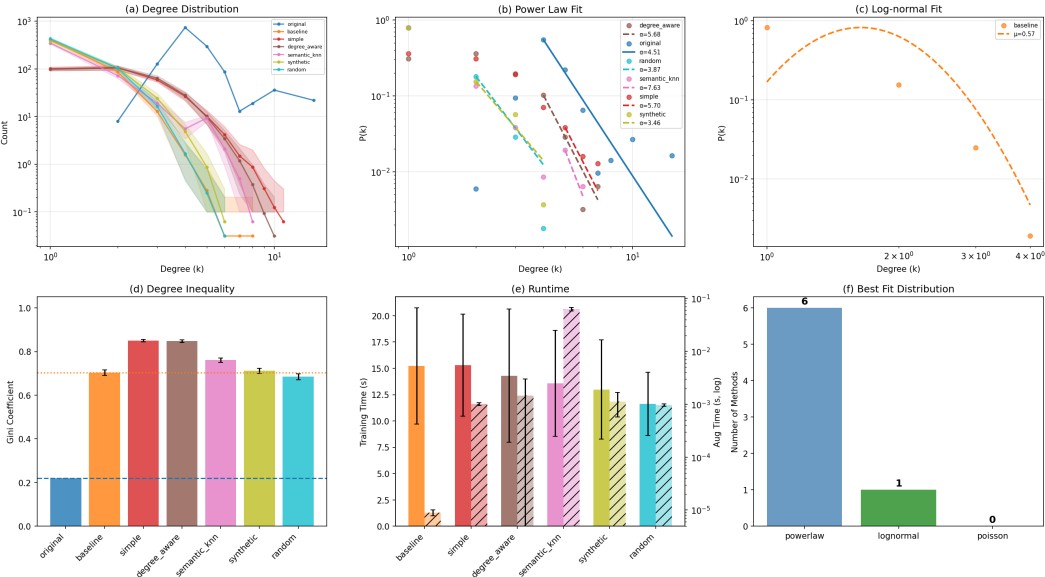

Figure 16: Amazon (product–category), GraphSAGE, $q$=0.10, $\phi$=2: Comprehensive analysis ($M \pm$ SD, $n = 32$ seeds) comparing baseline, augmentation methods, and original graph. Panel (a) shows degree distributions on log-log scale with confidence bands; (b) Power Law fits with exponent $\alpha$; (c) Log-normal fits with parameters $\mu$ and $\sigma$; (d) Gini coefficients quantifying degree inequality (lower = more uniform); (e) runtime comparison showing training time (left axis) and augmentation time (right axis, log scale); (f) best-fit distribution counts across methods.

### C.3.4    GCN

**Summary Analysis**    For Amazon GCN, `semantic_knn` provides the largest AUC lifts at $q$=0.05 ($\phi$=2: +0.091; $\phi$=5: +0.087), while `random` and `simple` often lag (e.g., `simple` $-0.083$ at

$q$=0.01, $\phi$=2). Brier improves most with degree_aware at $q$=0.05, $\phi$=5 ($-0.031$) and with simple/random/synthetic at $q$=0.10, $\phi$=5 ($-0.023/-0.036/-0.032$); semantic_knn typically worsens Brier despite AUC gains. Degree distributions move from extreme sparsity (mean degree 0.043, Gini 0.960) to moderately denser graphs (up to $\sim 0.59$ at $q$=0.05, $\phi$=5 and $\sim 1.19$ at $q$=0.10, $\phi$=5), with random/synthetic sharply lowering Gini and isolated nodes. Runtime overheads remain negligible: augmentation $< 0.063$s and training within a few seconds of baseline, whereas the dense original costs $\sim 219$s, so accuracy changes are attributable to graph quality rather than compute.

**AUC and Brier Score**

Table 18: Amazon (product–category) GCN: AUC-ROC ($M \pm SD$) with paired $t$-tests vs. sparse baseline ($n = 32$ seeds). A higher AUC is better.

| $q$ | $\phi$ | Method | AUC $M \pm SD$ | $\Delta$AUC | $t(31)$ | $p$ | $d$ |
|---|---|---|---|---|---|---|---|
| 0.01 | 5× | baseline | $0.478 \pm 0.196$ | +0.000 | — | — | — |
| 0.01 | 5× | degree_aware | $0.445 \pm 0.181$ | -0.033$^{\text{ns}}$ | 0.81 | 0.425 | +0.14 |
| 0.01 | 5× | simple | $0.446 \pm 0.177$ | -0.032$^{\text{ns}}$ | 0.73 | 0.468 | +0.13 |
| 0.01 | 5× | **semantic_knn** | $0.481 \pm 0.183$ | +0.003$^{\text{ns}}$ | $-0.07$ | 0.946 | $-0.01$ |
| 0.01 | 5× | synthetic | $0.418 \pm 0.158$ | -0.060$^{\text{ns}}$ | 1.35 | 0.188 | +0.24 |
| 0.01 | 5× | random | $0.399 \pm 0.179$ | -0.079$^{\text{ns}}$ | 1.78 | 0.084 | +0.32 |
| 0.01 | 5× | original | $0.907 \pm 0.015$ | +0.430$^{***}$ | $-12.34$ | <0.001 | $-2.18$ |
| 0.01 | 2× | **baseline** | $0.478 \pm 0.196$ | +0.000 | — | — | — |
| 0.01 | 2× | degree_aware | $0.465 \pm 0.199$ | -0.012$^{\text{ns}}$ | 0.34 | 0.736 | +0.06 |
| 0.01 | 2× | simple | $0.394 \pm 0.174$ | -0.083$^{*}$ | 2.09 | 0.045 | +0.37 |
| 0.01 | 2× | semantic_knn | $0.453 \pm 0.184$ | -0.024$^{\text{ns}}$ | 0.53 | 0.601 | +0.09 |
| 0.01 | 2× | synthetic | $0.429 \pm 0.193$ | -0.049$^{\text{ns}}$ | 1.22 | 0.230 | +0.22 |
| 0.01 | 2× | random | $0.454 \pm 0.193$ | -0.024$^{\text{ns}}$ | 0.55 | 0.583 | +0.10 |
| 0.01 | 2× | original | $0.907 \pm 0.015$ | +0.430$^{***}$ | $-12.34$ | <0.001 | $-2.18$ |
| 0.05 | 5× | baseline | $0.500 \pm 0.102$ | +0.000 | — | — | — |
| 0.05 | 5× | degree_aware | $0.598 \pm 0.116$ | +0.098$^{***}$ | $-3.72$ | <0.001 | $-0.66$ |
| 0.05 | 5× | **simple** | $0.611 \pm 0.119$ | +0.111$^{***}$ | $-4.79$ | <0.001 | $-0.85$ |
| 0.05 | 5× | semantic_knn | $0.587 \pm 0.102$ | +0.087$^{***}$ | $-4.65$ | <0.001 | $-0.82$ |
| 0.05 | 5× | synthetic | $0.567 \pm 0.100$ | +0.066$^{**}$ | $-3.10$ | 0.004 | $-0.55$ |
| 0.05 | 5× | random | $0.476 \pm 0.087$ | -0.024$^{\text{ns}}$ | 1.18 | 0.247 | +0.21 |
| 0.05 | 5× | original | $0.907 \pm 0.015$ | +0.407$^{***}$ | $-21.62$ | <0.001 | $-3.82$ |
| 0.05 | 2× | baseline | $0.500 \pm 0.102$ | +0.000 | — | — | — |
| 0.05 | 2× | degree_aware | $0.511 \pm 0.128$ | +0.011$^{\text{ns}}$ | $-0.49$ | 0.629 | $-0.09$ |
| 0.05 | 2× | simple | $0.508 \pm 0.126$ | +0.008$^{\text{ns}}$ | $-0.43$ | 0.670 | $-0.08$ |
| 0.05 | 2× | **semantic_knn** | $0.592 \pm 0.111$ | +0.091$^{***}$ | $-5.15$ | <0.001 | $-0.91$ |
| 0.05 | 2× | synthetic | $0.481 \pm 0.107$ | -0.019$^{\text{ns}}$ | 1.07 | 0.294 | +0.19 |
| 0.05 | 2× | random | $0.480 \pm 0.100$ | -0.020$^{\text{ns}}$ | 1.10 | 0.279 | +0.19 |
| 0.05 | 2× | original | $0.907 \pm 0.015$ | +0.407$^{***}$ | $-21.62$ | <0.001 | $-3.82$ |
| 0.10 | 5× | baseline | $0.604 \pm 0.078$ | +0.000 | — | — | — |
| 0.10 | 5× | **degree_aware** | $0.648 \pm 0.080$ | +0.044$^{**}$ | $-3.10$ | 0.004 | $-0.55$ |
| 0.10 | 5× | simple | $0.616 \pm 0.083$ | +0.013$^{\text{ns}}$ | $-0.66$ | 0.513 | $-0.12$ |
| 0.10 | 5× | semantic_knn | $0.602 \pm 0.072$ | -0.001$^{\text{ns}}$ | 0.09 | 0.927 | +0.02 |
| 0.10 | 5× | synthetic | $0.640 \pm 0.092$ | +0.036$^{\text{ns}}$ | $-1.87$ | 0.071 | $-0.33$ |
| 0.10 | 5× | random | $0.617 \pm 0.082$ | +0.014$^{\text{ns}}$ | $-0.79$ | 0.438 | $-0.14$ |
| 0.10 | 5× | original | $0.907 \pm 0.015$ | +0.304$^{***}$ | $-21.01$ | <0.001 | $-3.71$ |
| 0.10 | 2× | **baseline** | $0.604 \pm 0.078$ | +0.000 | — | — | — |
| 0.10 | 2× | degree_aware | $0.554 \pm 0.081$ | -0.050$^{**}$ | 3.31 | 0.002 | +0.58 |
| 0.10 | 2× | simple | $0.594 \pm 0.089$ | -0.009$^{\text{ns}}$ | 0.61 | 0.544 | +0.11 |
| 0.10 | 2× | semantic_knn | $0.602 \pm 0.072$ | -0.001$^{\text{ns}}$ | 0.09 | 0.927 | +0.02 |
| 0.10 | 2× | synthetic | $0.578 \pm 0.058$ | -0.026$^{\text{ns}}$ | 1.80 | 0.081 | +0.32 |
| 0.10 | 2× | random | $0.577 \pm 0.073$ | -0.027$^{\text{ns}}$ | 1.75 | 0.091 | +0.31 |

| $q$ | $\phi$ | Method | AUC $M \pm SD$ | $\Delta$AUC | $t(31)$ | $p$ | $d$ |
|---|---|---|---|---|---|---|---|
| 0.10 | 2× | original | $0.907 \pm 0.015$ | $+0.304^{***}$ | $-21.01$ | $<0.001$ | $-3.71$ |

Table 19: Amazon (product–category) GCN: Brier Score ($M \pm SD$) with paired $t$-tests vs. sparse baseline ($n = 32$ seeds, lower is better).

| $q$ | $\phi$ | Method | Brier $M \pm SD$ | $\Delta$Brier | $t(31)$ | $p$ | $d$ |
|---|---|---|---|---|---|---|---|
| 0.01 | 5× | baseline | $0.361 \pm 0.084$ | $+0.000$ | — | — | — |
| 0.01 | 5× | **degree_aware** | $0.340 \pm 0.081$ | $-0.022^{ns}$ | $1.56$ | $0.130$ | $+0.28$ |
| 0.01 | 5× | simple | $0.353 \pm 0.071$ | $-0.008^{ns}$ | $0.70$ | $0.492$ | $+0.13$ |
| 0.01 | 5× | semantic_knn | $0.371 \pm 0.064$ | $+0.010^{ns}$ | $-0.30$ | $0.767$ | $-0.05$ |
| 0.01 | 5× | synthetic | $0.372 \pm 0.052$ | $+0.011^{ns}$ | $-0.38$ | $0.704$ | $-0.07$ |
| 0.01 | 5× | random | $0.378 \pm 0.050$ | $+0.017^{ns}$ | $-0.83$ | $0.411$ | $-0.15$ |
| 0.01 | 5× | original | $0.121 \pm 0.018$ | $-0.240^{***}$ | $15.58$ | $<0.001$ | $+2.75$ |
| 0.01 | 2× | baseline | $0.361 \pm 0.084$ | $+0.000$ | — | — | — |
| 0.01 | 2× | **degree_aware** | $0.351 \pm 0.081$ | $-0.010^{ns}$ | $1.17$ | $0.253$ | $+0.21$ |
| 0.01 | 2× | simple | $0.360 \pm 0.061$ | $-0.001^{ns}$ | $0.38$ | $0.709$ | $+0.07$ |
| 0.01 | 2× | semantic_knn | $0.369 \pm 0.062$ | $+0.008^{ns}$ | $-0.22$ | $0.826$ | $-0.04$ |
| 0.01 | 2× | synthetic | $0.391 \pm 0.045$ | $+0.030^{ns}$ | $-1.61$ | $0.117$ | $-0.29$ |
| 0.01 | 2× | random | $0.363 \pm 0.061$ | $+0.002^{ns}$ | $0.16$ | $0.875$ | $+0.03$ |
| 0.01 | 2× | original | $0.121 \pm 0.018$ | $-0.240^{***}$ | $15.58$ | $<0.001$ | $+2.75$ |
| 0.05 | 5× | baseline | $0.326 \pm 0.054$ | $+0.000$ | — | — | — |
| 0.05 | 5× | **degree_aware** | $0.295 \pm 0.042$ | $-0.031^{**}$ | $2.85$ | $0.008$ | $+0.50$ |
| 0.05 | 5× | simple | $0.302 \pm 0.045$ | $-0.024^{*}$ | $2.19$ | $0.036$ | $+0.39$ |
| 0.05 | 5× | semantic_knn | $0.333 \pm 0.053$ | $+0.007^{ns}$ | $-0.60$ | $0.554$ | $-0.11$ |
| 0.05 | 5× | synthetic | $0.339 \pm 0.045$ | $+0.012^{ns}$ | $-0.97$ | $0.342$ | $-0.17$ |
| 0.05 | 5× | random | $0.324 \pm 0.049$ | $-0.003^{ns}$ | $0.28$ | $0.780$ | $+0.05$ |
| 0.05 | 5× | original | $0.121 \pm 0.018$ | $-0.205^{***}$ | $18.98$ | $<0.001$ | $+3.36$ |
| 0.05 | 2× | baseline | $0.326 \pm 0.054$ | $+0.000$ | — | — | — |
| 0.05 | 2× | **degree_aware** | $0.311 \pm 0.048$ | $-0.015^{ns}$ | $1.23$ | $0.230$ | $+0.22$ |
| 0.05 | 2× | simple | $0.321 \pm 0.050$ | $-0.005^{ns}$ | $0.51$ | $0.615$ | $+0.09$ |
| 0.05 | 2× | semantic_knn | $0.336 \pm 0.055$ | $+0.009^{ns}$ | $-0.75$ | $0.459$ | $-0.13$ |
| 0.05 | 2× | synthetic | $0.323 \pm 0.045$ | $-0.003^{ns}$ | $0.32$ | $0.755$ | $+0.06$ |
| 0.05 | 2× | random | $0.332 \pm 0.044$ | $+0.006^{ns}$ | $-0.54$ | $0.593$ | $-0.10$ |
| 0.05 | 2× | original | $0.121 \pm 0.018$ | $-0.205^{***}$ | $18.98$ | $<0.001$ | $+3.36$ |
| 0.10 | 5× | baseline | $0.291 \pm 0.053$ | $+0.000$ | — | — | — |
| 0.10 | 5× | degree_aware | $0.273 \pm 0.030$ | $-0.019^{ns}$ | $1.68$ | $0.103$ | $+0.30$ |
| 0.10 | 5× | simple | $0.268 \pm 0.023$ | $-0.023^{*}$ | $2.42$ | $0.022$ | $+0.43$ |
| 0.10 | 5× | semantic_knn | $0.307 \pm 0.050$ | $+0.015^{ns}$ | $-1.31$ | $0.199$ | $-0.23$ |
| 0.10 | 5× | synthetic | $0.260 \pm 0.046$ | $-0.032^{**}$ | $2.96$ | $0.006$ | $+0.52$ |
| 0.10 | 5× | **random** | $0.255 \pm 0.035$ | $-0.036^{***}$ | $4.28$ | $<0.001$ | $+0.76$ |
| 0.10 | 5× | original | $0.121 \pm 0.018$ | $-0.170^{***}$ | $16.88$ | $<0.001$ | $+2.98$ |
| 0.10 | 2× | baseline | $0.291 \pm 0.053$ | $+0.000$ | — | — | — |
| 0.10 | 2× | degree_aware | $0.286 \pm 0.028$ | $-0.006^{ns}$ | $0.60$ | $0.556$ | $+0.11$ |
| 0.10 | 2× | **simple** | $0.277 \pm 0.029$ | $-0.015^{*}$ | $2.23$ | $0.033$ | $+0.39$ |
| 0.10 | 2× | semantic_knn | $0.307 \pm 0.050$ | $+0.015^{ns}$ | $-1.31$ | $0.199$ | $-0.23$ |
| 0.10 | 2× | synthetic | $0.289 \pm 0.045$ | $-0.002^{ns}$ | $0.28$ | $0.785$ | $+0.05$ |
| 0.10 | 2× | random | $0.292 \pm 0.051$ | $+0.001^{ns}$ | $-0.08$ | $0.938$ | $-0.01$ |
| 0.10 | 2× | original | $0.121 \pm 0.018$ | $-0.170^{***}$ | $16.88$ | $<0.001$ | $+2.98$ |

**Degree Distribution Analysis**

Table 20: Amazon (product–category) GCN: Degree Distribution Statistics ($M \pm SD$, $n = 32$ seeds). Lower Gini coefficient indicates more uniform degree distribution.

| $q$ | $\phi$ | Method | Mean Degree | Gini Coeff. | Num. Isolated | Best Fit |
|---|---|---|---|---|---|---|
| 0.01 | 5× | baseline | $0.0426 \pm 0.0054$ | $0.960 \pm 0.005$ | $1404.4 \pm 7.7$ | lognormal |
| 0.01 | 5× | degree_aware | $0.1208 \pm 0.0147$ | $0.982 \pm 0.002$ | $1430.3 \pm 4.4$ | powerlaw |
| 0.01 | 5× | simple | $0.1208 \pm 0.0147$ | $0.982 \pm 0.002$ | $1430.3 \pm 4.4$ | powerlaw |
| 0.01 | 5× | semantic_knn | $0.1060 \pm 0.0096$ | $0.963 \pm 0.005$ | $1381.6 \pm 10.1$ | powerlaw |
| 0.01 | 5× | synthetic | $0.1208 \pm 0.0147$ | $0.925 \pm 0.010$ | $1329.9 \pm 17.3$ | powerlaw |
| 0.01 | 5× | **random** | $0.1208 \pm 0.0147$ | $0.893 \pm 0.012$ | $1298.8 \pm 19.1$ | lognormal |
| 0.01 | 5× | original | $4.3051 \pm 0.0000$ | $0.219 \pm 0.000$ | $114.0 \pm 0.0$ | powerlaw |
| 0.01 | 2× | baseline | $0.0426 \pm 0.0054$ | $0.960 \pm 0.005$ | $1404.4 \pm 7.7$ | lognormal |
| 0.01 | 2× | degree_aware | $0.0483 \pm 0.0059$ | $0.982 \pm 0.002$ | $1430.3 \pm 4.4$ | powerlaw |
| 0.01 | 2× | simple | $0.0483 \pm 0.0059$ | $0.982 \pm 0.002$ | $1430.3 \pm 4.4$ | powerlaw |
| 0.01 | 2× | semantic_knn | $0.0483 \pm 0.0059$ | $0.978 \pm 0.003$ | $1419.5 \pm 6.0$ | powerlaw |
| 0.01 | 2× | synthetic | $0.0483 \pm 0.0059$ | $0.960 \pm 0.005$ | $1401.4 \pm 8.1$ | lognormal |
| 0.01 | 2× | **random** | $0.0483 \pm 0.0059$ | $0.954 \pm 0.006$ | $1396.2 \pm 8.4$ | lognormal |
| 0.01 | 2× | original | $4.3051 \pm 0.0000$ | $0.219 \pm 0.000$ | $114.0 \pm 0.0$ | powerlaw |
| 0.05 | 5× | baseline | $0.2153 \pm 0.0133$ | $0.824 \pm 0.011$ | $1181.3 \pm 16.7$ | lognormal |
| 0.05 | 5× | degree_aware | $0.5930 \pm 0.0365$ | $0.916 \pm 0.005$ | $1301.9 \pm 9.3$ | powerlaw |
| 0.05 | 5× | simple | $0.5930 \pm 0.0365$ | $0.915 \pm 0.005$ | $1301.9 \pm 9.3$ | powerlaw |
| 0.05 | 5× | semantic_knn | $0.2682 \pm 0.0140$ | $0.858 \pm 0.009$ | $1188.1 \pm 16.1$ | powerlaw |
| 0.05 | 5× | synthetic | $0.5930 \pm 0.0365$ | $0.718 \pm 0.016$ | $910.9 \pm 31.0$ | powerlaw |
| 0.05 | 5× | **random** | $0.5930 \pm 0.0365$ | $0.639 \pm 0.018$ | $811.7 \pm 33.7$ | powerlaw |
| 0.05 | 5× | original | $4.3051 \pm 0.0000$ | $0.219 \pm 0.000$ | $114.0 \pm 0.0$ | powerlaw |
| 0.05 | 2× | baseline | $0.2153 \pm 0.0133$ | $0.824 \pm 0.011$ | $1181.3 \pm 16.7$ | lognormal |
| 0.05 | 2× | degree_aware | $0.2372 \pm 0.0146$ | $0.919 \pm 0.005$ | $1301.9 \pm 9.3$ | powerlaw |
| 0.05 | 2× | simple | $0.2372 \pm 0.0146$ | $0.920 \pm 0.004$ | $1301.9 \pm 9.3$ | powerlaw |
| 0.05 | 2× | semantic_knn | $0.2372 \pm 0.0146$ | $0.876 \pm 0.010$ | $1221.0 \pm 17.0$ | powerlaw |
| 0.05 | 2× | synthetic | $0.2372 \pm 0.0146$ | $0.829 \pm 0.011$ | $1174.2 \pm 17.8$ | powerlaw |
| 0.05 | 2× | **random** | $0.2372 \pm 0.0146$ | $0.810 \pm 0.012$ | $1156.3 \pm 18.2$ | lognormal |
| 0.05 | 2× | original | $4.3051 \pm 0.0000$ | $0.219 \pm 0.000$ | $114.0 \pm 0.0$ | powerlaw |
| 0.10 | 5× | baseline | $0.4323 \pm 0.0163$ | $0.703 \pm 0.012$ | $950.6 \pm 19.3$ | lognormal |
| 0.10 | 5× | degree_aware | $1.1873 \pm 0.0446$ | $0.841 \pm 0.006$ | $1154.6 \pm 11.2$ | powerlaw |
| 0.10 | 5× | simple | $1.1873 \pm 0.0446$ | $0.843 \pm 0.006$ | $1154.6 \pm 11.2$ | powerlaw |
| 0.10 | 5× | semantic_knn | $0.4346 \pm 0.0171$ | $0.760 \pm 0.010$ | $1007.2 \pm 17.0$ | powerlaw |
| 0.10 | 5× | synthetic | $1.1873 \pm 0.0446$ | $0.576 \pm 0.015$ | $572.2 \pm 27.0$ | powerlaw |
| 0.10 | 5× | **random** | $1.1873 \pm 0.0446$ | $0.489 \pm 0.013$ | $450.4 \pm 26.1$ | powerlaw |
| 0.10 | 5× | original | $4.3051 \pm 0.0000$ | $0.219 \pm 0.000$ | $114.0 \pm 0.0$ | powerlaw |
| 0.10 | 2× | baseline | $0.4323 \pm 0.0163$ | $0.703 \pm 0.012$ | $950.6 \pm 19.3$ | lognormal |
| 0.10 | 2× | degree_aware | $0.4749 \pm 0.0179$ | $0.848 \pm 0.006$ | $1154.6 \pm 11.2$ | powerlaw |
| 0.10 | 2× | simple | $0.4749 \pm 0.0179$ | $0.850 \pm 0.006$ | $1154.6 \pm 11.2$ | powerlaw |
| 0.10 | 2× | semantic_knn | $0.4346 \pm 0.0171$ | $0.760 \pm 0.010$ | $1007.2 \pm 17.0$ | powerlaw |
| 0.10 | 2× | synthetic | $0.4749 \pm 0.0179$ | $0.712 \pm 0.012$ | $941.3 \pm 17.4$ | powerlaw |
| 0.10 | 2× | **random** | $0.4749 \pm 0.0179$ | $0.685 \pm 0.014$ | $911.9 \pm 21.6$ | powerlaw |
| 0.10 | 2× | original | $4.3051 \pm 0.0000$ | $0.219 \pm 0.000$ | $114.0 \pm 0.0$ | powerlaw |

**Runtime Analysis**

Table 21: Amazon (product–category) GCN: Runtime Statistics ($M \pm SD$, seconds, $n = 32$ seeds). Lower times are better.

| $q$ | $\phi$ | Method | Aug. Time (s) | Train Time (s) |
|---|---|---|---|---|
| 0.01 | 5× | baseline | 0.0000 ± 0.0000 | 3.72 ± 1.07 |
| 0.01 | 5× | degree_aware | 0.0012 ± 0.0001 | 3.36 ± 1.13 |
| 0.01 | 5× | **simple** | 0.0010 ± 0.0001 | 3.39 ± 1.17 |
| 0.01 | 5× | semantic_knn | 0.0108 ± 0.0012 | 3.89 ± 1.70 |
| 0.01 | 5× | synthetic | 0.0010 ± 0.0001 | 3.88 ± 1.61 |
| 0.01 | 5× | random | 0.0010 ± 0.0001 | 3.73 ± 1.25 |
| 0.01 | 5× | original | 0.0000 ± 0.0000 | 219.23 ± 37.81 |
| 0.01 | 2× | baseline | 0.0000 ± 0.0000 | 3.72 ± 1.07 |
| 0.01 | 2× | degree_aware | 0.0012 ± 0.0001 | 3.73 ± 1.78 |
| 0.01 | 2× | simple | 0.0010 ± 0.0001 | 3.45 ± 1.68 |
| 0.01 | 2× | semantic_knn | 0.0049 ± 0.0009 | 3.64 ± 1.51 |
| 0.01 | 2× | synthetic | 0.0010 ± 0.0001 | 3.79 ± 1.47 |
| 0.01 | 2× | **random** | 0.0009 ± 0.0001 | 3.33 ± 1.08 |
| 0.01 | 2× | original | 0.0000 ± 0.0000 | 220.21 ± 38.20 |
| 0.05 | 5× | baseline | 0.0000 ± 0.0000 | 7.02 ± 2.10 |
| 0.05 | 5× | degree_aware | 0.0014 ± 0.0009 | 6.50 ± 1.43 |
| 0.05 | 5× | **simple** | 0.0010 ± 0.0000 | 7.05 ± 2.29 |
| 0.05 | 5× | semantic_knn | 0.0340 ± 0.0028 | 9.95 ± 3.79 |
| 0.05 | 5× | synthetic | 0.0012 ± 0.0006 | 10.06 ± 3.76 |
| 0.05 | 5× | random | 0.0010 ± 0.0000 | 8.04 ± 3.33 |
| 0.05 | 5× | original | 0.0000 ± 0.0000 | 218.97 ± 37.68 |
| 0.05 | 2× | baseline | 0.0000 ± 0.0000 | 7.06 ± 2.10 |
| 0.05 | 2× | degree_aware | 0.0012 ± 0.0005 | 6.55 ± 1.70 |
| 0.05 | 2× | **simple** | 0.0010 ± 0.0000 | 6.92 ± 3.42 |
| 0.05 | 2× | semantic_knn | 0.0217 ± 0.0073 | 8.86 ± 3.56 |
| 0.05 | 2× | synthetic | 0.0011 ± 0.0004 | 7.50 ± 2.72 |
| 0.05 | 2× | random | 0.0010 ± 0.0000 | 8.40 ± 2.97 |
| 0.05 | 2× | original | 0.0000 ± 0.0000 | 219.24 ± 37.79 |
| 0.10 | 5× | baseline | 0.0000 ± 0.0000 | 14.04 ± 7.23 |
| 0.10 | 5× | degree_aware | 0.0013 ± 0.0001 | 14.82 ± 4.01 |
| 0.10 | 5× | **simple** | 0.0010 ± 0.0001 | 14.02 ± 5.16 |
| 0.10 | 5× | semantic_knn | 0.0624 ± 0.0028 | 16.59 ± 7.88 |
| 0.10 | 5× | synthetic | 0.0011 ± 0.0001 | 14.54 ± 3.93 |
| 0.10 | 5× | random | 0.0010 ± 0.0001 | 13.90 ± 5.33 |
| 0.10 | 5× | original | 0.0000 ± 0.0000 | 219.50 ± 37.87 |
| 0.10 | 2× | baseline | 0.0000 ± 0.0000 | 13.91 ± 7.15 |
| 0.10 | 2× | degree_aware | 0.0012 ± 0.0001 | 11.99 ± 4.25 |
| 0.10 | 2× | simple | 0.0011 ± 0.0001 | 11.94 ± 5.13 |
| 0.10 | 2× | semantic_knn | 0.0615 ± 0.0026 | 16.35 ± 7.63 |
| 0.10 | 2× | synthetic | 0.0011 ± 0.0001 | 12.52 ± 3.77 |
| 0.10 | 2× | **random** | 0.0010 ± 0.0001 | 13.85 ± 6.61 |
| 0.10 | 2× | original | 0.0000 ± 0.0000 | 218.81 ± 38.02 |

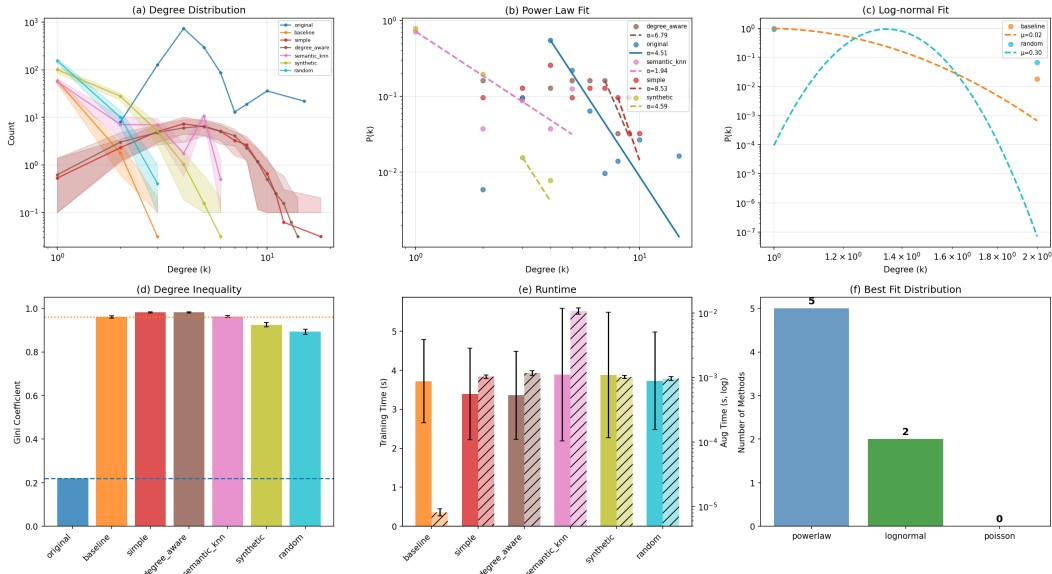

Figure 17: Amazon (product–category), GCN, $q$=0.01, $\phi$=5: Comprehensive analysis ($M \pm$ SD, $n = 32$ seeds) comparing baseline, augmentation methods, and original graph. Panel (a) shows degree distributions on log-log scale with confidence bands; (b) Power Law fits with exponent $\alpha$; (c) Log-normal fits with parameters $\mu$ and $\sigma$; (d) Gini coefficients quantifying degree inequality (lower = more uniform); (e) runtime comparison showing training time (left axis) and augmentation time (right axis, log scale); (f) best-fit distribution counts across methods.

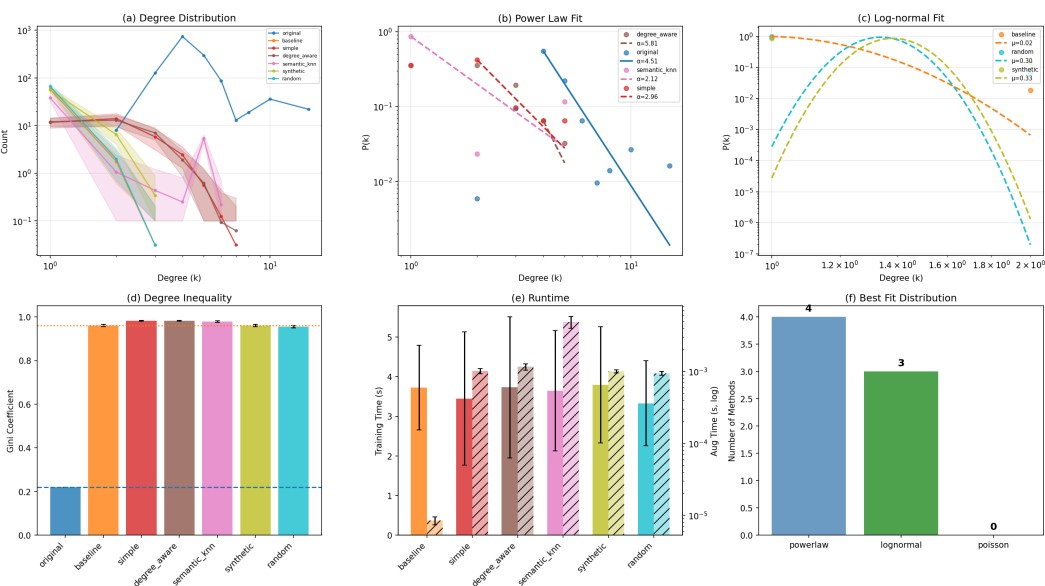

Figure 18: Amazon (product–category), GCN, $q$=0.01, $\phi$=2: Comprehensive analysis ($M \pm$ SD, $n = 32$ seeds) comparing baseline, augmentation methods, and original graph. Panel (a) shows degree distributions on log-log scale with confidence bands; (b) Power Law fits with exponent $\alpha$; (c) Log-normal fits with parameters $\mu$ and $\sigma$; (d) Gini coefficients quantifying degree inequality (lower = more uniform); (e) runtime comparison showing training time (left axis) and augmentation time (right axis, log scale); (f) best-fit distribution counts across methods.

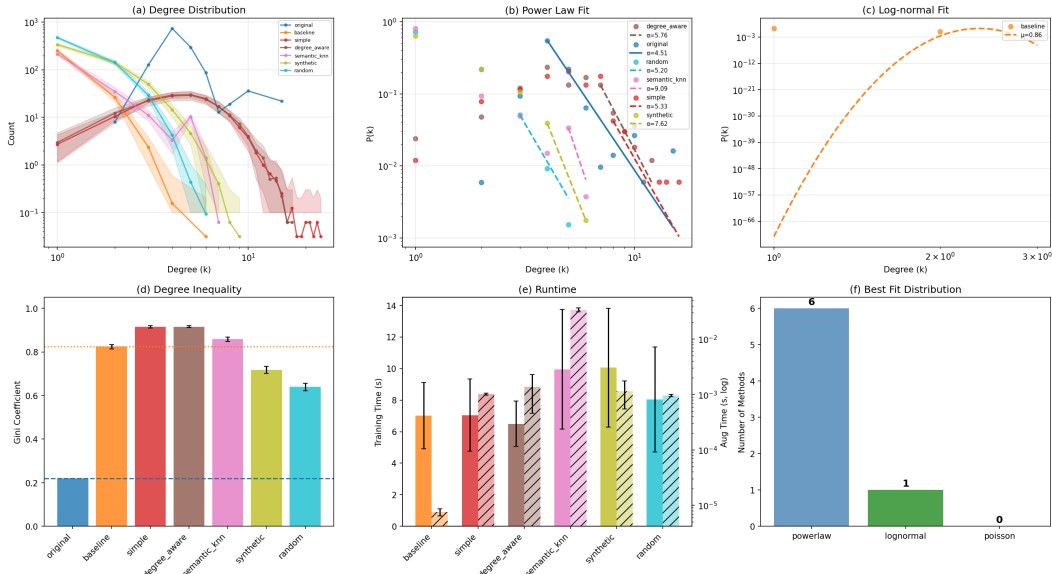

Figure 19: Amazon (product–category), GCN, $q$=0.05, $\phi$=5: Comprehensive analysis ($M \pm$ SD, $n = 32$ seeds) comparing baseline, augmentation methods, and original graph. Panel (a) shows degree distributions on log-log scale with confidence bands; (b) Power Law fits with exponent $\alpha$; (c) Log-normal fits with parameters $\mu$ and $\sigma$; (d) Gini coefficients quantifying degree inequality (lower = more uniform); (e) runtime comparison showing training time (left axis) and augmentation time (right axis, log scale); (f) best-fit distribution counts across methods.

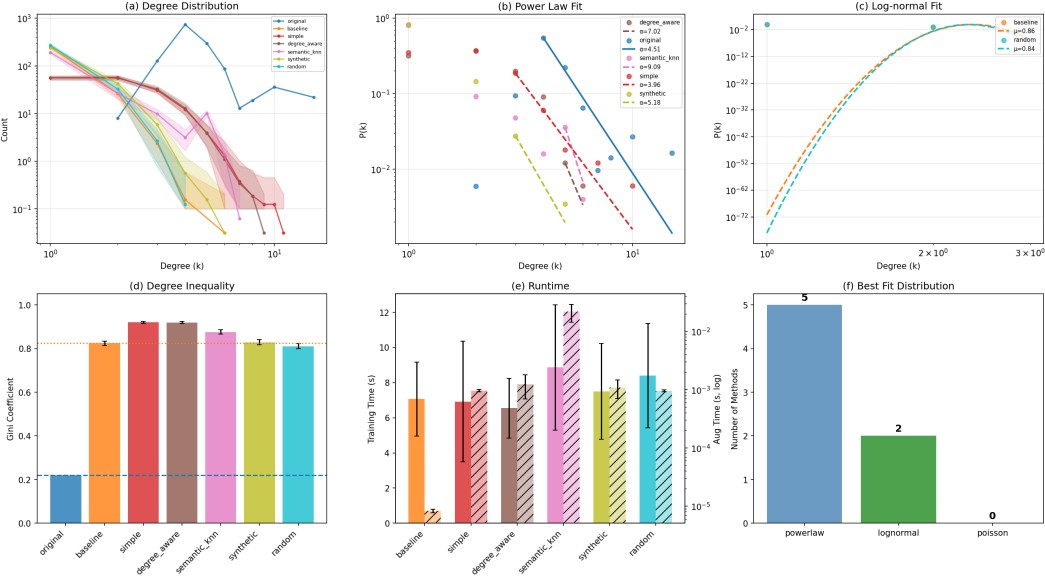

Figure 20: Amazon (product–category), GCN, $q$=0.05, $\phi$=2: Comprehensive analysis ($M \pm$ SD, $n = 32$ seeds) comparing baseline, augmentation methods, and original graph. Panel (a) shows degree distributions on log-log scale with confidence bands; (b) Power Law fits with exponent $\alpha$; (c) Log-normal fits with parameters $\mu$ and $\sigma$; (d) Gini coefficients quantifying degree inequality (lower = more uniform); (e) runtime comparison showing training time (left axis) and augmentation time (right axis, log scale); (f) best-fit distribution counts across methods.

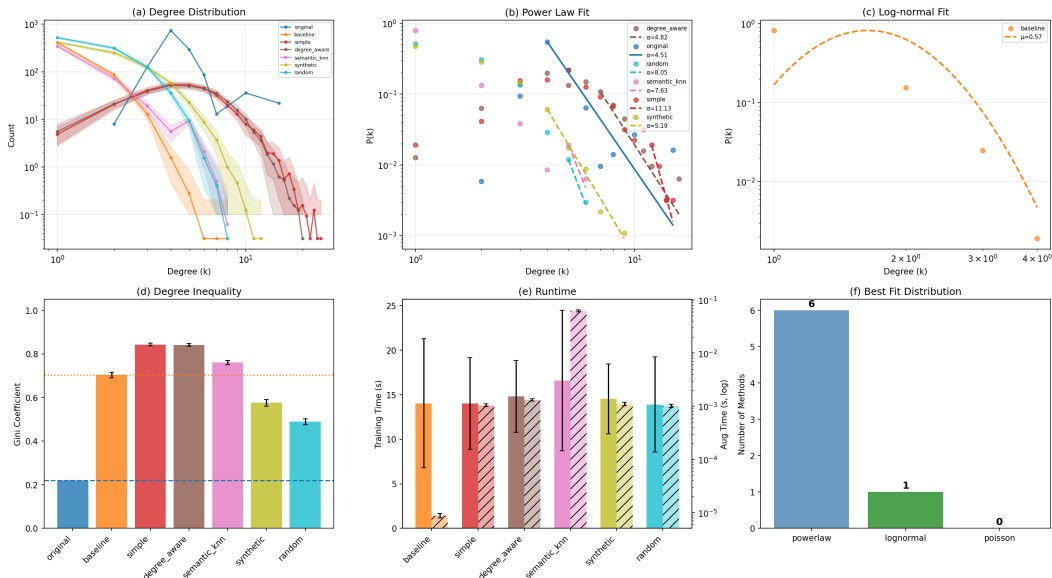

Figure 21: Amazon (product–category), GCN, $q$=0.10, $\phi$=5: Comprehensive analysis ($M \pm$ SD, $n = 32$ seeds) comparing baseline, augmentation methods, and original graph. Panel (a) shows degree distributions on log-log scale with confidence bands; (b) Power Law fits with exponent $\alpha$; (c) Log-normal fits with parameters $\mu$ and $\sigma$; (d) Gini coefficients quantifying degree inequality (lower = more uniform); (e) runtime comparison showing training time (left axis) and augmentation time (right axis, log scale); (f) best-fit distribution counts across methods.

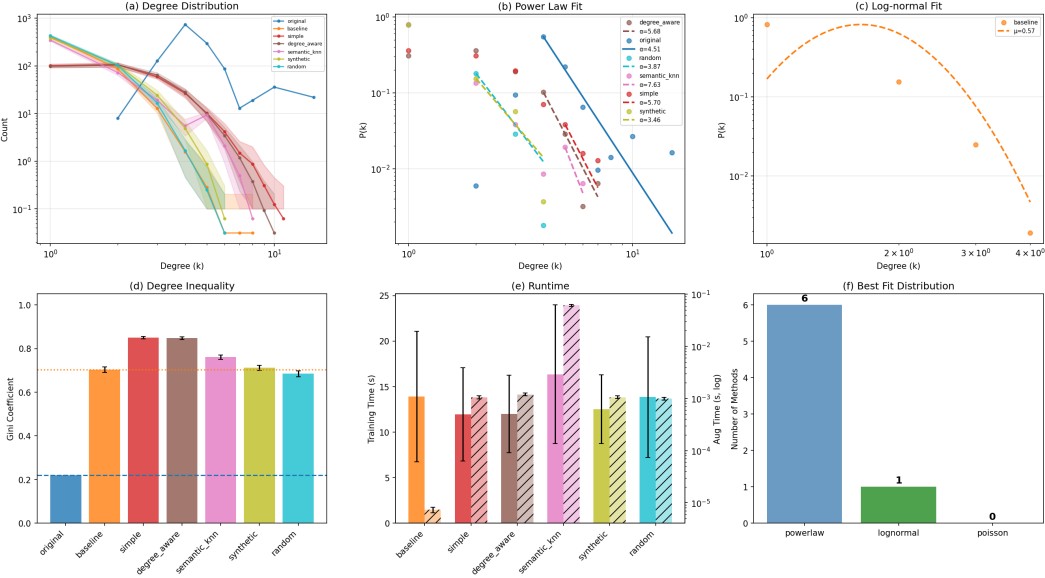

Figure 22: Amazon (product–category), GCN, $q$=0.10, $\phi$=2: Comprehensive analysis ($M \pm$ SD, $n = 32$ seeds) comparing baseline, augmentation methods, and original graph. Panel (a) shows degree distributions on log-log scale with confidence bands; (b) Power Law fits with exponent $\alpha$; (c) Log-normal fits with parameters $\mu$ and $\sigma$; (d) Gini coefficients quantifying degree inequality (lower = more uniform); (e) runtime comparison showing training time (left axis) and augmentation time (right axis, log scale); (f) best-fit distribution counts across methods.

### C.4 BENCHMARK DATASET - MOVIELENS

#### C.4.1 SUMMARY

On MovieLens, gains are modest and depend on encoder and objective: GAT sees only small AUC changes (best at $q$=0.01 with minor lifts from simple/degree_aware; semantic_knn often hurts), while Brier improves slightly for degree_aware/simple at low $q$. GraphSAGE benefits most in AUC from degree_aware at $q$=0.01, and Brier from synthetic at the same setting, reflecting calibration gains without large ranking shifts. GCN shows the clearest AUC gains from simple/degree_aware/semantic_knn at $q$=0.01 (up to +0.075), with the strongest Brier drops from synthetic (down to $-0.082$). Degree distributions remain very sparse for semantic_knn but rise toward mean degree $\sim 1.25$ for other augmentations, lowering isolated nodes; runtime overheads stay negligible (sub-0.16s augmentation, training within a couple seconds of baseline), so quality differences are not driven by cost.

#### C.4.2 GAT

**Summary Analysis** MovieLens GAT shows very small AUC shifts: best cases are minor lifts from simple/degree_aware at $q$=0.01 (e.g., $\phi$=100: +0.007) and stability near baseline elsewhere, while semantic_knn often decreases AUC (notably $q$=0.05) and random stays flat or negative. Brier improvements are modest and concentrated at $q$=0.01, $\phi$=5/2, where simple/degree_aware reduce error by up to $-0.006$; semantic_knn raises Brier at $q$=0.05, and synthetic/random frequently worsen it. Degree-wise, most augmentations keep graphs sparse (mean degree $\approx 0.04$–$0.12$ at $\phi$=5/2) with high Gini, while semantic_knn remains closest to the ultra-sparse baseline. Runtime overhead is negligible (aug $\leq 0.062$s, training within $\sim \pm 2$s of baseline), so differences reflect graph quality rather than cost.

**AUC and Brier Score**

Table 22: MovieLens (movie–genre) GAT: AUC-ROC ($M \pm SD$) with paired $t$-tests vs. sparse baseline ($n = 32$ seeds). A higher AUC is better.

| $q$ | $\phi$ | Method | AUC $M \pm SD$ | $\Delta$AUC | $t(31)$ | $p$ | $d$ |
|---|---|---|---|---|---|---|---|
| 0.01 | 100× | baseline | $0.710 \pm 0.061$ | +0.000 | — | — | — |
| 0.01 | 100× | degree_aware | $0.713 \pm 0.064$ | +0.003$^{\text{ns}}$ | $-0.42$ | 0.681 | $-0.07$ |
| 0.01 | 100× | **simple** | $0.717 \pm 0.063$ | +0.007$^{\text{ns}}$ | $-1.33$ | 0.195 | $-0.23$ |
| 0.01 | 100× | semantic_knn | $0.708 \pm 0.064$ | -0.002$^{\text{ns}}$ | 0.35 | 0.732 | +0.06 |
| 0.01 | 100× | synthetic | $0.679 \pm 0.075$ | -0.031* | 2.15 | 0.039 | +0.38 |
| 0.01 | 100× | random | $0.652 \pm 0.089$ | -0.059*** | 3.66 | <0.001 | +0.65 |
| 0.01 | 100× | original | $0.811 \pm 0.015$ | +0.101*** | $-9.95$ | <0.001 | $-1.76$ |
| 0.01 | 5× | baseline | $0.712 \pm 0.052$ | +0.000 | — | — | — |
| 0.01 | 5× | **degree_aware** | $0.718 \pm 0.060$ | +0.006$^{\text{ns}}$ | $-1.23$ | 0.230 | $-0.22$ |
| 0.01 | 5× | simple | $0.714 \pm 0.050$ | +0.002$^{\text{ns}}$ | $-0.47$ | 0.640 | $-0.08$ |
| 0.01 | 5× | semantic_knn | $0.701 \pm 0.063$ | -0.011$^{\text{ns}}$ | 1.63 | 0.112 | +0.29 |
| 0.01 | 5× | synthetic | $0.712 \pm 0.057$ | -0.000$^{\text{ns}}$ | 0.06 | 0.955 | +0.01 |
| 0.01 | 5× | random | $0.705 \pm 0.066$ | -0.007$^{\text{ns}}$ | 1.04 | 0.308 | +0.18 |
| 0.01 | 5× | original | $0.812 \pm 0.014$ | +0.100*** | $-11.34$ | <0.001 | $-2.01$ |
| 0.01 | 2× | **baseline** | $0.712 \pm 0.052$ | +0.000 | — | — | — |
| 0.01 | 2× | degree_aware | $0.708 \pm 0.058$ | -0.004$^{\text{ns}}$ | 0.99 | 0.328 | +0.18 |
| 0.01 | 2× | simple | $0.711 \pm 0.059$ | -0.001$^{\text{ns}}$ | 0.16 | 0.877 | +0.03 |
| 0.01 | 2× | semantic_knn | $0.701 \pm 0.063$ | -0.011$^{\text{ns}}$ | 1.63 | 0.112 | +0.29 |
| 0.01 | 2× | synthetic | $0.710 \pm 0.077$ | -0.002$^{\text{ns}}$ | 0.21 | 0.834 | +0.04 |
| 0.01 | 2× | random | $0.710 \pm 0.057$ | -0.002$^{\text{ns}}$ | 0.29 | 0.773 | +0.05 |
| 0.01 | 2× | original | $0.812 \pm 0.014$ | +0.100*** | $-11.34$ | <0.001 | $-2.01$ |
| 0.05 | 5× | baseline | $0.719 \pm 0.034$ | +0.000 | — | — | — |
| 0.05 | 5× | degree_aware | $0.717 \pm 0.035$ | -0.002$^{\text{ns}}$ | 0.45 | 0.655 | +0.08 |
| 0.05 | 5× | simple | $0.718 \pm 0.033$ | -0.001$^{\text{ns}}$ | 0.29 | 0.777 | +0.05 |

| $q$ | $\phi$ | Method | AUC $M \pm SD$ | $\Delta$AUC | $t(31)$ | $p$ | $d$ |
|---|---|---|---|---|---|---|---|
| 0.05 | 5× | semantic_knn | $0.701 \pm 0.034$ | -0.018*** | 3.97 | <0.001 | +0.70 |
| 0.05 | 5× | **synthetic** | $0.723 \pm 0.037$ | +0.003$^{ns}$ | −0.53 | 0.600 | −0.09 |
| 0.05 | 5× | random | $0.710 \pm 0.042$ | -0.009$^{ns}$ | 1.54 | 0.133 | +0.27 |
| 0.05 | 5× | original | $0.812 \pm 0.014$ | +0.092*** | −14.43 | <0.001 | −2.55 |
| 0.05 | 2× | **baseline** | $0.719 \pm 0.034$ | +0.000 | — | — | — |
| 0.05 | 2× | degree_aware | $0.716 \pm 0.028$ | -0.004$^{ns}$ | 0.78 | 0.440 | +0.14 |
| 0.05 | 2× | simple | $0.716 \pm 0.030$ | -0.004$^{ns}$ | 0.80 | 0.432 | +0.14 |
| 0.05 | 2× | semantic_knn | $0.701 \pm 0.034$ | -0.018*** | 3.97 | <0.001 | +0.70 |
| 0.05 | 2× | synthetic | $0.706 \pm 0.039$ | -0.013* | 2.31 | 0.027 | +0.41 |
| 0.05 | 2× | random | $0.705 \pm 0.045$ | -0.015$^{ns}$ | 1.87 | 0.071 | +0.33 |
| 0.05 | 2× | original | $0.812 \pm 0.014$ | +0.092*** | −14.43 | <0.001 | −2.55 |
| 0.10 | 5× | baseline | $0.713 \pm 0.029$ | +0.000 | — | — | — |
| 0.10 | 5× | degree_aware | $0.714 \pm 0.027$ | +0.002$^{ns}$ | −0.75 | 0.456 | −0.13 |
| 0.10 | 5× | **simple** | $0.715 \pm 0.025$ | +0.002$^{ns}$ | −0.75 | 0.461 | −0.13 |
| 0.10 | 5× | semantic_knn | $0.706 \pm 0.028$ | -0.006* | 2.44 | 0.021 | +0.43 |
| 0.10 | 5× | synthetic | $0.709 \pm 0.026$ | -0.003$^{ns}$ | 0.79 | 0.434 | +0.14 |
| 0.10 | 5× | random | $0.709 \pm 0.025$ | -0.004$^{ns}$ | 0.84 | 0.407 | +0.15 |
| 0.10 | 5× | original | $0.812 \pm 0.014$ | +0.099*** | −17.98 | <0.001 | −3.18 |

Table 23: MovieLens (movie–genre) GAT: Brier Score ($M \pm SD$) with paired $t$-tests vs. sparse baseline ($n = 32$ seeds, lower is better).

| $q$ | $\phi$ | Method | Brier $M \pm SD$ | $\Delta$Brier | $t(31)$ | $p$ | $d$ |
|---|---|---|---|---|---|---|---|
| 0.01 | 100× | **baseline** | $0.231 \pm 0.016$ | +0.000 | — | — | — |
| 0.01 | 100× | degree_aware | $0.233 \pm 0.014$ | +0.001$^{ns}$ | −0.72 | 0.474 | −0.13 |
| 0.01 | 100× | simple | $0.231 \pm 0.012$ | -0.000$^{ns}$ | 0.12 | 0.907 | +0.02 |
| 0.01 | 100× | semantic_knn | $0.235 \pm 0.014$ | +0.004$^{ns}$ | −1.43 | 0.162 | −0.25 |
| 0.01 | 100× | synthetic | $0.245 \pm 0.008$ | +0.014*** | −4.82 | <0.001 | −0.85 |
| 0.01 | 100× | random | $0.245 \pm 0.009$ | +0.013*** | −3.89 | <0.001 | −0.69 |
| 0.01 | 100× | original | $0.218 \pm 0.004$ | -0.013*** | 5.22 | <0.001 | +0.92 |
| 0.01 | 5× | baseline | $0.235 \pm 0.015$ | +0.000 | — | — | — |
| 0.01 | 5× | degree_aware | $0.230 \pm 0.017$ | -0.005** | 3.01 | 0.005 | +0.53 |
| 0.01 | 5× | **simple** | $0.229 \pm 0.014$ | -0.006** | 3.34 | 0.002 | +0.59 |
| 0.01 | 5× | semantic_knn | $0.236 \pm 0.015$ | +0.001$^{ns}$ | −0.49 | 0.627 | −0.09 |
| 0.01 | 5× | synthetic | $0.238 \pm 0.015$ | +0.003$^{ns}$ | −1.35 | 0.188 | −0.24 |
| 0.01 | 5× | random | $0.236 \pm 0.013$ | +0.001$^{ns}$ | −0.64 | 0.526 | −0.11 |
| 0.01 | 5× | original | $0.218 \pm 0.004$ | -0.017*** | 6.33 | <0.001 | +1.12 |
| 0.01 | 2× | baseline | $0.235 \pm 0.015$ | +0.000 | — | — | — |
| 0.01 | 2× | **degree_aware** | $0.231 \pm 0.016$ | -0.004$^{ns}$ | 1.94 | 0.061 | +0.34 |
| 0.01 | 2× | simple | $0.232 \pm 0.015$ | -0.003$^{ns}$ | 1.87 | 0.071 | +0.33 |
| 0.01 | 2× | semantic_knn | $0.236 \pm 0.015$ | +0.001$^{ns}$ | −0.49 | 0.627 | −0.09 |
| 0.01 | 2× | synthetic | $0.235 \pm 0.013$ | +0.000$^{ns}$ | −0.11 | 0.917 | −0.02 |
| 0.01 | 2× | random | $0.238 \pm 0.010$ | +0.003$^{ns}$ | −1.41 | 0.168 | −0.25 |
| 0.01 | 2× | original | $0.218 \pm 0.004$ | -0.017*** | 6.33 | <0.001 | +1.12 |
| 0.05 | 5× | baseline | $0.240 \pm 0.008$ | +0.000 | — | — | — |
| 0.05 | 5× | degree_aware | $0.239 \pm 0.006$ | -0.001$^{ns}$ | 0.90 | 0.378 | +0.16 |
| 0.05 | 5× | simple | $0.240 \pm 0.009$ | +0.000$^{ns}$ | −0.46 | 0.650 | −0.08 |
| 0.05 | 5× | semantic_knn | $0.245 \pm 0.007$ | +0.005*** | −4.83 | <0.001 | −0.85 |
| 0.05 | 5× | **synthetic** | $0.237 \pm 0.011$ | -0.003$^{ns}$ | 1.76 | 0.088 | +0.31 |
| 0.05 | 5× | random | $0.239 \pm 0.008$ | -0.001$^{ns}$ | 0.55 | 0.589 | +0.10 |
| 0.05 | 5× | original | $0.218 \pm 0.004$ | -0.022*** | 14.99 | <0.001 | +2.65 |
| 0.05 | 2× | **baseline** | $0.240 \pm 0.008$ | +0.000 | — | — | — |

| $q$ | $\phi$ | Method | Brier $M \pm SD$ | $\Delta$Brier | $t(31)$ | $p$ | $d$ |
|---|---|---|---|---|---|---|---|
| 0.05 | 2$\times$ | degree_aware | $0.241 \pm 0.007$ | $+0.001^{\text{ns}}$ | $-1.24$ | 0.224 | $-0.22$ |
| 0.05 | 2$\times$ | simple | $0.242 \pm 0.007$ | $+0.002^{\text{ns}}$ | $-1.95$ | 0.060 | $-0.34$ |
| 0.05 | 2$\times$ | semantic_knn | $0.245 \pm 0.007$ | $+0.005^{***}$ | $-4.83$ | $<0.001$ | $-0.85$ |
| 0.05 | 2$\times$ | synthetic | $0.244 \pm 0.008$ | $+0.004^{*}$ | $-2.55$ | 0.016 | $-0.45$ |
| 0.05 | 2$\times$ | random | $0.244 \pm 0.009$ | $+0.004^{**}$ | $-3.10$ | 0.004 | $-0.55$ |
| 0.05 | 2$\times$ | original | $0.218 \pm 0.004$ | $-0.022^{***}$ | 14.99 | $<0.001$ | $+2.65$ |
| 0.10 | 5$\times$ | baseline | $0.238 \pm 0.007$ | $+0.000$ | — | — | — |
| 0.10 | 5$\times$ | degree_aware | $0.238 \pm 0.005$ | $-0.000^{\text{ns}}$ | 0.51 | 0.613 | $+0.09$ |
| 0.10 | 5$\times$ | simple | $0.238 \pm 0.006$ | $+0.000^{\text{ns}}$ | $-0.20$ | 0.842 | $-0.04$ |
| 0.10 | 5$\times$ | semantic_knn | $0.240 \pm 0.006$ | $+0.002^{*}$ | $-2.36$ | 0.025 | $-0.42$ |
| 0.10 | 5$\times$ | **synthetic** | $0.237 \pm 0.007$ | $-0.001^{\text{ns}}$ | 0.41 | 0.682 | $+0.07$ |
| 0.10 | 5$\times$ | random | $0.238 \pm 0.009$ | $+0.000^{\text{ns}}$ | $-0.17$ | 0.870 | $-0.03$ |
| 0.10 | 5$\times$ | original | $0.218 \pm 0.004$ | $-0.020^{***}$ | 12.93 | $<0.001$ | $+2.29$ |

**Degree Distribution Analysis**

Table 24: MovieLens (movie–genre) GAT: Degree Distribution Statistics ($M \pm SD$, $n = 32$ seeds). Lower Gini coefficient indicates more uniform degree distribution.

| $q$ | $\phi$ | Method | Mean Degree | Gini Coeff. | Num. Isolated | Best Fit |
|---|---|---|---|---|---|---|
| 0.01 | 100$\times$ | baseline | $0.0176 \pm 0.0052$ | $0.983 \pm 0.005$ | $9538.7 \pm 49.7$ | lognormal |
| 0.01 | 100$\times$ | degree_aware | $1.2506 \pm 0.0794$ | $0.989 \pm 0.001$ | $9587.2 \pm 7.8$ | lognormal |
| 0.01 | 100$\times$ | simple | $1.2506 \pm 0.0794$ | $0.988 \pm 0.001$ | $9587.2 \pm 7.8$ | lognormal |
| 0.01 | 100$\times$ | semantic_knn | $0.0190 \pm 0.0009$ | $0.985 \pm 0.001$ | $9546.3 \pm 8.7$ | powerlaw |
| 0.01 | 100$\times$ | synthetic | $1.2506 \pm 0.0794$ | $0.898 \pm 0.007$ | $8452.6 \pm 80.0$ | powerlaw |
| 0.01 | 100$\times$ | **random** | $1.2506 \pm 0.0794$ | $0.476 \pm 0.013$ | $2784.9 \pm 217.0$ | lognormal |
| 0.01 | 100$\times$ | original | $2.2713 \pm 0.0000$ | $0.266 \pm 0.000$ | $0.0 \pm 0.0$ | lognormal |
| 0.01 | 5$\times$ | baseline | $0.0226 \pm 0.0014$ | $0.978 \pm 0.001$ | $9490.3 \pm 13.9$ | lognormal |
| 0.01 | 5$\times$ | degree_aware | $0.0625 \pm 0.0040$ | $0.990 \pm 0.001$ | $9587.2 \pm 7.8$ | powerlaw |
| 0.01 | 5$\times$ | simple | $0.0625 \pm 0.0040$ | $0.990 \pm 0.001$ | $9587.2 \pm 7.8$ | powerlaw |
| 0.01 | 5$\times$ | semantic_knn | $0.0190 \pm 0.0009$ | $0.985 \pm 0.001$ | $9546.3 \pm 8.7$ | powerlaw |
| 0.01 | 5$\times$ | synthetic | $0.0625 \pm 0.0040$ | $0.959 \pm 0.003$ | $9227.0 \pm 31.4$ | powerlaw |
| 0.01 | 5$\times$ | **random** | $0.0625 \pm 0.0040$ | $0.941 \pm 0.004$ | $9120.0 \pm 36.3$ | lognormal |
| 0.01 | 5$\times$ | original | $2.2713 \pm 0.0000$ | $0.266 \pm 0.000$ | $0.0 \pm 0.0$ | lognormal |
| 0.01 | 2$\times$ | baseline | $0.0226 \pm 0.0014$ | $0.978 \pm 0.001$ | $9490.3 \pm 13.9$ | lognormal |
| 0.01 | 2$\times$ | degree_aware | $0.0250 \pm 0.0016$ | $0.991 \pm 0.001$ | $9587.2 \pm 7.8$ | powerlaw |
| 0.01 | 2$\times$ | simple | $0.0250 \pm 0.0016$ | $0.991 \pm 0.001$ | $9587.2 \pm 7.8$ | powerlaw |
| 0.01 | 2$\times$ | semantic_knn | $0.0190 \pm 0.0009$ | $0.985 \pm 0.001$ | $9546.3 \pm 8.7$ | powerlaw |
| 0.01 | 2$\times$ | synthetic | $0.0250 \pm 0.0016$ | $0.978 \pm 0.002$ | $9482.6 \pm 14.9$ | lognormal |
| 0.01 | 2$\times$ | **random** | $0.0250 \pm 0.0016$ | $0.976 \pm 0.002$ | $9468.2 \pm 15.8$ | lognormal |
| 0.01 | 2$\times$ | original | $2.2713 \pm 0.0000$ | $0.266 \pm 0.000$ | $0.0 \pm 0.0$ | lognormal |
| 0.05 | 5$\times$ | baseline | $0.1144 \pm 0.0038$ | $0.895 \pm 0.004$ | $8647.3 \pm 35.3$ | lognormal |
| 0.05 | 5$\times$ | degree_aware | $0.3139 \pm 0.0105$ | $0.953 \pm 0.002$ | $9114.0 \pm 20.2$ | powerlaw |
| 0.05 | 5$\times$ | simple | $0.3139 \pm 0.0105$ | $0.953 \pm 0.002$ | $9114.0 \pm 20.2$ | lognormal |
| 0.05 | 5$\times$ | semantic_knn | $0.0704 \pm 0.0022$ | $0.938 \pm 0.002$ | $9065.5 \pm 21.8$ | powerlaw |
| 0.05 | 5$\times$ | synthetic | $0.3139 \pm 0.0105$ | $0.824 \pm 0.005$ | $7528.5 \pm 66.1$ | powerlaw |
| 0.05 | 5$\times$ | **random** | $0.3139 \pm 0.0105$ | $0.763 \pm 0.007$ | $7090.3 \pm 80.0$ | lognormal |
| 0.05 | 5$\times$ | original | $2.2713 \pm 0.0000$ | $0.266 \pm 0.000$ | $0.0 \pm 0.0$ | lognormal |
| 0.05 | 2$\times$ | baseline | $0.1144 \pm 0.0038$ | $0.895 \pm 0.004$ | $8647.3 \pm 35.3$ | lognormal |
| 0.05 | 2$\times$ | degree_aware | $0.1256 \pm 0.0042$ | $0.955 \pm 0.002$ | $9114.0 \pm 20.2$ | powerlaw |
| 0.05 | 2$\times$ | simple | $0.1256 \pm 0.0042$ | $0.955 \pm 0.002$ | $9114.0 \pm 20.2$ | powerlaw |
| 0.05 | 2$\times$ | semantic_knn | $0.0704 \pm 0.0022$ | $0.938 \pm 0.002$ | $9065.5 \pm 21.8$ | powerlaw |

| $q$ | $\phi$ | Method | Mean Degree | Gini Coeff. | Num. Isolated | Best Fit |
|---|---|---|---|---|---|---|
| 0.05 | 2× | synthetic | $0.1256 \pm 0.0042$ | $0.900 \pm 0.004$ | $8629.1 \pm 36.0$ | lognormal |
| 0.05 | 2× | **random** | $0.1256 \pm 0.0042$ | $0.888 \pm 0.004$ | $8559.1 \pm 36.5$ | lognormal |
| 0.05 | 2× | original | $2.2713 \pm 0.0000$ | $0.266 \pm 0.000$ | $0.0 \pm 0.0$ | lognormal |
| 0.10 | 5× | baseline | $0.2285 \pm 0.0050$ | $0.808 \pm 0.004$ | $7684.8 \pm 39.8$ | lognormal |
| 0.10 | 5× | degree_aware | $0.6263 \pm 0.0136$ | $0.908 \pm 0.002$ | $8550.8 \pm 25.1$ | lognormal |
| 0.10 | 5× | simple | $0.6263 \pm 0.0136$ | $0.909 \pm 0.002$ | $8550.8 \pm 25.1$ | lognormal |
| 0.10 | 5× | semantic_knn | $0.1331 \pm 0.0027$ | $0.884 \pm 0.003$ | $8503.1 \pm 24.7$ | lognormal |
| 0.10 | 5× | synthetic | $0.6263 \pm 0.0136$ | $0.703 \pm 0.006$ | $5837.7 \pm 72.7$ | lognormal |
| 0.10 | 5× | **random** | $0.6263 \pm 0.0136$ | $0.625 \pm 0.006$ | $5181.8 \pm 75.1$ | lognormal |
| 0.10 | 5× | original | $2.2713 \pm 0.0000$ | $0.266 \pm 0.000$ | $0.0 \pm 0.0$ | lognormal |

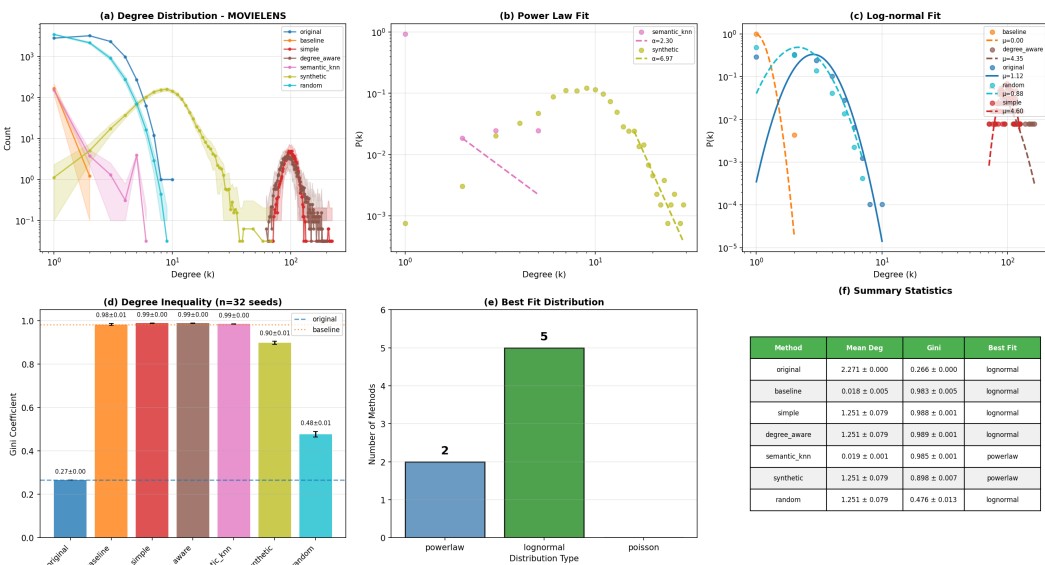

Figure 23: MovieLens (movie–genre), GAT, $q$=0.01, $\phi$=100: Comprehensive analysis ($M \pm$ SD, $n = 32$ seeds) comparing baseline, augmentation methods, and original graph. Panel (a) shows degree distributions on log-log scale with confidence bands; (b) Power Law fits with exponent $\alpha$; (c) Log-normal fits with parameters $\mu$ and $\sigma$; (d) Gini coefficients quantifying degree inequality (lower = more uniform); (e) runtime comparison showing training time (left axis) and augmentation time (right axis, log scale); (f) best-fit distribution counts across methods.

**Runtime Analysis**

Table 25: MovieLens (movie–genre) GAT: Runtime Statistics ($M \pm SD$, seconds, $n = 32$ seeds). Lower times are better.

| $q$ | $\phi$ | Method | Aug. Time (s) | Train Time (s) |
|---|---|---|---|---|
| 0.01 | 5× | baseline | $0.0000 \pm 0.0000$ | $10.75 \pm 3.96$ |
| 0.01 | 5× | degree_aware | $0.0025 \pm 0.0002$ | $10.79 \pm 4.00$ |
| 0.01 | 5× | simple | $0.0023 \pm 0.0002$ | $10.38 \pm 3.83$ |
| 0.01 | 5× | semantic_knn | $0.1278 \pm 0.0128$ | $9.82 \pm 3.43$ |
| 0.01 | 5× | synthetic | $0.0024 \pm 0.0005$ | $11.32 \pm 4.33$ |
| 0.01 | 5× | **random** | $0.0022 \pm 0.0002$ | $12.13 \pm 5.32$ |
| 0.01 | 5× | original | $0.0000 \pm 0.0000$ | $1139.45 \pm 329.59$ |
| 0.01 | 2× | baseline | $0.0000 \pm 0.0000$ | $10.55 \pm 3.97$ |
| 0.01 | 2× | degree_aware | $0.0024 \pm 0.0003$ | $10.17 \pm 3.84$ |

| $q$ | $\phi$ | Method | Aug. Time (s) | Train Time (s) |
|---|---|---|---|---|
| 0.01 | 2× | **simple** | $0.0022 \pm 0.0002$ | $10.48 \pm 3.56$ |
| 0.01 | 2× | semantic_knn | $0.1284 \pm 0.0104$ | $9.64 \pm 3.38$ |
| 0.01 | 2× | synthetic | $0.0023 \pm 0.0002$ | $10.89 \pm 5.18$ |
| 0.01 | 2× | random | $0.0022 \pm 0.0001$ | $11.70 \pm 5.32$ |
| 0.01 | 2× | original | $0.0000 \pm 0.0000$ | $1126.47 \pm 323.34$ |
| 0.05 | 5× | baseline | $0.0000 \pm 0.0000$ | $37.68 \pm 13.98$ |
| 0.05 | 5× | degree_aware | $0.0029 \pm 0.0008$ | $41.96 \pm 14.37$ |
| 0.05 | 5× | **simple** | $0.0023 \pm 0.0003$ | $44.30 \pm 15.61$ |
| 0.05 | 5× | semantic_knn | $0.2163 \pm 0.0124$ | $31.06 \pm 11.65$ |
| 0.05 | 5× | synthetic | $0.0025 \pm 0.0004$ | $42.68 \pm 13.99$ |
| 0.05 | 5× | random | $0.0023 \pm 0.0002$ | $42.32 \pm 12.28$ |
| 0.05 | 5× | original | $0.0000 \pm 0.0000$ | $1109.65 \pm 315.29$ |
| 0.05 | 2× | baseline | $0.0000 \pm 0.0000$ | $38.14 \pm 14.05$ |
| 0.05 | 2× | degree_aware | $0.0025 \pm 0.0002$ | $34.76 \pm 9.98$ |
| 0.05 | 2× | **simple** | $0.0022 \pm 0.0002$ | $37.27 \pm 11.95$ |
| 0.05 | 2× | semantic_knn | $0.2098 \pm 0.0120$ | $31.42 \pm 11.68$ |
| 0.05 | 2× | synthetic | $0.0025 \pm 0.0003$ | $38.47 \pm 11.55$ |
| 0.05 | 2× | random | $0.0023 \pm 0.0004$ | $37.69 \pm 14.78$ |
| 0.05 | 2× | original | $0.0000 \pm 0.0000$ | $1120.95 \pm 318.64$ |
| 0.10 | 5× | baseline | $0.0000 \pm 0.0000$ | $68.52 \pm 28.41$ |
| 0.10 | 5× | degree_aware | $0.0031 \pm 0.0010$ | $89.01 \pm 36.01$ |
| 0.10 | 5× | **simple** | $0.0022 \pm 0.0003$ | $79.85 \pm 26.68$ |
| 0.10 | 5× | semantic_knn | $0.3095 \pm 0.0100$ | $71.80 \pm 25.47$ |
| 0.10 | 5× | synthetic | $0.0025 \pm 0.0004$ | $83.07 \pm 31.44$ |
| 0.10 | 5× | random | $0.0023 \pm 0.0002$ | $82.97 \pm 22.81$ |
| 0.10 | 5× | original | $0.0000 \pm 0.0000$ | $1123.86 \pm 319.62$ |

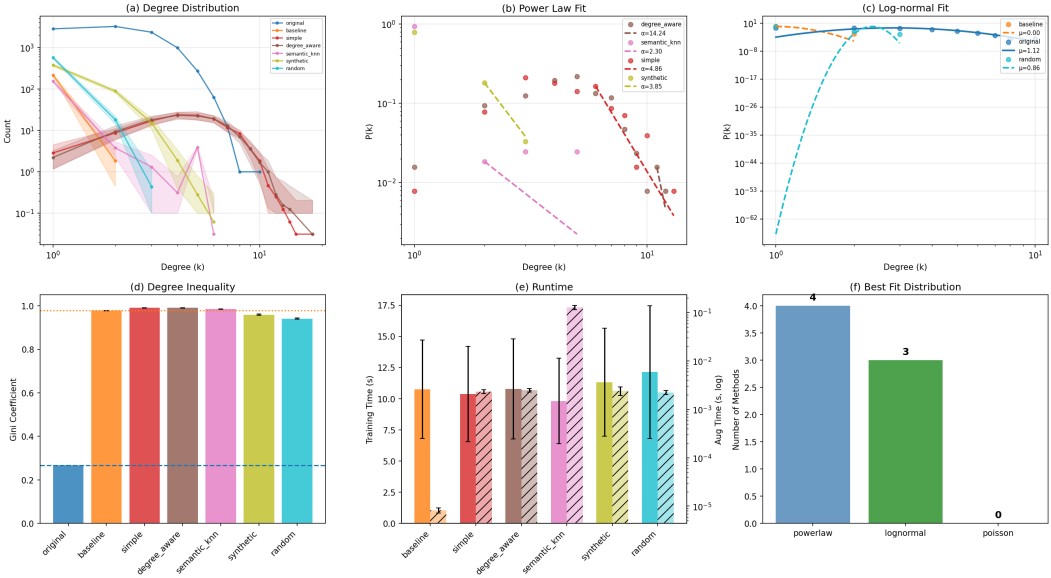

Figure 24: MovieLens (movie–genre), GAT, $q$=0.01, $\phi$=5: Comprehensive analysis ($M \pm$ SD, $n = 32$ seeds) comparing baseline, augmentation methods, and original graph. Panel (a) shows degree distributions on log-log scale with confidence bands; (b) Power Law fits with exponent $\alpha$; (c) Log-normal fits with parameters $\mu$ and $\sigma$; (d) Gini coefficients quantifying degree inequality (lower = more uniform); (e) runtime comparison showing training time (left axis) and augmentation time (right axis, log scale); (f) best-fit distribution counts across methods.

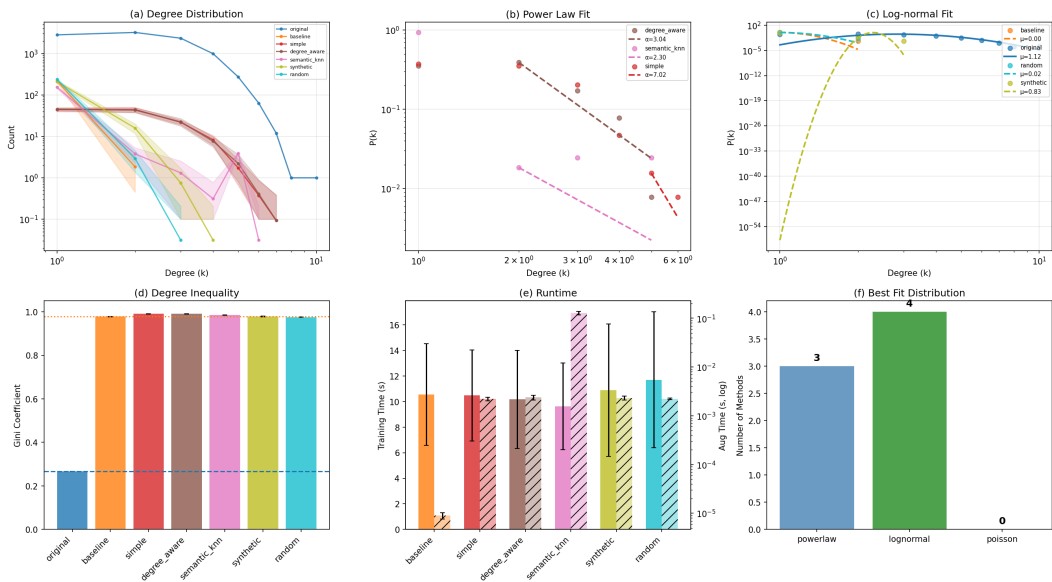

Figure 25: MovieLens (movie–genre), GAT, $q$=0.01, $\phi$=2: Comprehensive analysis ($M \pm \mathrm{SD}$, $n = 32$ seeds) comparing baseline, augmentation methods, and original graph. Panels (a)–(f) follow the same structure as Fig. 24.

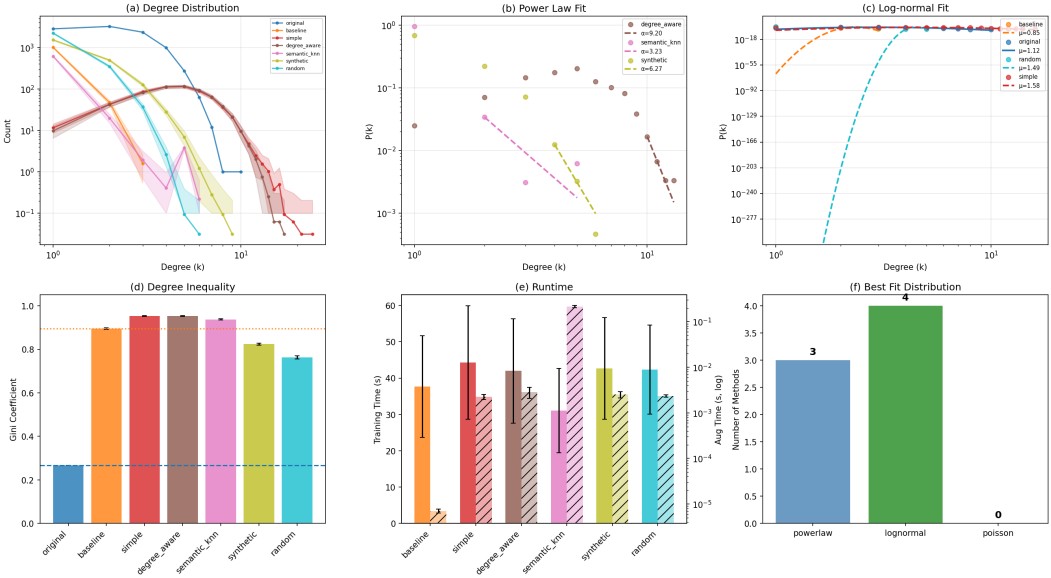

Figure 26: MovieLens (movie–genre), GAT, $q$=0.05, $\phi$=5: Comprehensive analysis ($M \pm \mathrm{SD}$, $n = 32$ seeds) comparing baseline, augmentation methods, and original graph. Panel (a) shows degree distributions on log-log scale with confidence bands; (b) Power Law fits with exponent $\alpha$; (c) Log-normal fits with parameters $\mu$ and $\sigma$; (d) Gini coefficients quantifying degree inequality (lower = more uniform); (e) runtime comparison showing training time (left axis) and augmentation time (right axis, log scale); (f) best-fit distribution counts across methods.

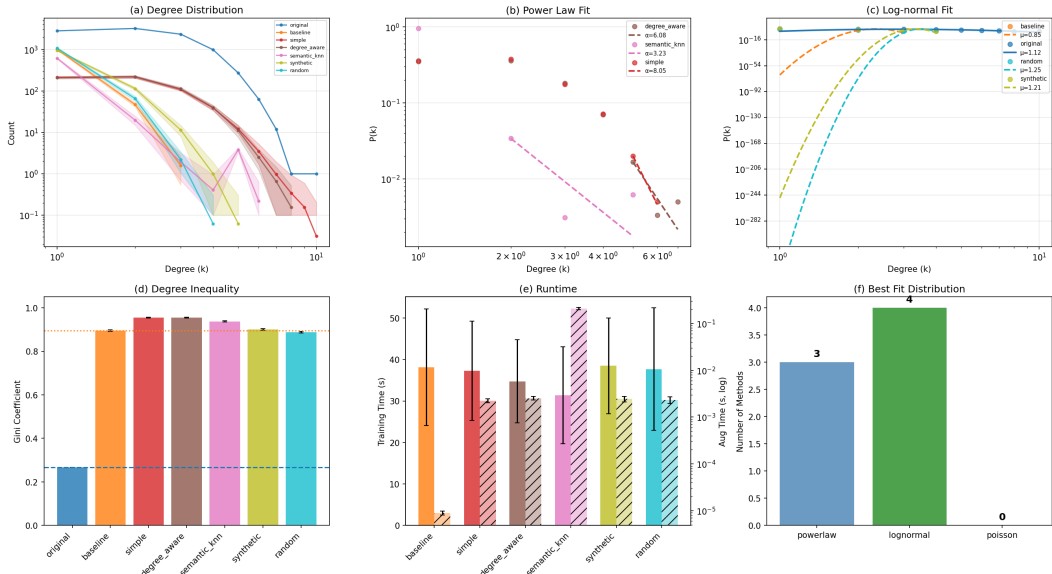

Figure 27: MovieLens (movie–genre), GAT, $q$=0.05, $\phi$=2: Comprehensive analysis ($M \pm \mathrm{SD}$, $n = 32$ seeds) comparing baseline, augmentation methods, and original graph. Panels (a)–(f) follow the same structure as Fig. 26.

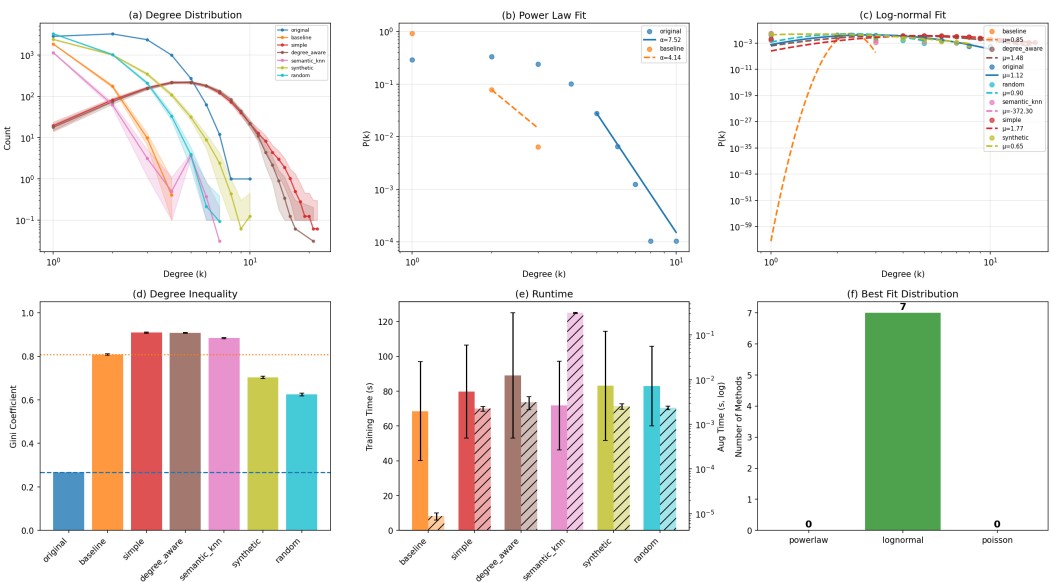

Figure 28: MovieLens (movie–genre), GAT, $q$=0.10, $\phi$=5: Comprehensive analysis ($M \pm \mathrm{SD}$, $n = 32$ seeds) comparing baseline, augmentation methods, and original graph. Panels (a)–(f) follow the standard analysis layout.

### C.4.3    GRAPHSAGE

**Summary Analysis**    MovieLens GraphSAGE sees its main AUC gains from `degree_aware` at $q$=0.01 (up to +0.024), with other methods near baseline and `synthetic` negative. Brier is led by `synthetic` at $q$=0.01, $\phi$=5 ($-0.014$), while `semantic_knn` shows mixed or negligible effects and `random` offers little benefit. Degree statistics mirror the ultra-sparse regime: augmentations at $\phi$=5/2 raise mean degree to $\sim 0.12$ with high Gini; `semantic_knn` keeps graphs closest to baseline, and `random`/`synthetic` reduce isolated counts. Runtime overhead is minimal (aug

$\leq 0.012$s, training within $\sim \pm 1$s of baseline vs. dense-original cost), so performance differences stem from graph quality rather than compute.

**AUC and Brier Score**

Table 26: MovieLens (movie–genre) GraphSAGE: AUC-ROC ($M \pm SD$) with paired $t$-tests vs. sparse baseline ($n = 32$ seeds). A higher AUC is better.

| $q$ | $\phi$ | Method | AUC $M \pm SD$ | $\Delta$AUC | $t(31)$ | $p$ | $d$ |
|---|---|---|---|---|---|---|---|
| 0.01 | 5$\times$ | baseline | $0.637 \pm 0.072$ | +0.000 | — | — | — |
| 0.01 | 5$\times$ | **degree_aware** | $0.653 \pm 0.063$ | +0.015$^{ns}$ | $-1.33$ | 0.192 | $-0.24$ |
| 0.01 | 5$\times$ | simple | $0.642 \pm 0.064$ | +0.005$^{ns}$ | $-0.36$ | 0.724 | $-0.06$ |
| 0.01 | 5$\times$ | semantic_knn | $0.652 \pm 0.054$ | +0.015$^{ns}$ | $-1.55$ | 0.132 | $-0.27$ |
| 0.01 | 5$\times$ | synthetic | $0.608 \pm 0.062$ | -0.030$^{*}$ | 2.41 | 0.022 | +0.43 |
| 0.01 | 5$\times$ | random | $0.620 \pm 0.074$ | -0.017$^{ns}$ | 1.08 | 0.287 | +0.19 |
| 0.01 | 5$\times$ | original | $0.677 \pm 0.016$ | +0.040$^{**}$ | $-3.13$ | 0.004 | $-0.55$ |
| 0.01 | 2$\times$ | baseline | $0.637 \pm 0.072$ | +0.000 | — | — | — |
| 0.01 | 2$\times$ | **degree_aware** | $0.661 \pm 0.056$ | +0.024$^{ns}$ | $-1.63$ | 0.112 | $-0.29$ |
| 0.01 | 2$\times$ | simple | $0.642 \pm 0.070$ | +0.005$^{ns}$ | $-0.34$ | 0.738 | $-0.06$ |
| 0.01 | 2$\times$ | semantic_knn | $0.652 \pm 0.054$ | +0.015$^{ns}$ | $-1.55$ | 0.132 | $-0.27$ |
| 0.01 | 2$\times$ | synthetic | $0.613 \pm 0.066$ | -0.025$^{ns}$ | 1.59 | 0.121 | +0.28 |
| 0.01 | 2$\times$ | random | $0.625 \pm 0.078$ | -0.013$^{ns}$ | 0.87 | 0.390 | +0.15 |
| 0.01 | 2$\times$ | original | $0.677 \pm 0.016$ | +0.040$^{**}$ | $-3.13$ | 0.004 | $-0.55$ |

Table 27: MovieLens (movie–genre) GraphSAGE: Brier Score ($M \pm SD$) with paired $t$-tests vs. sparse baseline ($n = 32$ seeds, lower is better).

| $q$ | $\phi$ | Method | Brier $M \pm SD$ | $\Delta$Brier | $t(31)$ | $p$ | $d$ |
|---|---|---|---|---|---|---|---|
| 0.01 | 5$\times$ | baseline | $0.387 \pm 0.037$ | +0.000 | — | — | — |
| 0.01 | 5$\times$ | degree_aware | $0.374 \pm 0.044$ | -0.013$^{ns}$ | 1.90 | 0.066 | +0.34 |
| 0.01 | 5$\times$ | simple | $0.387 \pm 0.037$ | +0.001$^{ns}$ | $-0.08$ | 0.937 | $-0.01$ |
| 0.01 | 5$\times$ | semantic_knn | $0.389 \pm 0.037$ | +0.003$^{ns}$ | $-0.61$ | 0.548 | $-0.11$ |
| 0.01 | 5$\times$ | **synthetic** | $0.373 \pm 0.047$ | -0.014$^{ns}$ | 1.90 | 0.066 | +0.34 |
| 0.01 | 5$\times$ | random | $0.383 \pm 0.035$ | -0.004$^{ns}$ | 0.51 | 0.612 | +0.09 |
| 0.01 | 5$\times$ | original | $0.273 \pm 0.026$ | -0.114$^{***}$ | 13.32 | <0.001 | +2.36 |
| 0.01 | 2$\times$ | baseline | $0.387 \pm 0.037$ | +0.000 | — | — | — |
| 0.01 | 2$\times$ | degree_aware | $0.391 \pm 0.033$ | +0.005$^{ns}$ | $-1.12$ | 0.272 | $-0.20$ |
| 0.01 | 2$\times$ | simple | $0.388 \pm 0.034$ | +0.001$^{ns}$ | $-0.13$ | 0.899 | $-0.02$ |
| 0.01 | 2$\times$ | semantic_knn | $0.389 \pm 0.037$ | +0.003$^{ns}$ | $-0.61$ | 0.548 | $-0.11$ |
| 0.01 | 2$\times$ | synthetic | $0.377 \pm 0.043$ | -0.010$^{ns}$ | 1.06 | 0.299 | +0.19 |
| 0.01 | 2$\times$ | **random** | $0.368 \pm 0.044$ | -0.019$^{*}$ | 2.17 | 0.038 | +0.38 |
| 0.01 | 2$\times$ | original | $0.273 \pm 0.026$ | -0.114$^{***}$ | 13.32 | <0.001 | +2.36 |

**Degree Distribution Analysis**

Table 28: MovieLens (movie–genre) GraphSAGE: Degree Distribution Statistics ($M \pm SD$, $n = 32$ seeds). Lower Gini coefficient indicates more uniform degree distribution.

| $q$ | $\phi$ | Method | Mean Degree | Gini Coeff. | Num. Isolated | Best Fit |
|---|---|---|---|---|---|---|
| 0.01 | 5$\times$ | baseline | $0.0226 \pm 0.0014$ | $0.978 \pm 0.001$ | $9490.3 \pm 13.9$ | lognormal |
| 0.01 | 5$\times$ | degree_aware | $0.0625 \pm 0.0040$ | $0.990 \pm 0.001$ | $9587.2 \pm 7.8$ | powerlaw |
| 0.01 | 5$\times$ | simple | $0.0625 \pm 0.0040$ | $0.990 \pm 0.001$ | $9587.2 \pm 7.8$ | powerlaw |

| $q$ | $\phi$ | Method | Mean Degree | Gini Coeff. | Num. Isolated | Best Fit |
|---|---|---|---|---|---|---|
| 0.01 | 5× | semantic_knn | $0.0190 \pm 0.0009$ | $0.985 \pm 0.001$ | $9546.3 \pm 8.7$ | powerlaw |
| 0.01 | 5× | synthetic | $0.0625 \pm 0.0040$ | $0.959 \pm 0.003$ | $9227.0 \pm 31.4$ | powerlaw |
| 0.01 | 5× | **random** | $0.0625 \pm 0.0040$ | $0.941 \pm 0.004$ | $9120.0 \pm 36.3$ | lognormal |
| 0.01 | 5× | original | $2.2713 \pm 0.0000$ | $0.266 \pm 0.000$ | $0.0 \pm 0.0$ | lognormal |
| 0.01 | 2× | baseline | $0.0226 \pm 0.0014$ | $0.978 \pm 0.001$ | $9490.3 \pm 13.9$ | lognormal |
| 0.01 | 2× | degree_aware | $0.0250 \pm 0.0016$ | $0.991 \pm 0.001$ | $9587.2 \pm 7.8$ | powerlaw |
| 0.01 | 2× | simple | $0.0250 \pm 0.0016$ | $0.991 \pm 0.001$ | $9587.2 \pm 7.8$ | powerlaw |
| 0.01 | 2× | semantic_knn | $0.0190 \pm 0.0009$ | $0.985 \pm 0.001$ | $9546.3 \pm 8.7$ | powerlaw |
| 0.01 | 2× | synthetic | $0.0250 \pm 0.0016$ | $0.978 \pm 0.002$ | $9482.6 \pm 14.9$ | lognormal |
| 0.01 | 2× | **random** | $0.0250 \pm 0.0016$ | $0.976 \pm 0.002$ | $9468.2 \pm 15.8$ | lognormal |
| 0.01 | 2× | original | $2.2713 \pm 0.0000$ | $0.266 \pm 0.000$ | $0.0 \pm 0.0$ | lognormal |

**Runtime Analysis**

Table 29: MovieLens (movie–genre) GraphSAGE: Runtime Statistics ($M \pm SD$, seconds, $n = 32$ seeds). Lower times are better.

| $q$ | $\phi$ | Method | Aug. Time (s) | Train Time (s) |
|---|---|---|---|---|
| 0.01 | 5× | baseline | $0.0000 \pm 0.0000$ | $7.20 \pm 3.13$ |
| 0.01 | 5× | degree_aware | $0.0027 \pm 0.0010$ | $7.81 \pm 4.17$ |
| 0.01 | 5× | **simple** | $0.0022 \pm 0.0004$ | $7.59 \pm 3.63$ |
| 0.01 | 5× | semantic_knn | $0.1358 \pm 0.0137$ | $8.38 \pm 3.66$ |
| 0.01 | 5× | synthetic | $0.0028 \pm 0.0019$ | $6.96 \pm 3.09$ |
| 0.01 | 5× | random | $0.0023 \pm 0.0005$ | $8.23 \pm 4.69$ |
| 0.01 | 5× | original | $0.0000 \pm 0.0000$ | $355.69 \pm 27.09$ |
| 0.01 | 2× | baseline | $0.0000 \pm 0.0000$ | $7.19 \pm 3.15$ |
| 0.01 | 2× | degree_aware | $0.0024 \pm 0.0005$ | $8.23 \pm 3.96$ |
| 0.01 | 2× | **simple** | $0.0021 \pm 0.0003$ | $7.74 \pm 3.43$ |
| 0.01 | 2× | semantic_knn | $0.1349 \pm 0.0100$ | $8.38 \pm 3.66$ |
| 0.01 | 2× | synthetic | $0.0023 \pm 0.0003$ | $7.41 \pm 2.77$ |
| 0.01 | 2× | random | $0.0022 \pm 0.0005$ | $7.12 \pm 3.06$ |
| 0.01 | 2× | original | $0.0000 \pm 0.0000$ | $355.45 \pm 26.42$ |

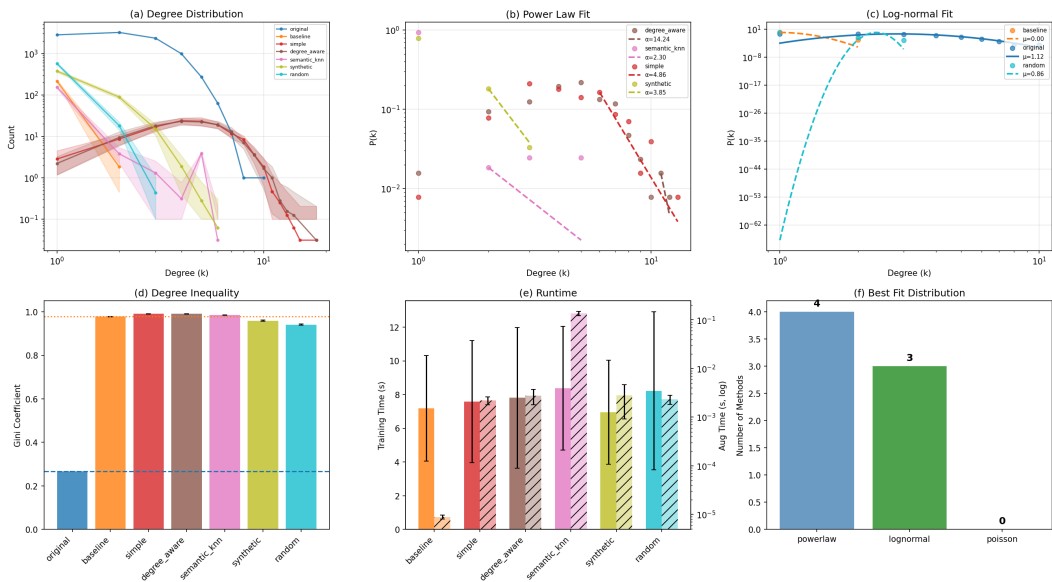

Figure 29: MovieLens (movie–genre), GraphSAGE, $q$=0.01, $\phi$=5: Comprehensive analysis ($M\pm$SD, $n=32$ seeds) comparing baseline, augmentation methods, and original graph. Panel (a) shows degree distributions on log-log scale with confidence bands; (b) Power Law fits with exponent $\alpha$; (c) Log-normal fits with parameters $\mu$ and $\sigma$; (d) Gini coefficients quantifying degree inequality (lower = more uniform); (e) runtime comparison showing training time (left axis) and augmentation time (right axis, log scale); (f) best-fit distribution counts across methods.

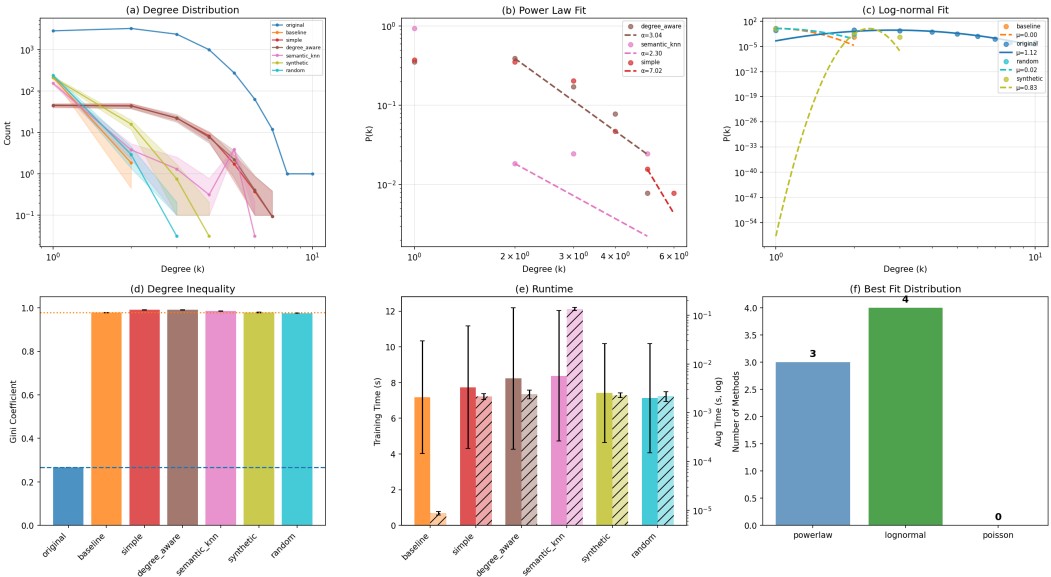

Figure 30: MovieLens (movie–genre), GraphSAGE, $q$=0.01, $\phi$=2: Comprehensive analysis ($M\pm$SD, $n=32$ seeds) comparing baseline, augmentation methods, and original graph. Panel (a) shows degree distributions on log-log scale with confidence bands; (b) Power Law fits with exponent $\alpha$; (c) Log-normal fits with parameters $\mu$ and $\sigma$; (d) Gini coefficients quantifying degree inequality (lower = more uniform); (e) runtime comparison showing training time (left axis) and augmentation time (right axis, log scale); (f) best-fit distribution counts across methods.

### C.4.4   GCN

**Summary Analysis** MovieLens GCN shows the clearest AUC gains among the three augmentation methods: `simple` (and `degree_aware`/`synthetic`) improve ranking at

$q$=0.01 ($\phi$=5:  +0.075;  $\phi$=2:  +0.027), with `semantic_knn` also positive but smaller; `random` trails.    Brier  improves  most  with `synthetic`  at  $q$=0.01,  $\phi$=5  ($-0.082$)  and with `degree_aware`/`simple`/`random`/`synthetic` at $q$=0.10, $\phi$=5 (up to $-0.036$), while `semantic_knn` often raises Brier. Degree-wise, augmentations lift mean degree from the ultra-sparse baseline ($\approx 0.018$) to $\sim 0.12$ at $q$=0.01 and up to $\sim 1.19$ at $q$=0.10, reducing isolated nodes and Gini most for `random`/`synthetic`; `semantic_knn` keeps graphs sparsest. Runtime over-head stays tiny (aug $\leq 0.062$s; training within a few seconds of baseline vs. $\sim 219$s for the dense original), so effectiveness hinges on graph quality rather than cost.

**AUC and Brier Score**

Table 30: MovieLens (movie–genre) GCN: AUC-ROC ($M \pm SD$) with paired $t$-tests vs. sparse baseline ($n = 32$ seeds). A higher AUC is better.

| $q$ | $\phi$ | Method | AUC $M \pm SD$ | $\Delta$AUC | $t(31)$ | $p$ | $d$ |
|---|---|---|---|---|---|---|---|
| 0.01 | 5× | baseline | $0.412 \pm 0.088$ | $+0.000$ | — | — | — |
| 0.01 | 5× | degree_aware | $0.473 \pm 0.089$ | $+0.062^{**}$ | $-3.46$ | 0.002 | $-0.61$ |
| 0.01 | 5× | **simple** | $0.487 \pm 0.099$ | $+0.075^{**}$ | $-3.40$ | 0.002 | $-0.60$ |
| 0.01 | 5× | semantic_knn | $0.455 \pm 0.093$ | $+0.044^{*}$ | $-2.22$ | 0.034 | $-0.39$ |
| 0.01 | 5× | synthetic | $0.479 \pm 0.116$ | $+0.067^{**}$ | $-2.88$ | 0.007 | $-0.51$ |
| 0.01 | 5× | random | $0.447 \pm 0.102$ | $+0.036^{\text{ns}}$ | $-1.62$ | 0.116 | $-0.29$ |
| 0.01 | 5× | original | $0.644 \pm 0.017$ | $+0.232^{***}$ | $-14.58$ | <0.001 | $-2.58$ |
| 0.01 | 2× | baseline | $0.412 \pm 0.088$ | $+0.000$ | — | — | — |
| 0.01 | 2× | degree_aware | $0.434 \pm 0.081$ | $+0.022^{\text{ns}}$ | $-1.20$ | 0.240 | $-0.21$ |
| 0.01 | 2× | simple | $0.439 \pm 0.079$ | $+0.027^{\text{ns}}$ | $-1.37$ | 0.179 | $-0.24$ |
| 0.01 | 2× | **semantic_knn** | $0.455 \pm 0.093$ | $+0.044^{*}$ | $-2.22$ | 0.034 | $-0.39$ |
| 0.01 | 2× | synthetic | $0.435 \pm 0.078$ | $+0.023^{\text{ns}}$ | $-1.17$ | 0.249 | $-0.21$ |
| 0.01 | 2× | random | $0.443 \pm 0.096$ | $+0.032^{\text{ns}}$ | $-1.86$ | 0.072 | $-0.33$ |
| 0.01 | 2× | original | $0.644 \pm 0.017$ | $+0.232^{***}$ | $-14.58$ | <0.001 | $-2.58$ |

Table 31: MovieLens (movie–genre) GCN: Brier Score ($M \pm SD$) with paired $t$-tests vs. sparse baseline ($n = 32$ seeds, lower is better).

| $q$ | $\phi$ | Method | Brier $M \pm SD$ | $\Delta$Brier | $t(31)$ | $p$ | $d$ |
|---|---|---|---|---|---|---|---|
| 0.01 | 5× | baseline | $0.395 \pm 0.022$ | $+0.000$ | — | — | — |
| 0.01 | 5× | degree_aware | $0.372 \pm 0.040$ | $-0.023^{**}$ | 3.63 | 0.001 | $+0.64$ |
| 0.01 | 5× | simple | $0.376 \pm 0.067$ | $-0.019^{\text{ns}}$ | 1.73 | 0.094 | $+0.31$ |
| 0.01 | 5× | semantic_knn | $0.366 \pm 0.041$ | $-0.029^{***}$ | 3.84 | <0.001 | $+0.68$ |
| 0.01 | 5× | **synthetic** | $0.313 \pm 0.051$ | $-0.082^{***}$ | 7.45 | <0.001 | $+1.32$ |
| 0.01 | 5× | random | $0.347 \pm 0.052$ | $-0.047^{***}$ | 4.67 | <0.001 | $+0.83$ |
| 0.01 | 5× | original | $0.240 \pm 0.008$ | $-0.154^{***}$ | 38.96 | <0.001 | $+6.89$ |
| 0.01 | 2× | baseline | $0.395 \pm 0.022$ | $+0.000$ | — | — | — |
| 0.01 | 2× | degree_aware | $0.377 \pm 0.035$ | $-0.017^{*}$ | 2.39 | 0.023 | $+0.42$ |
| 0.01 | 2× | simple | $0.391 \pm 0.036$ | $-0.004^{\text{ns}}$ | 0.50 | 0.618 | $+0.09$ |
| 0.01 | 2× | semantic_knn | $0.366 \pm 0.041$ | $-0.029^{***}$ | 3.84 | <0.001 | $+0.68$ |
| 0.01 | 2× | synthetic | $0.363 \pm 0.035$ | $-0.032^{***}$ | 4.54 | <0.001 | $+0.80$ |
| 0.01 | 2× | **random** | $0.358 \pm 0.037$ | $-0.036^{***}$ | 4.98 | <0.001 | $+0.88$ |
| 0.01 | 2× | original | $0.240 \pm 0.008$ | $-0.154^{***}$ | 38.96 | <0.001 | $+6.89$ |

**Degree Distribution Analysis**

Table 32: MovieLens (movie–genre) GCN: Degree Distribution Statistics ($M \pm SD$, $n = 32$ seeds). Lower Gini coefficient indicates more uniform degree distribution.

| $q$ | $\phi$ | Method | Mean Degree | Gini Coeff. | Num. Isolated | Best Fit |
|---|---|---|---|---|---|---|
| 0.01 | 5× | baseline | 0.0226 ± 0.0014 | 0.978 ± 0.001 | 9490.3 ± 13.9 | lognormal |
| 0.01 | 5× | degree_aware | 0.0625 ± 0.0040 | 0.990 ± 0.001 | 9587.2 ± 7.8 | powerlaw |
| 0.01 | 5× | simple | 0.0625 ± 0.0040 | 0.990 ± 0.001 | 9587.2 ± 7.8 | powerlaw |
| 0.01 | 5× | semantic_knn | 0.0190 ± 0.0009 | 0.985 ± 0.001 | 9546.3 ± 8.7 | powerlaw |
| 0.01 | 5× | synthetic | 0.0625 ± 0.0040 | 0.959 ± 0.003 | 9227.0 ± 31.4 | powerlaw |
| 0.01 | 5× | **random** | 0.0625 ± 0.0040 | 0.941 ± 0.004 | 9120.0 ± 36.3 | lognormal |
| 0.01 | 5× | original | 2.2713 ± 0.0000 | 0.266 ± 0.000 | 0.0 ± 0.0 | lognormal |
| 0.01 | 2× | baseline | 0.0226 ± 0.0014 | 0.978 ± 0.001 | 9490.3 ± 13.9 | lognormal |
| 0.01 | 2× | degree_aware | 0.0250 ± 0.0016 | 0.991 ± 0.001 | 9587.2 ± 7.8 | powerlaw |
| 0.01 | 2× | simple | 0.0250 ± 0.0016 | 0.991 ± 0.001 | 9587.2 ± 7.8 | powerlaw |
| 0.01 | 2× | semantic_knn | 0.0190 ± 0.0009 | 0.985 ± 0.001 | 9546.3 ± 8.7 | powerlaw |
| 0.01 | 2× | synthetic | 0.0250 ± 0.0016 | 0.978 ± 0.002 | 9482.6 ± 14.9 | lognormal |
| 0.01 | 2× | **random** | 0.0250 ± 0.0016 | 0.976 ± 0.002 | 9468.2 ± 15.8 | lognormal |
| 0.01 | 2× | original | 2.2713 ± 0.0000 | 0.266 ± 0.000 | 0.0 ± 0.0 | lognormal |

**Runtime Analysis**

Table 33: MovieLens (movie–genre) GCN: Runtime Statistics ($M \pm SD$, seconds, $n = 32$ seeds). Lower times are better.

| $q$ | $\phi$ | Method | Aug. Time (s) | Train Time (s) |
|---|---|---|---|---|
| 0.01 | 5× | baseline | 0.0000 ± 0.0000 | 8.53 ± 3.69 |
| 0.01 | 5× | degree_aware | 0.0028 ± 0.0004 | 6.95 ± 3.18 |
| 0.01 | 5× | **simple** | 0.0024 ± 0.0002 | 7.38 ± 4.11 |
| 0.01 | 5× | semantic_knn | 0.1441 ± 0.0025 | 7.30 ± 3.77 |
| 0.01 | 5× | synthetic | 0.0026 ± 0.0004 | 5.17 ± 1.42 |
| 0.01 | 5× | random | 0.0026 ± 0.0006 | 6.28 ± 2.98 |
| 0.01 | 5× | original | 0.0000 ± 0.0000 | 349.84 ± 24.68 |
| 0.01 | 2× | baseline | 0.0000 ± 0.0000 | 8.62 ± 3.74 |
| 0.01 | 2× | degree_aware | 0.0025 ± 0.0004 | 7.44 ± 3.23 |
| 0.01 | 2× | **simple** | 0.0023 ± 0.0002 | 7.29 ± 3.16 |
| 0.01 | 2× | semantic_knn | 0.1270 ± 0.0135 | 7.37 ± 3.82 |
| 0.01 | 2× | synthetic | 0.0023 ± 0.0004 | 8.00 ± 4.02 |
| 0.01 | 2× | random | 0.0023 ± 0.0005 | 7.50 ± 3.64 |
| 0.01 | 2× | original | 0.0000 ± 0.0000 | 352.65 ± 24.33 |

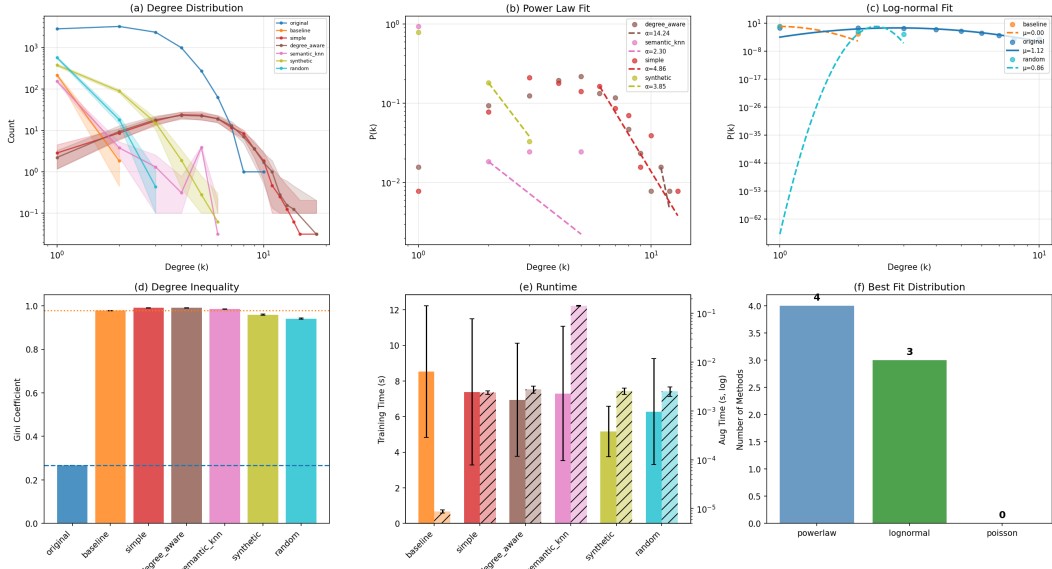

Figure 31: MovieLens (movie–genre), GCN, $q$=0.01, $\phi$=5: Comprehensive analysis ($M \pm$ SD, $n = 32$ seeds) comparing baseline, augmentation methods, and original graph. Panel (a) shows degree distributions on log-log scale with confidence bands; (b) Power Law fits with exponent $\alpha$; (c) Log-normal fits with parameters $\mu$ and $\sigma$; (d) Gini coefficients quantifying degree inequality (lower = more uniform); (e) runtime comparison showing training time (left axis) and augmentation time (right axis, log scale); (f) best-fit distribution counts across methods.

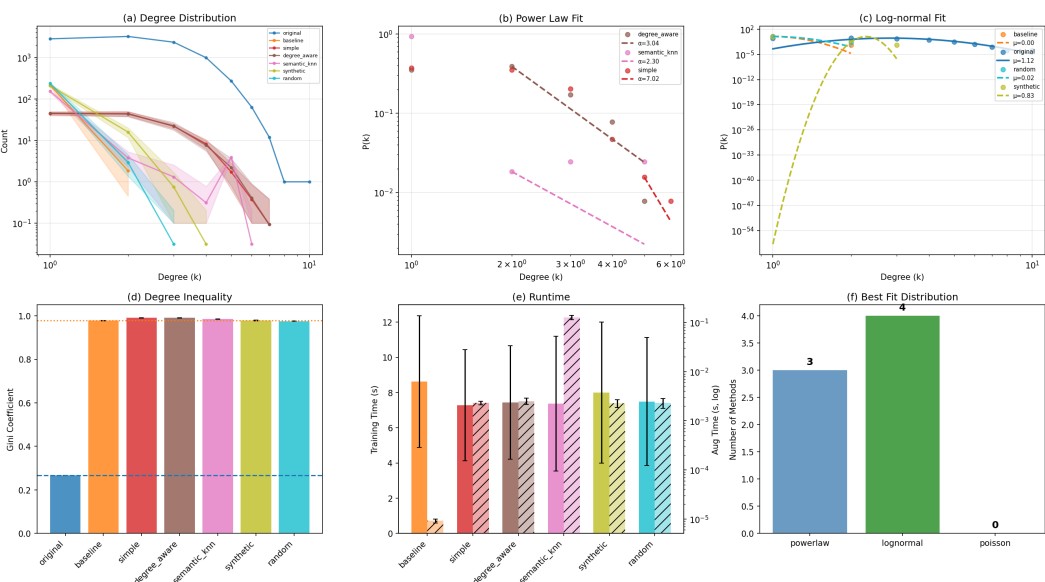

Figure 32: MovieLens (movie–genre), GCN, $q$=0.01, $\phi$=2: Comprehensive analysis ($M \pm$ SD, $n = 32$ seeds) comparing baseline, augmentation methods, and original graph. Panel (a) shows degree distributions on log-log scale with confidence bands; (b) Power Law fits with exponent $\alpha$; (c) Log-normal fits with parameters $\mu$ and $\sigma$; (d) Gini coefficients quantifying degree inequality (lower = more uniform); (e) runtime comparison showing training time (left axis) and augmentation time (right axis, log scale); (f) best-fit distribution counts across methods.

## C.5 DOMAIN CASE STUDY - GDP

### C.5.1 SUMMARY

On GDP, effectiveness depends heavily on encoder and objective. For GAT, `semantic_knn` is the only method with clear AUC gains (at $\phi$=100: +0.014), while `synthetic`/`random` degrade AUC and most methods hurt at lower $\phi$. Brier flips: `semantic_knn` yields the largest reduction at $\phi$=100 ($-0.054$), with `simple`/`degree_aware` giving mild gains at the same setting; lower $\phi$ generally raises Brier. GraphSAGE shows slight AUC upticks only from `simple` at $\phi$=2 (+0.002) and consistent AUC/Brier drops from `semantic_knn`, while `degree_aware` is near baseline. GCN favors neither augmentation strongly: AUC edges up modestly for `simple`/`semantic_knn` at $\phi$=5/2, but Brier often worsens except small gains from `degree_aware` and `simple` at $\phi$=5. Degree-wise, augmentations at $\phi$=100 massively densify graphs (mean degree $\sim$ 189 for several methods), while lower $\phi$ still raise connectivity and cut isolated nodes; `semantic_knn` tends to retain higher Gini but reduces isolation. Runtime overheads stay small (aug $\leq$ 0.10s) though training increases vs. baseline, so accuracy changes reflect graph quality, not augmentation cost.

### C.5.2 GAT

**Summary Analysis** GDP GAT shows limited AUC gains: only `semantic_knn` improves at $\phi$=100 (+0.014), while `simple`/`degree_aware` are near baseline at $\phi$=5 and all methods decline at $\phi$=2. Brier improvements are concentrated at $\phi$=100 with `semantic_knn` leading ($-0.054$) and `simple`/`synthetic`/`random` also lowering error; at lower $\phi$, most methods raise Brier. Degree distributions at $\phi$=100 explode in density for most augmentations (mean degree $\sim$ 189) with reduced inequality for `random`/`synthetic`, while `semantic_knn` keeps graphs closer to the sparse baseline (mean 5.14, lower Gini). Runtime overheads are small for augmentation ($< 0.10$s) though training is higher than baseline; effects are driven by graph changes, not compute.

**AUC and Brier Score**

Table 34: GDP (game–pattern) GAT: AUC-ROC ($M \pm SD$) with paired $t$-tests vs. sparse baseline ($n = 32$ seeds). A higher AUC is better.

| $\phi$ | Method | AUC $M \pm SD$ | $\Delta$AUC | $t(31)$ | $p$ | $d$ |
|---|---|---|---|---|---|---|
| 100$\times$ | baseline | $0.800 \pm 0.022$ | +0.000 | — | — | — |
| 100$\times$ | degree_aware | $0.772 \pm 0.026$ | -0.028*** | 5.29 | <0.001 | +0.93 |
| 100$\times$ | simple | $0.793 \pm 0.023$ | -0.007$^{ns}$ | 1.67 | 0.104 | +0.30 |
| 100$\times$ | **semantic_knn** | $0.814 \pm 0.017$ | +0.014** | $-2.83$ | 0.008 | $-0.50$ |
| 100$\times$ | synthetic | $0.645 \pm 0.061$ | -0.155*** | 13.42 | <0.001 | +2.37 |
| 100$\times$ | random | $0.613 \pm 0.076$ | -0.187*** | 13.14 | <0.001 | +2.32 |
| 5$\times$ | baseline | $0.781 \pm 0.020$ | +0.000 | — | — | — |
| 5$\times$ | degree_aware | $0.782 \pm 0.017$ | +0.001$^{ns}$ | $-0.32$ | 0.754 | $-0.06$ |
| 5$\times$ | **simple** | $0.786 \pm 0.016$ | +0.005$^{ns}$ | $-1.17$ | 0.253 | $-0.21$ |
| 5$\times$ | semantic_knn | $0.772 \pm 0.025$ | -0.009$^{ns}$ | 1.59 | 0.122 | +0.28 |
| 5$\times$ | synthetic | $0.745 \pm 0.023$ | -0.036*** | 6.79 | <0.001 | +1.20 |
| 5$\times$ | random | $0.752 \pm 0.026$ | -0.029*** | 4.42 | <0.001 | +0.78 |
| 2$\times$ | **baseline** | $0.781 \pm 0.020$ | +0.000 | — | — | — |
| 2$\times$ | degree_aware | $0.777 \pm 0.021$ | -0.004$^{ns}$ | 0.91 | 0.368 | +0.16 |
| 2$\times$ | simple | $0.777 \pm 0.020$ | -0.004$^{ns}$ | 0.73 | 0.469 | +0.13 |
| 2$\times$ | semantic_knn | $0.769 \pm 0.021$ | -0.012* | 2.18 | 0.037 | +0.39 |
| 2$\times$ | synthetic | $0.769 \pm 0.027$ | -0.012* | 2.59 | 0.014 | +0.46 |
| 2$\times$ | random | $0.773 \pm 0.025$ | -0.008$^{ns}$ | 1.35 | 0.187 | +0.24 |

Table 35: GDP (game–pattern) GAT: Brier Score ($M \pm SD$) with paired $t$-tests vs. sparse baseline ($n = 32$ seeds, lower is better).

| $\phi$ | Method | Brier $M \pm SD$ | $\Delta$Brier | $t(31)$ | $p$ | $d$ |
|---|---|---|---|---|---|---|
| 100× | baseline | $0.302 \pm 0.040$ | +0.000 | — | — | — |
| 100× | degree_aware | $0.337 \pm 0.079$ | +0.036* | $-2.59$ | 0.015 | $-0.46$ |
| 100× | simple | $0.289 \pm 0.018$ | -0.013* | 2.41 | 0.022 | +0.43 |
| 100× | **semantic_knn** | $0.247 \pm 0.017$ | -0.054*** | 7.06 | <0.001 | +1.25 |
| 100× | synthetic | $0.269 \pm 0.013$ | -0.032*** | 4.76 | <0.001 | +0.84 |
| 100× | random | $0.266 \pm 0.017$ | -0.036*** | 4.92 | <0.001 | +0.87 |
| 5× | baseline | $0.200 \pm 0.010$ | +0.000 | — | — | — |
| 5× | **degree_aware** | $0.197 \pm 0.011$ | -0.003$^{\mathrm{ns}}$ | 1.16 | 0.253 | +0.21 |
| 5× | simple | $0.199 \pm 0.011$ | -0.001$^{\mathrm{ns}}$ | 0.56 | 0.580 | +0.10 |
| 5× | semantic_knn | $0.213 \pm 0.011$ | +0.013*** | $-6.02$ | <0.001 | $-1.06$ |
| 5× | synthetic | $0.288 \pm 0.022$ | +0.088*** | $-20.21$ | <0.001 | $-3.57$ |
| 5× | random | $0.310 \pm 0.027$ | +0.110*** | $-24.36$ | <0.001 | $-4.31$ |
| 2× | **baseline** | $0.200 \pm 0.010$ | +0.000 | — | — | — |
| 2× | degree_aware | $0.202 \pm 0.008$ | +0.002$^{\mathrm{ns}}$ | $-1.35$ | 0.187 | $-0.24$ |
| 2× | simple | $0.200 \pm 0.008$ | +0.000$^{\mathrm{ns}}$ | $-0.00$ | 0.997 | $-0.00$ |
| 2× | semantic_knn | $0.210 \pm 0.009$ | +0.010*** | $-4.51$ | <0.001 | $-0.80$ |
| 2× | synthetic | $0.212 \pm 0.014$ | +0.012*** | $-3.98$ | <0.001 | $-0.70$ |
| 2× | random | $0.219 \pm 0.013$ | +0.019*** | $-6.67$ | <0.001 | $-1.18$ |

**Degree Distribution Analysis**

Table 36: GDP (game–pattern) GAT: Degree Distribution Statistics ($M \pm SD$, $n = 32$ seeds). Lower Gini coefficient indicates more uniform degree distribution.

| $\phi$ | Method | Mean Degree | Gini Coeff. | Num. Isolated | Best Fit |
|---|---|---|---|---|---|
| 100× | baseline | $3.4375 \pm 0.0000$ | $0.540 \pm 0.000$ | $0.0 \pm 0.0$ | powerlaw |
| 100× | degree_aware | $188.9423 \pm 0.0000$ | $0.518 \pm 0.018$ | $53.8 \pm 4.4$ | lognormal |
| 100× | simple | $188.9423 \pm 0.0000$ | $0.614 \pm 0.015$ | $53.8 \pm 4.4$ | powerlaw |
| 100× | semantic_knn | $5.1447 \pm 0.0796$ | $0.331 \pm 0.020$ | $19.2 \pm 4.2$ | powerlaw |
| 100× | synthetic | $188.9423 \pm 0.0000$ | $0.264 \pm 0.018$ | $0.0 \pm 0.0$ | lognormal |
| 100× | **random** | $188.9423 \pm 0.0000$ | $0.042 \pm 0.002$ | $0.0 \pm 0.0$ | lognormal |
| 5× | baseline | $3.4375 \pm 0.0000$ | $0.540 \pm 0.000$ | $0.0 \pm 0.0$ | powerlaw |
| 5× | degree_aware | $17.1875 \pm 0.0000$ | $0.442 \pm 0.008$ | $0.0 \pm 0.0$ | powerlaw |
| 5× | simple | $17.1875 \pm 0.0000$ | $0.559 \pm 0.008$ | $0.0 \pm 0.0$ | lognormal |
| 5× | semantic_knn | $7.1106 \pm 0.0000$ | $0.296 \pm 0.000$ | $0.0 \pm 0.0$ | powerlaw |
| 5× | synthetic | $17.1875 \pm 0.0000$ | $0.296 \pm 0.007$ | $0.0 \pm 0.0$ | lognormal |
| 5× | **random** | $17.1875 \pm 0.0000$ | $0.181 \pm 0.006$ | $0.0 \pm 0.0$ | powerlaw |
| 2× | baseline | $3.4375 \pm 0.0000$ | $0.540 \pm 0.000$ | $0.0 \pm 0.0$ | powerlaw |
| 2× | degree_aware | $6.8750 \pm 0.0000$ | $0.498 \pm 0.009$ | $0.0 \pm 0.0$ | powerlaw |
| 2× | simple | $6.8750 \pm 0.0000$ | $0.567 \pm 0.009$ | $0.0 \pm 0.0$ | powerlaw |
| 2× | **semantic_knn** | $6.8750 \pm 0.0000$ | $0.315 \pm 0.000$ | $0.0 \pm 0.0$ | powerlaw |
| 2× | synthetic | $6.8750 \pm 0.0000$ | $0.384 \pm 0.009$ | $0.0 \pm 0.0$ | powerlaw |
| 2× | random | $6.8750 \pm 0.0000$ | $0.340 \pm 0.008$ | $0.0 \pm 0.0$ | powerlaw |

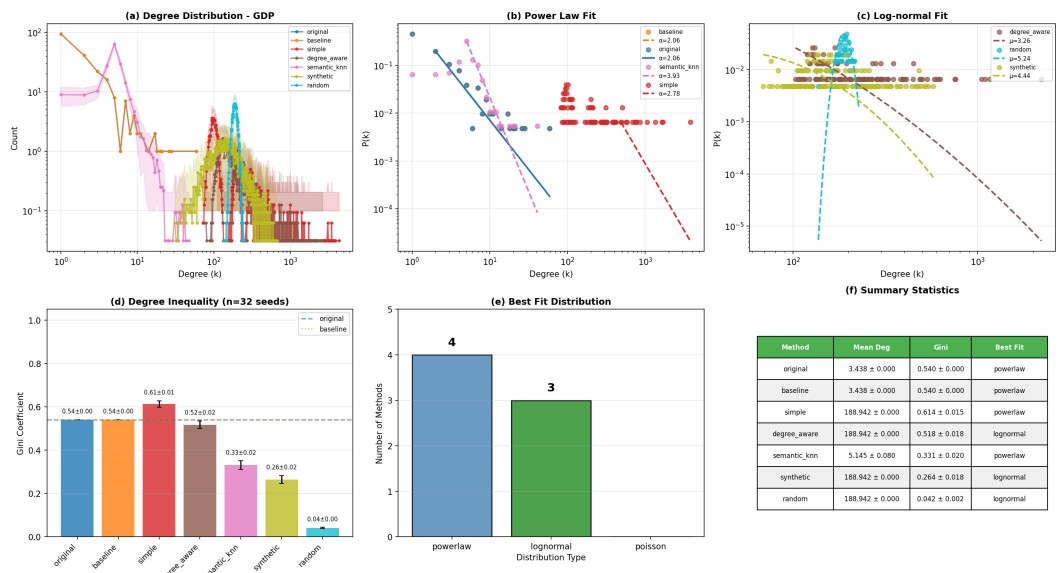

Figure 33: GDP (game–pattern), GAT, $\phi{=}100$: Comprehensive analysis ($M \pm$ SD, $n = 32$ seeds) comparing baseline, augmentation methods. Panel (a) shows degree distributions on log-log scale with confidence bands; (b) Power Law fits with exponent $\alpha$; (c) Log-normal fits with parameters $\mu$ and $\sigma$; (d) Gini coefficients quantifying degree inequality (lower = more uniform); (e) runtime comparison showing training time (left axis) and augmentation time (right axis, log scale); (f) best-fit distribution counts across methods.

**Runtime Analysis**

Table 37: GDP (game–pattern) GAT: Runtime Statistics ($M \pm SD$, seconds, $n = 32$ seeds). Lower times are better.

| $\phi$ | Method | Aug. Time (s) | Train Time (s) |
|---|---|---|---|
| 5× | baseline | $0.0008 \pm 0.0000$ | $70.48 \pm 2.32$ |
| 5× | degree_aware | $0.0029 \pm 0.0077$ | $114.87 \pm 4.60$ |
| 5× | **simple** | $0.0010 \pm 0.0001$ | $109.93 \pm 3.83$ |
| 5× | semantic_knn | $0.0992 \pm 0.0004$ | $95.71 \pm 20.63$ |
| 5× | synthetic | $0.0012 \pm 0.0004$ | $247.25 \pm 85.61$ |
| 5× | random | $0.0010 \pm 0.0000$ | $259.04 \pm 82.96$ |
| 2× | baseline | $0.0008 \pm 0.0000$ | $69.11 \pm 2.07$ |
| 2× | degree_aware | $0.0014 \pm 0.0015$ | $79.00 \pm 1.28$ |
| 2× | **simple** | $0.0009 \pm 0.0000$ | $79.37 \pm 2.71$ |
| 2× | semantic_knn | $0.0795 \pm 0.0020$ | $89.45 \pm 7.87$ |
| 2× | synthetic | $0.0010 \pm 0.0004$ | $100.68 \pm 15.59$ |
| 2× | random | $0.0009 \pm 0.0000$ | $102.15 \pm 24.68$ |

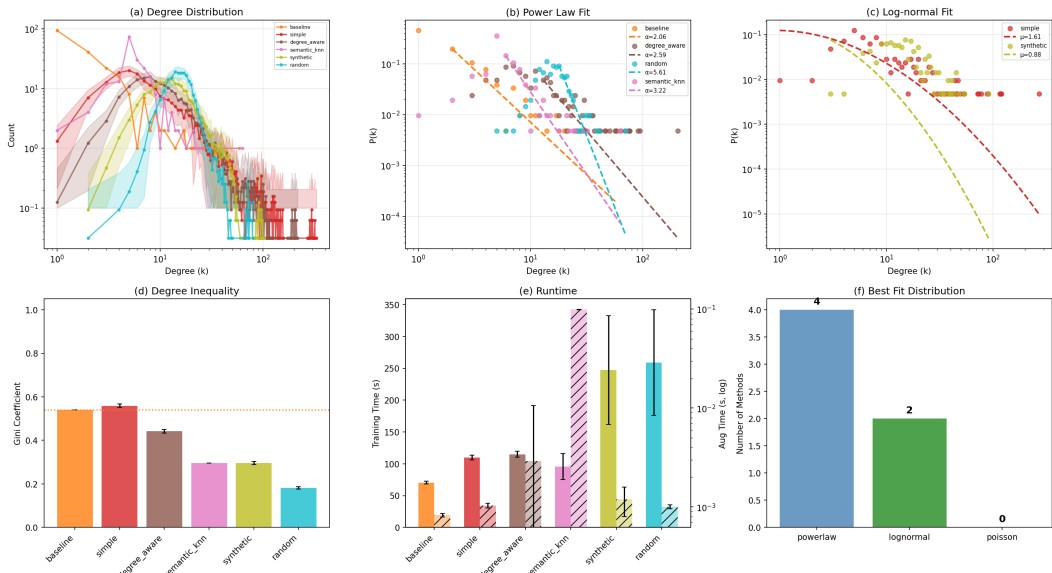

Figure 34: GDP (game–pattern), GAT, $\phi$=5: Comprehensive analysis ($M \pm$ SD, $n = 32$ seeds) comparing baseline, augmentation methods, and original graph. Panel (a) shows degree distributions on log-log scale with confidence bands; (b) Power Law fits with exponent $\alpha$; (c) Log-normal fits with parameters $\mu$ and $\sigma$; (d) Gini coefficients quantifying degree inequality (lower = more uniform); (e) runtime comparison showing training time (left axis) and augmentation time (right axis, log scale); (f) best-fit distribution counts across methods.

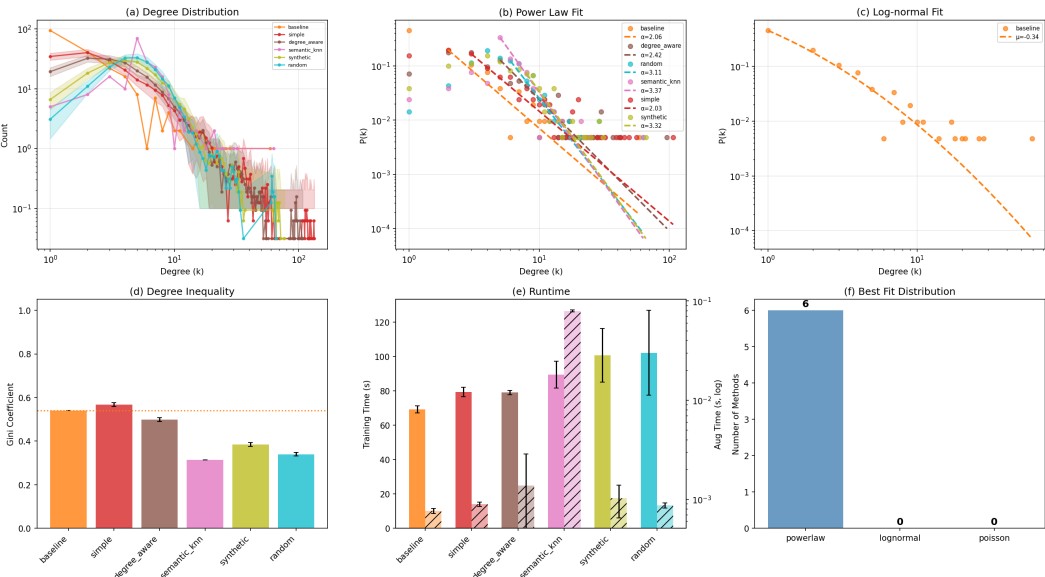

Figure 35: GDP (game–pattern), GAT, $\phi$=2: Comprehensive analysis ($M \pm$ SD, $n = 32$ seeds) comparing baseline, augmentation methods, and original graph. Panel (a) shows degree distributions on log-log scale with confidence bands; (b) Power Law fits with exponent $\alpha$; (c) Log-normal fits with parameters $\mu$ and $\sigma$; (d) Gini coefficients quantifying degree inequality (lower = more uniform); (e) runtime comparison showing training time (left axis) and augmentation time (right axis, log scale); (f) best-fit distribution counts across methods.

### C.5.3 GRAPHSAGE

**Summary Analysis**  GDP GraphSAGE largely tracks baseline: `simple` offers a small AUC lift at $\phi$=2 (+0.002), while `semantic_knn` consistently lowers AUC (down to $-0.040$ at $\phi$=5) and `synthetic`/`random` also degrade. Brier favors `simple`/`degree_aware` with minor reductions, whereas `semantic_knn` and especially `synthetic`/`random` sharply increase error. De-

gree distributions show augmentations densify graphs (mean degree 17 at $\phi=5$, 6.9 at $\phi=2$) with varied inequality: `random`/`synthetic` reduce Gini and isolate counts most, while `semantic_knn` yields lower Gini but modest density. Augmentation costs remain negligible ($< 0.10$s); training grows vs. baseline but differences stem from graph structure changes.

**AUC and Brier Score**

Table 38: GDP (game–pattern) GraphSAGE: AUC-ROC ($M \pm SD$) with paired $t$-tests vs. sparse baseline ($n = 32$ seeds). A higher AUC is better.

| $\phi$ | Method | AUC $M \pm SD$ | $\Delta$AUC | $t(31)$ | $p$ | $d$ |
|---|---|---|---|---|---|---|
| 5× | **baseline** | $0.802 \pm 0.012$ | +0.000 | — | — | — |
| 5× | degree_aware | $0.799 \pm 0.013$ | $-0.003^{\text{ns}}$ | 1.07 | 0.292 | +0.19 |
| 5× | simple | $0.798 \pm 0.013$ | $-0.004^{\text{ns}}$ | 1.90 | 0.067 | +0.34 |
| 5× | semantic_knn | $0.763 \pm 0.016$ | $-0.040^{***}$ | 15.41 | <0.001 | +2.72 |
| 5× | synthetic | $0.780 \pm 0.019$ | $-0.022^{***}$ | 5.37 | <0.001 | +0.95 |
| 5× | random | $0.789 \pm 0.021$ | $-0.013^{***}$ | 3.66 | <0.001 | +0.65 |
| 2× | baseline | $0.802 \pm 0.012$ | +0.000 | — | — | — |
| 2× | degree_aware | $0.801 \pm 0.014$ | $-0.001^{\text{ns}}$ | 0.51 | 0.611 | +0.09 |
| 2× | **simple** | $0.805 \pm 0.015$ | $+0.002^{\text{ns}}$ | $-0.76$ | 0.454 | $-0.13$ |
| 2× | semantic_knn | $0.775 \pm 0.017$ | $-0.028^{***}$ | 7.77 | <0.001 | +1.37 |
| 2× | synthetic | $0.767 \pm 0.017$ | $-0.036^{***}$ | 9.70 | <0.001 | +1.72 |
| 2× | random | $0.764 \pm 0.018$ | $-0.039^{***}$ | 11.53 | <0.001 | +2.04 |

Table 39: GDP (game–pattern) GraphSAGE: Brier Score ($M \pm SD$) with paired $t$-tests vs. sparse baseline ($n = 32$ seeds, lower is better).

| $\phi$ | Method | Brier $M \pm SD$ | $\Delta$Brier | $t(31)$ | $p$ | $d$ |
|---|---|---|---|---|---|---|
| 5× | baseline | $0.175 \pm 0.013$ | +0.000 | — | — | — |
| 5× | degree_aware | $0.173 \pm 0.016$ | $-0.002^{\text{ns}}$ | 0.42 | 0.676 | +0.07 |
| 5× | **simple** | $0.172 \pm 0.012$ | $-0.003^{\text{ns}}$ | 0.88 | 0.383 | +0.16 |
| 5× | semantic_knn | $0.192 \pm 0.013$ | $+0.017^{***}$ | $-5.95$ | <0.001 | $-1.05$ |
| 5× | synthetic | $0.233 \pm 0.012$ | $+0.059^{***}$ | $-18.91$ | <0.001 | $-3.34$ |
| 5× | random | $0.229 \pm 0.012$ | $+0.055^{***}$ | $-17.04$ | <0.001 | $-3.01$ |
| 2× | baseline | $0.175 \pm 0.013$ | +0.000 | — | — | — |
| 2× | degree_aware | $0.170 \pm 0.012$ | $-0.005^{\text{ns}}$ | 1.75 | 0.090 | +0.31 |
| 2× | **simple** | $0.168 \pm 0.011$ | $-0.007^{**}$ | 2.82 | 0.008 | +0.50 |
| 2× | semantic_knn | $0.188 \pm 0.013$ | $+0.013^{***}$ | $-4.25$ | <0.001 | $-0.75$ |
| 2× | synthetic | $0.191 \pm 0.013$ | $+0.016^{***}$ | $-5.52$ | <0.001 | $-0.98$ |
| 2× | random | $0.197 \pm 0.015$ | $+0.022^{***}$ | $-6.13$ | <0.001 | $-1.08$ |

**Degree Distribution Analysis**

Table 40: GDP (game–pattern) GraphSAGE: Degree Distribution Statistics ($M \pm SD$, $n = 32$ seeds). Lower Gini coefficient indicates more uniform degree distribution.

| $q$ | $\phi$ | Method | Mean Degree | Gini Coeff. | Num. Isolated | Best Fit |
|---|---|---|---|---|---|---|
| 5× | baseline | | $3.4375 \pm 0.0000$ | $0.540 \pm 0.000$ | $0.0 \pm 0.0$ | powerlaw |
| 5× | degree_aware | | $17.1875 \pm 0.0000$ | $0.442 \pm 0.008$ | $0.0 \pm 0.0$ | powerlaw |
| 5× | simple | | $17.1875 \pm 0.0000$ | $0.559 \pm 0.008$ | $0.0 \pm 0.0$ | lognormal |
| 5× | semantic_knn | | $7.1106 \pm 0.0000$ | $0.296 \pm 0.000$ | $0.0 \pm 0.0$ | powerlaw |
| 5× | synthetic | | $17.1875 \pm 0.0000$ | $0.296 \pm 0.007$ | $0.0 \pm 0.0$ | lognormal |

| $q$ | $\phi$ | Method | Mean Degree | Gini Coeff. | Num. Isolated | Best Fit |
|---|---|---|---|---|---|---|
| 5× | **random** | | $17.1875 \pm 0.0000$ | $0.181 \pm 0.006$ | $0.0 \pm 0.0$ | powerlaw |
| 2× | baseline | | $3.4375 \pm 0.0000$ | $0.540 \pm 0.000$ | $0.0 \pm 0.0$ | powerlaw |
| 2× | degree_aware | | $6.8750 \pm 0.0000$ | $0.498 \pm 0.009$ | $0.0 \pm 0.0$ | powerlaw |
| 2× | simple | | $6.8750 \pm 0.0000$ | $0.567 \pm 0.009$ | $0.0 \pm 0.0$ | powerlaw |
| 2× | **semantic_knn** | | $6.8750 \pm 0.0000$ | $0.315 \pm 0.000$ | $0.0 \pm 0.0$ | powerlaw |
| 2× | synthetic | | $6.8750 \pm 0.0000$ | $0.384 \pm 0.009$ | $0.0 \pm 0.0$ | powerlaw |
| 2× | random | | $6.8750 \pm 0.0000$ | $0.340 \pm 0.008$ | $0.0 \pm 0.0$ | powerlaw |

### Runtime Analysis

Table 41: GDP (game–pattern) GraphSAGE: Runtime Statistics ($M \pm SD$, seconds, $n = 32$ seeds). Lower times are better.

| $\phi$ | Method | Aug. Time (s) | Train Time (s) |
|---|---|---|---|
| 5× | baseline | $0.0007 \pm 0.0000$ | $50.46 \pm 3.40$ |
| 5× | degree_aware | $0.0015 \pm 0.0008$ | $89.71 \pm 8.85$ |
| 5× | **simple** | $0.0009 \pm 0.0001$ | $84.11 \pm 5.51$ |
| 5× | semantic_knn | $0.0976 \pm 0.0031$ | $64.55 \pm 9.27$ |
| 5× | synthetic | $0.0011 \pm 0.0006$ | $285.53 \pm 59.60$ |
| 5× | random | $0.0017 \pm 0.0046$ | $294.93 \pm 50.21$ |
| 2× | baseline | $0.0008 \pm 0.0001$ | $51.52 \pm 3.29$ |
| 2× | degree_aware | $0.0015 \pm 0.0010$ | $59.70 \pm 3.63$ |
| 2× | simple | $0.0016 \pm 0.0039$ | $58.55 \pm 3.02$ |
| 2× | semantic_knn | $0.0882 \pm 0.0350$ | $65.54 \pm 12.46$ |
| 2× | synthetic | $0.0014 \pm 0.0018$ | $69.10 \pm 16.66$ |
| 2× | **random** | $0.0009 \pm 0.0001$ | $73.48 \pm 23.85$ |

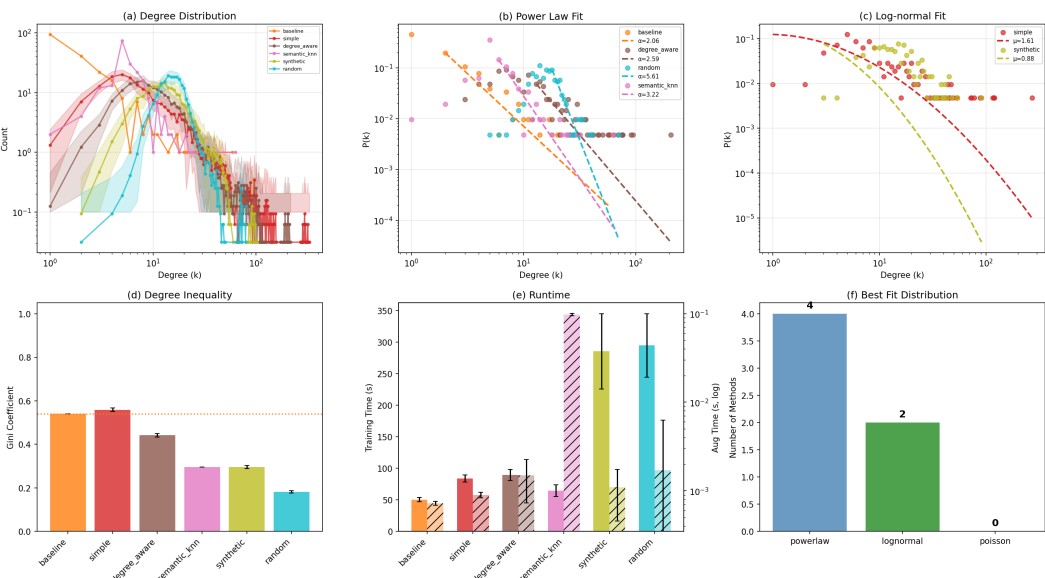

Figure 36: GDP (game–pattern), GraphSAGE, $\phi$=5: Comprehensive analysis ($M \pm$ SD, $n = 32$ seeds) comparing baseline, augmentation methods, and original graph. Panel (a) shows degree distributions on log-log scale with confidence bands; (b) Power Law fits with exponent $\alpha$; (c) Log-normal fits with parameters $\mu$ and $\sigma$; (d) Gini coefficients quantifying degree inequality (lower = more uniform); (e) runtime comparison showing training time (left axis) and augmentation time (right axis, log scale); (f) best-fit distribution counts across methods.

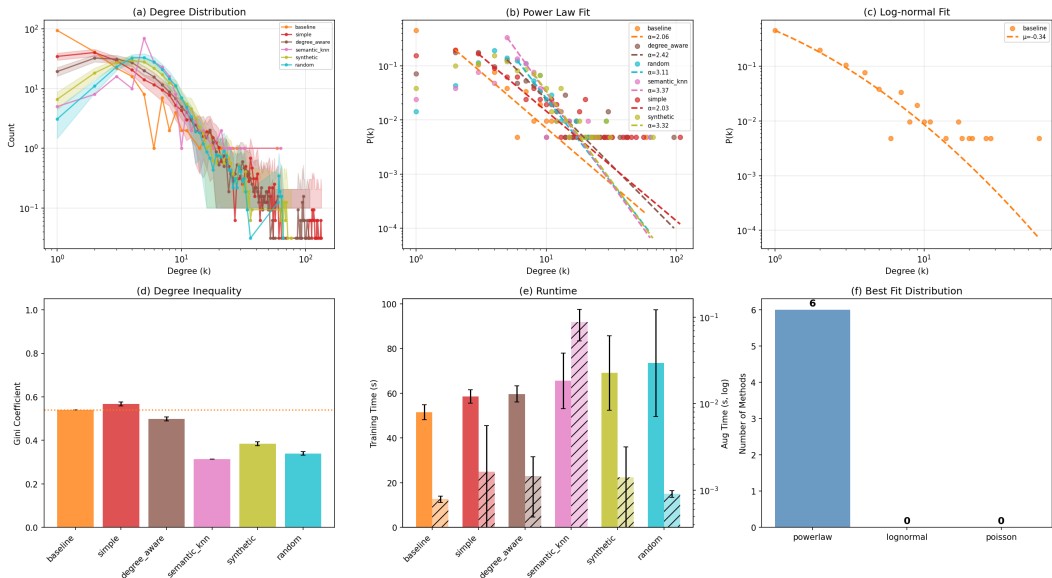

Figure 37: GDP (game–pattern), GraphSAGE, $\phi$=2: Comprehensive analysis ($M \pm$ SD, $n = 32$ seeds) comparing baseline, augmentation methods, and original graph. Panel (a) shows degree distributions on log-log scale with confidence bands; (b) Power Law fits with exponent $\alpha$; (c) Log-normal fits with parameters $\mu$ and $\sigma$; (d) Gini coefficients quantifying degree inequality (lower = more uniform); (e) runtime comparison showing training time (left axis) and augmentation time (right axis, log scale); (f) best-fit distribution counts across methods.

### C.5.4 GCN

**Summary Analysis** GDP GCN yields limited gains: `simple` and `semantic_knn` provide modest AUC at $\phi$=5/2 (up to +0.020), while `degree_aware` lags. Brier improves slightly for `degree_aware`/`simple` at $\phi$=5 ($-0.031/-0.024$) and for `semantic_knn` at $\phi$=2 ($-0.009$); `synthetic`/`random` generally worsen calibration. Degree distributions densify with larger $\phi$ (raising mean degree and reducing isolates), with `random`/`synthetic` lowering Gini most, while `semantic_knn` keeps graphs relatively sparse. Augmentation costs stay tiny ($< 0.063$s); training is higher than baseline but differences reflect graph quality, not augmentation overhead.

**AUC and Brier Score**

Table 42: GDP (game–pattern) GCN: AUC-ROC ($M \pm SD$) with paired $t$-tests vs. sparse baseline ($n = 32$ seeds). A higher AUC is better.

| $\phi$ | Method | AUC $M \pm SD$ | $\Delta$AUC | $t(31)$ | $p$ | $d$ |
|---|---|---|---|---|---|---|
| 5× | **baseline** | $0.767 \pm 0.028$ | +0.000 | — | — | — |
| 5× | degree_aware | $0.605 \pm 0.049$ | -0.162*** | 16.14 | <0.001 | +2.85 |
| 5× | simple | $0.620 \pm 0.043$ | -0.147*** | 15.89 | <0.001 | +2.81 |
| 5× | semantic_knn | $0.733 \pm 0.036$ | -0.034*** | 3.91 | <0.001 | +0.69 |
| 5× | synthetic | $0.634 \pm 0.062$ | -0.132*** | 10.84 | <0.001 | +1.92 |
| 5× | random | $0.605 \pm 0.079$ | -0.161*** | 11.05 | <0.001 | +1.95 |
| 2× | **baseline** | $0.767 \pm 0.028$ | +0.000 | — | — | — |
| 2× | degree_aware | $0.700 \pm 0.027$ | -0.067*** | 9.52 | <0.001 | +1.68 |
| 2× | simple | $0.705 \pm 0.031$ | -0.062*** | 7.93 | <0.001 | +1.40 |
| 2× | semantic_knn | $0.746 \pm 0.029$ | -0.020** | 2.83 | 0.008 | +0.50 |
| 2× | synthetic | $0.731 \pm 0.035$ | -0.035*** | 5.01 | <0.001 | +0.89 |
| 2× | random | $0.732 \pm 0.030$ | -0.034*** | 4.90 | <0.001 | +0.87 |

Table 43: GDP (game–pattern) GCN: Brier Score ($M \pm SD$) with paired $t$-tests vs. sparse baseline ($n = 32$ seeds, lower is better).

| $\phi$ | Method | Brier $M \pm SD$ | $\Delta$Brier | $t(31)$ | $p$ | $d$ |
|---|---|---|---|---|---|---|
| 5× | baseline | $0.209 \pm 0.012$ | +0.000 | — | — | — |
| 5× | degree_aware | $0.249 \pm 0.001$ | +0.041*** | −19.14 | <0.001 | −3.38 |
| 5× | simple | $0.249 \pm 0.002$ | +0.040*** | −18.89 | <0.001 | −3.34 |
| 5× | **semantic_knn** | $0.203 \pm 0.016$ | -0.005[ns] | 1.47 | 0.152 | +0.26 |
| 5× | synthetic | $0.233 \pm 0.022$ | +0.025*** | −5.75 | <0.001 | −1.02 |
| 5× | random | $0.240 \pm 0.020$ | +0.031*** | −7.44 | <0.001 | −1.32 |
| 2× | baseline | $0.209 \pm 0.012$ | +0.000 | — | — | — |
| 2× | degree_aware | $0.229 \pm 0.011$ | +0.020*** | −7.84 | <0.001 | −1.39 |
| 2× | simple | $0.225 \pm 0.009$ | +0.016*** | −6.40 | <0.001 | −1.13 |
| 2× | **semantic_knn** | $0.200 \pm 0.015$ | -0.009* | 2.64 | 0.013 | +0.47 |
| 2× | synthetic | $0.202 \pm 0.014$ | -0.007* | 2.15 | 0.040 | +0.38 |
| 2× | random | $0.203 \pm 0.016$ | -0.006[ns] | 1.64 | 0.112 | +0.29 |

## Degree Distribution Analysis

Table 44: GDP (game–pattern) GCN: Degree Distribution Statistics ($M \pm SD$, $n = 32$ seeds). Lower Gini coefficient indicates more uniform degree distribution.

| $q$ | $\phi$ | Method | Mean Degree | Gini Coeff. | Num. Isolated | Best Fit |
|---|---|---|---|---|---|---|
| 5× | baseline | | $3.4375 \pm 0.0000$ | $0.540 \pm 0.000$ | $0.0 \pm 0.0$ | powerlaw |
| 5× | degree_aware | | $17.1875 \pm 0.0000$ | $0.442 \pm 0.008$ | $0.0 \pm 0.0$ | powerlaw |
| 5× | simple | | $17.1875 \pm 0.0000$ | $0.559 \pm 0.008$ | $0.0 \pm 0.0$ | powerlaw |
| 5× | semantic_knn | | $7.1106 \pm 0.0000$ | $0.296 \pm 0.000$ | $0.0 \pm 0.0$ | powerlaw |
| 5× | synthetic | | $17.1875 \pm 0.0000$ | $0.296 \pm 0.007$ | $0.0 \pm 0.0$ | lognormal |
| 5× | **random** | | $17.1875 \pm 0.0000$ | $0.181 \pm 0.006$ | $0.0 \pm 0.0$ | powerlaw |
| 2× | baseline | | $3.4375 \pm 0.0000$ | $0.540 \pm 0.000$ | $0.0 \pm 0.0$ | powerlaw |
| 2× | degree_aware | | $6.8750 \pm 0.0000$ | $0.498 \pm 0.009$ | $0.0 \pm 0.0$ | powerlaw |
| 2× | simple | | $6.8750 \pm 0.0000$ | $0.567 \pm 0.009$ | $0.0 \pm 0.0$ | powerlaw |
| 2× | **semantic_knn** | | $6.8750 \pm 0.0000$ | $0.315 \pm 0.000$ | $0.0 \pm 0.0$ | powerlaw |
| 2× | synthetic | | $6.8750 \pm 0.0000$ | $0.384 \pm 0.009$ | $0.0 \pm 0.0$ | powerlaw |
| 2× | random | | $6.8750 \pm 0.0000$ | $0.340 \pm 0.008$ | $0.0 \pm 0.0$ | powerlaw |

## Runtime Analysis

Table 45: GDP (game–pattern) GCN: Runtime Statistics ($M \pm SD$, seconds, $n = 32$ seeds). Lower times are better.

| $\phi$ | Method | Aug. Time (s) | Train Time (s) |
|---|---|---|---|
| 5× | baseline | $0.0009 \pm 0.0008$ | $47.42 \pm 3.34$ |
| 5× | degree_aware | $0.0016 \pm 0.0012$ | $175.64 \pm 74.79$ |
| 5× | **simple** | $0.0009 \pm 0.0001$ | $169.94 \pm 85.77$ |
| 5× | semantic_knn | $0.0980 \pm 0.0033$ | $90.26 \pm 18.84$ |
| 5× | synthetic | $0.0010 \pm 0.0003$ | $260.38 \pm 118.23$ |
| 5× | random | $0.0009 \pm 0.0001$ | $243.10 \pm 114.05$ |
| 2× | baseline | $0.0008 \pm 0.0000$ | $47.79 \pm 3.36$ |
| 2× | degree_aware | $0.0012 \pm 0.0000$ | $120.59 \pm 27.40$ |
| 2× | **simple** | $0.0009 \pm 0.0000$ | $114.75 \pm 18.65$ |
| 2× | semantic_knn | $0.0812 \pm 0.0010$ | $87.45 \pm 23.11$ |

| $\phi$ | Method | Aug. Time (s) | Train Time (s) |
|---|---|---|---|
| $2\times$ | synthetic | $0.0010 \pm 0.0000$ | $88.50 \pm 16.09$ |
| $2\times$ | random | $0.0009 \pm 0.0000$ | $88.82 \pm 15.28$ |

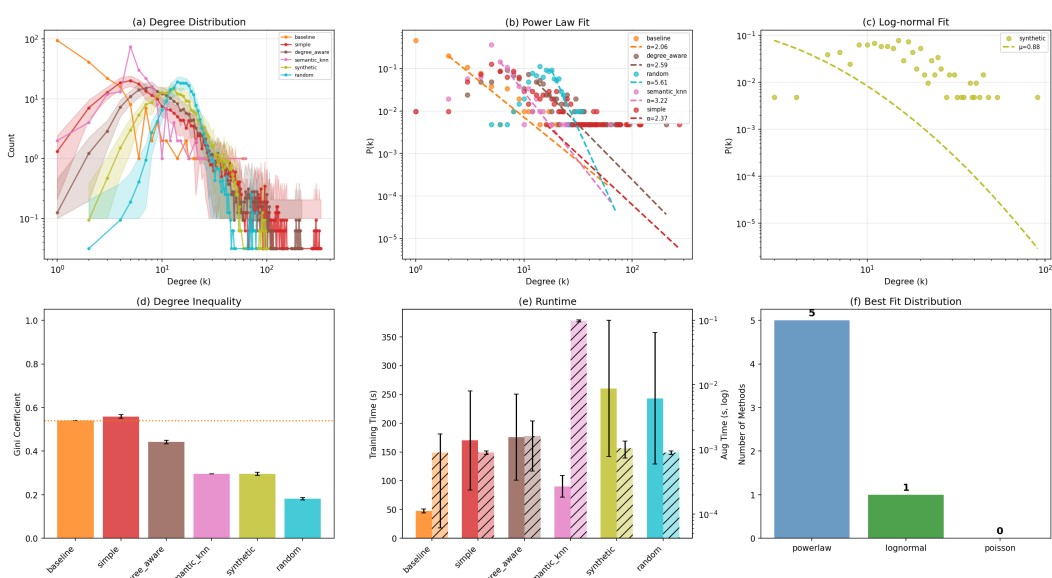

Figure 38: GDP (game–pattern), GCN, $\phi$=5: Comprehensive analysis ($M \pm$ SD, $n = 32$ seeds) comparing baseline, and augmentation methods. Panel (a) shows degree distributions on log-log scale with confidence bands; (b) Power Law fits with exponent $\alpha$; (c) Log-normal fits with parameters $\mu$ and $\sigma$; (d) Gini coefficients quantifying degree inequality (lower = more uniform); (e) runtime comparison showing training time (left axis) and augmentation time (right axis, log scale); (f) best-fit distribution counts across methods.

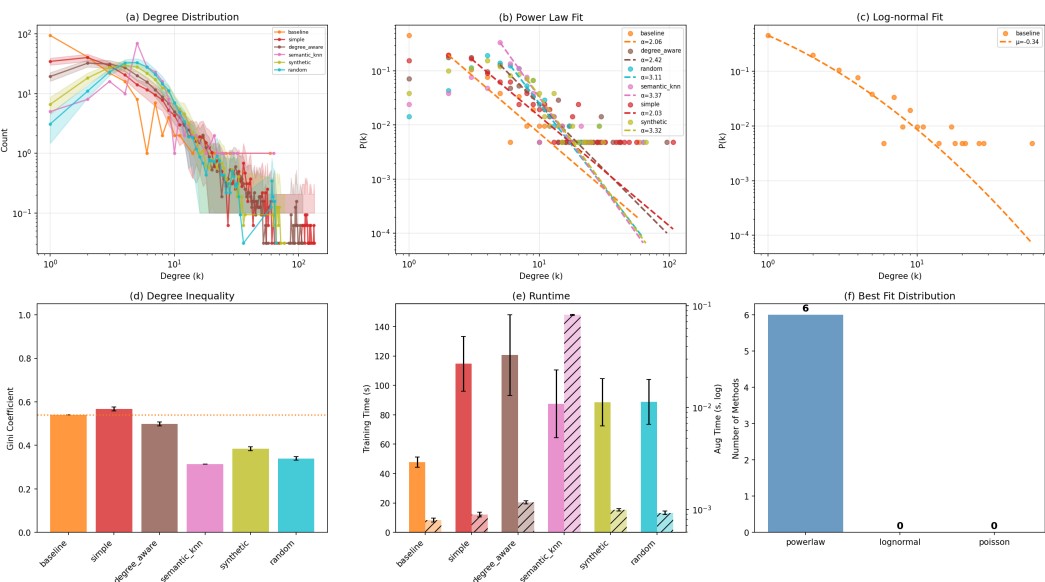

Figure 39: GDP (game–pattern), GCN, $\phi$=2: Comprehensive analysis ($M \pm$ SD, $n = 32$ seeds) comparing baseline and augmentation methods. Panel (a) shows degree distributions on log-log scale with confidence bands; (b) Power Law fits with exponent $\alpha$; (c) Log-normal fits with parameters $\mu$ and $\sigma$; (d) Gini coefficients quantifying degree inequality (lower = more uniform); (e) runtime comparison showing training time (left axis) and augmentation time (right axis, log scale); (f) best-fit distribution counts across methods.

