# OpenReview forum: "AEGIS: Authentic Edge Growth In Sparsity for Link Prediction in Edge-Sparse Bipartite Knowledge Graphs"
_ICLR.cc/2026/Conference — Submitted to ICLR 2026_

### Official Review · Reviewer_f1XV · 2025-10-18

**Soundness:** 2
**Presentation:** 2
**Contribution:** 2
**Rating:** 2
**Confidence:** 5

**Summary:**

The paper tackles the challenge of link prediction in edge-sparse bipartite knowledge graphs. Traditional graph augmentation methods—especially those involving random or synthetic edge generation—tend to distort structure and yield unreliable gains. The authors propose AEGIS (Authentic Edge Growth in Sparsity), a framework for authenticity-constrained edge-only augmentation. Rather than synthesizing new nodes or endpoints, AEGIS resamples existing edges within the training graph, preserving the node set and two-mode bipartite structure. Two policies are introduced: (i) AEGIS-Simple (uniform resampling of existing edges) and (ii) AEGIS-Degree (inverse-degree biased resampling to favor low-degree “cold-start” nodes). To benchmark performance under sparsity, the authors simulate extreme edge removal (bond percolation at 0.99 drop rate) on MovieLens and Amazon datasets, and evaluate on a naturally sparse Game Design Pattern (GDP) graph. Complementary metrics—AUC-ROC (ranking ability) and Brier score (probabilistic calibration)—are reported using paired t-tests. Five augmentation strategies are compared: AEGIS-Simple, AEGIS-Degree, random Erdos–Rényi additions, perturbation-based synthetic edges (SMOTE-style), and semantic-KNN completion (adding links between semantically similar nodes using cosine similarity). Results show that copy-based AEGIS variants maintain parity with sparse baselines, while semantic-KNN augmentation substantially improves both AUC and Brier. Random and synthetic edges generally degrade performance. The analysis highlights that semantic richness of node attributes governs augmentation success: GDP’s detailed textual descriptions yield the strongest calibration gains, whereas MovieLens’s minimal metadata offers little benefit.

**Strengths:**

1. The authors propose a well-defined and technically clear framework for the Rule-based augmentation setting over Sparse Single-relation Bipartite KGs. This makes the analysis straightforward to assess.

2. The use of AUC and Brier are good evaluation criteria; this I commend the authors focusing on these instead of the usual metrics which provide an incomplete picture.

**Weaknesses:**

1. The experimental analysis is severely lacking. The authors experimented with the single 0.99 percolation pass, however, it is important to demonstrate what happens at different values such as – 0.95, 0.9, 0.8 and such. Furthemore, the choice of 1% seems arbitrary, without any citation or specific design choice.

2. Similar to the above point, the choice of 100x upscaling/augmetation is also interesting, but there is no supporting experiment for the variation of this augmentation factor.

3. A major concern is the lack of experimental details around the Hetero GAT (I am assuming the authors mean Graph Attention Network’s Heterogeneous variant here [2])

4. Some concerns about the benchmarks used. Especially, the size of the benchmarks in terms of the numbers of nodes in each of the partitions. Practically speaking, one would also want to see the behavior on larger sized bipartite graphs, akin to the link prediction literature for KGs [1]. Can the authors shed some light on the selection of the benchmarks, the GDP benchmark is especially small.

5. The AEGIS-simple and AEGIS-degree methods are quite common in the literature ([3] as an example). Why did the authors then need to rename those? More specifically, of the first two claims on page 1 in the introduction, only claim 1 seems to be reasonable, albeit with the caveats mentioned above.

[1] Sardina, J., Kelleher, J. D., & O’Sullivan, D. A Survey on Knowledge Graph Structure and Knowledge Graph Embeddings. arXiv preprint arXiv:2412.10092 (2024).

[2] Heterogeneous Graph Attention Network Xiao Wang, Houye Ji, Chuan Shi, Bai Wang, Peng Cui, P. Yu, Yanfang Ye

[3] Fitting the Linear Preferential Attachment Model Phyllis Wan1 , Tiandong Wang2 , Richard A. Davis1 , and Sidney I. Resnick2

**Questions:**

1. The major questions surround the experimental choice for the bond percolation value, the augmentation scale.

2. Since the Hetero GAT experimental details are lacking in the paper, there are further questions around that. Especially, depending upon the number of GAT layers used, the effective aggregation radius around the nodes changes. That had a huge impact on the downstream performance, involving all the 5 upstream augmentation methods considered here.

3. Can the authors provide some runtime analysis and degree distributions of the nodes in the benchmarks considered?

---

> ### Author Response · Authors · 2025-11-19
>
> Dear Reviewer f1XV,
>
> Thank you for reviewing our submission and providing valuable feedback. Your comments on the Hetero-GAT (Graph Attention Network’s heterogeneous variant) are particularly insightful and will help us strengthen the paper.
>
> We appreciate your commendation of our use of AUC and Brier scores. We invested significant effort in selecting evaluation metrics that provide a comprehensive view of both ranking and calibration, and it is encouraging to see this recognized. We are also pleased that you found our proposed framework well-defined and technically clear, making the analysis straightforward to assess.
>
> We have provided a global response summarizing the motivation and context of this study. Please refer to that section for additional background.
>
> Below, we present detailed and structured responses to your specific comments, focusing on the weaknesses and questions you highlighted.
>
> > Weakness 1: The experimental analysis is severely lacking.
>
> > Weakness 2: ... the choice of 100x upscaling/augmetation... no supporting experiment for the variation of this augmentation factor
>
> > Question 1: The major questions surround the experimental choice for the bond percolation value, the augmentation scale
>
> We fully acknowledge these concerns regarding the generalizability of our study. A more comprehensive exploration of key factors—such as training algorithms, bond percolation values, and augmentation scales—is essential to strengthen the rigor and applicability of our findings.
>
> In the revised version, we will address this by expanding our experimental design to include multiple sparsity levels, varied augmentation budgets, and alternative model architectures. This will allow us to systematically evaluate the sensitivity of our approach and provide a more robust understanding of its performance across different conditions.
>
> > Weakness 3: A major concern is the lack of experimental details around the Hetero GAT (I am assuming the authors mean Graph Attention Network’s Heterogeneous variant here [2])
>
> > Question 2: Since the Hetero GAT experimental details are lacking in the paper, there are further questions around that...
>
> Thank you for these comments. Yes, Hetero-GAT refers to the heterogeneous variant of the Graph Attention Network. Currently, the configuration details are available in our supporting code repository, but we acknowledge that omitting them from the manuscript was an oversight. In the revised version, we will explicitly report these details in both the main text and the appendix to ensure transparency and reproducibility.
>
> Below are key configuration parameters for reference:
>
> Data split & loader settings:
>
> num_val = 0.1, num_test = 0.1
>
> disjoint_train_ratio = 0.3154
>
> neg_sampling_ratio = 1.4875
>
> add_negative_train_samples = True
>
> num_neighbors = [20, 10]
>
> batch_size = 128, shuffle = True
>
> Validation: val_num_neighbors = num_neighbors, val_batch_size = 64, val_shuffle = False
>
> Model and training schedule:
>
> gnn_type = "GAT"
>
> num_layers = 3, hidden_channels = 768
>
> learning_rate = 4.5e-4, num_epochs = 779
>
> eval_frequency = 15
>
> Early stopping: patience = 10, min_delta = 1e-3, warmup_validations = 3
>
> These details will be fully documented in the revised manuscript to address reproducibility concerns.
>
>
> > Weakness 4: Some concerns about the benchmarks used...Can the authors shed some light on the selection of the benchmarks, the GDP benchmark is especially small.
>
> Thank you for raising this point. We would like to clarify that GDP is not intended as a benchmark but rather as a domain-specific case study. The benchmark datasets used in this study are Amazon and MovieLens.
>
> This work represents an initial investigation into sparse bipartite knowledge graphs, motivated by real-world datasets like GDP in the game design domain. Our goal was to explore link prediction as a means to support knowledge discovery in this context. However, as you noted, GDP is naturally sparse and relatively small, which reflects the practical challenges of domain-specific knowledge graphs.
>
> We selected Amazon and MovieLens because they are widely recognized benchmarks for bipartite link prediction and share similar characteristics with GDP in terms of graph structure and scale. This alignment allows us to study augmentation strategies under controlled conditions while maintaining relevance to our domain use case.
>
> In the revised version, we will expand our experiments to include larger and more diverse datasets such as DBLP, ogbl-wikikg2, and ogbl-biokg to strengthen the generalizability and robustness of our findings.
>
> (To be continued in next comment)

---

> ### Author Response · Authors · 2025-11-19
>
> (Continued from previous comment)
>
> > Weakness 5: The AEGIS-simple and AEGIS-degree methods are quite common in the literature ([3] as an example). Why did the authors then need to rename those?...
>
> Thank you for pointing us to the literature [3]. After reviewing it, we would like to clarify the distinction between our AEGIS methods and the approaches discussed in [3]:
>
> - Different objectives: While [3] and related work focus on preferential attachment models for network generation, AEGIS addresses a different problem—data augmentation for graph neural network training.
>   - [3]: Simulates network growth under preferential attachment dynamics.
>   - AEGIS: Augments existing graph data to improve GNN generalization under extreme sparsity.
> - Different sampling logic: Preferential attachment favors high-degree nodes, whereas AEGIS-degree applies an inverse-degree bias to prioritize low-degree “cold-start” nodes, aiming to mitigate sparsity-driven performance degradation.
>
> We acknowledge that AEGIS-simple and AEGIS-degree are conceptually straightforward. In future work, we plan to explore more sophisticated augmentation strategies that build on these foundations.
>
> Regarding naming, we chose AEGIS to emphasize our focus on authentic edge growth in sparsity, reflecting the core principle of preserving semantic integrity in expert-verified knowledge graphs. This naming also aligns with our broader goal of developing a family of authenticity-constrained augmentation methods, of which these two variants are the starting point. If the reviewers prefer alternative naming conventions, we are open to discussion.
>
>
> > Question 3: Can the authors provide some runtime analysis and degree distributions of the nodes in the benchmarks considered?
>
> Thank you for raising this point. We acknowledge the absence of runtime analysis and degree distribution statistics in the current version and will include them in the revision to enhance completeness and transparency.
>
> Preliminary observations::
> - AEGIS-Simple/Degree: Complexity is O(|E|), linear in edge count, making these methods very fast.
> - Semantic-KNN: Complexity is O(|V|²) due to cosine similarity computations, which becomes a bottleneck for larger graphs.
> - Random/Synthetic: Comparable to AEGIS with O(|E|) complexity.
>
> For the sparse Amazon graph (67 edges), all augmentation methods complete in under one second on CPU. However, we will provide detailed runtime benchmarks for larger graphs in the revised version to highlight scalability considerations.
>
> Degree distribution:
> Preliminary analysis of the Amazon sparse graph shows:
> - Highly skewed distribution with many isolated or low-degree nodes.
> - A pronounced long-tail pattern with a few high-degree “hub” nodes.
> This motivated our inverse-degree bias in AEGIS-Degree to prioritize cold-start nodes.
>
> One clarification: Given the extreme sparsity, most nodes have degree zero. Should we report degree distribution for:
>   (a) Only connected nodes, or
>   (b) All nodes (which would show mostly zeros)?
> We welcome your input on this point.
>
> Thank you for these comments. We believe adding runtime and degree distribution analyses will significantly strengthen the paper.

---

> > ### Comment · Reviewer_f1XV · 2025-11-20
> > **Response to Authors**
> >
> > Hi, I appreciate the authors ' comments, especially on the background details of the HGAT.
> >
> > I understand the authors' intention to strengthen the manuscript by adding all the details in the updated draft. However, given the timeline and the duration of rebuttal, I will strongly encourage the authors to provide atleast experimental results and updates in the manuscript.
> >
> > Specifically, the sensitivity analysis of various parameters discussed above - multiple sparsity levels, varied augmentation budgets, and alternative model architectures - alongside some experimental data of the degree distributions and such will be very helpful for me to assess the work properly and if satisfactory, I will increase the score.
> >
> > Thanks for the effort in this initial phase of the responses.

---

> > > ### Author Response · Authors · 2025-11-27
> > >
> > > Dear Reviewer f1XV,
> > >
> > > Thank you very much for the constructive feedback and for indicating clearly which experimental additions would help you evaluate the work more thoroughly. We appreciate your encouragement regarding the initial clarifications, and we agree that further empirical evidence is essential.
> > >
> > > We have uploaded a revised version just now. Given the rebuttal window, we prioritized the items you specifically requested. In the revised manuscript, we have added:
> > >
> > > - A concise sensitivity-analysis summary in the main text, to make the key trends directly visible without navigating the Appendix.
> > > - A comprehensive set of experiments in the Appendix, covering
> > >    - multiple sparsity levels,
> > >    - varied augmentation budgets,
> > >    - alternative model architectures, and
> > >    - detailed degree-distribution and runtime diagnostics.
> > > - Expanded implementation specifics, including the HGAT-related details you previously asked about.
> > >
> > > We hope these additions make it easier for you to assess the contribution and its robustness. If there is any particular configuration or comparison you would like us to highlight more explicitly, please let us know—we would be happy to clarify within the rebuttal constraints.
> > >
> > > Thank you again for your guidance at this stage.

---

### Official Review · Reviewer_xXuc · 2025-11-01

**Soundness:** 1
**Presentation:** 3
**Contribution:** 1
**Rating:** 2
**Confidence:** 5

**Summary:**

This paper explores the impact of edge augmentation methods in the context of link prediction over bipartite graphs. To this end, it considers five different techniques aiming to augmented sparse edge information, each with its own variation (e.g., leveraging semantic node information, fully random, degree-biased, ...). The main empirical setting for this problem is a set of dense graphs (Amazon, MovieLens, GDP) that are sparsified, with the resulting link prediction performance following each augmentation strategy being compared against both the sparse baseline and the original graph.

Given its experimental results, the paper offers a discussion on the significance and effectiveness of each of its 5 proposed techniques, highlighting that only semantically-driven augmentation was a meaningful improvement over the sparse baseline in this context.

**Strengths:**

- The setting is well-designed: Starting with a dense graph and working this into a sparse setting offers natural upper and lower bounds for performance, and sets the stage well for an analysis of augmentation techniques.
- The intuitions behind the paper's methods are easy to follow.

**Weaknesses:**

- The paper makes no substantial contribution, as its main insights are highly predictable just from the conceptual empirical setting. Indeed, given a 99% sparsification and the introduction of such a large number of ultimately random edges, it is hard to imagine any other outcome than those reported in this paper. The authors mention exploring other sparsification settings as a future work. However, I think this is a necessary step towards obtaining any meaningful insights that may add more color to the different augmentation algorithm dynamics. At the moment, the extreme sparsification and randomization levels render any comparison hard to interpret.

I suggest to the authors that they explore a more continuous augmentation gradient, showcasing how performance evolves as augmentation levels increase (fixing sparsification level), and then doing the opposite (changing sparsification level, i.e., how much of the original graph is preserved, given a fixed level of augmentation), to actually study the robustness and sensitivity of the 5 techniques. I also highly recommend emphasizing performance in more realistic settings (augmentation will usually be within the same order of magnitude as the number of existing edges to maintain a reasonable amount of signal to noise).

- The paper lacks meaningful baselines to compare against its own approaches, and doesn't seem to bring any of the works from the related work section into its experimental analysis. More generally, the paper offers limited experimental insights and needs a substantial amount of additional work (including the above suggestions) with other baselines to be ready for publication.

Overall, this work is insufficiently developed in my opinion, and is based in a sparsification setting that is not optimal for the link prediction and augmentation analyses being targeted. Moreover, it doesn't offer any conceptually novel approaches or techniques. As a result, I don't think this work should be published in its current form.

**Questions:**

N/A

---

> ### Author Response · Authors · 2025-11-19
>
> Dear Reviewer xXuc,
>
> Thank you for reviewing our submission and providing valuable feedback.
>
> We appreciate your recognition that the experimental setting is well-designed. Starting with a dense graph and then applying edge removal to create sparsity was an intentional choice, as it provides natural upper and lower bounds for performance and enables a controlled analysis of augmentation strategies. We are also pleased that you found the stages of analysis logical and the underlying intuition clear.
>
> We have provided a comprehensive global response outlining the motivation and context of this study. Please refer to that section for additional background.
>
> Below, we present detailed and structured responses to your specific comments, focusing on the weaknesses and questions you highlighted.
>
> > Weakness: The paper makes no substantial contribution, as its main insights are highly predictable just from the conceptual empirical setting...sensitivity of the 5 techniques.
>
> We acknowledge your critique and agree that broader experimentation is essential to strengthen the contribution. As noted by other reviewers, we plan to expand our study to include additional datasets, alternative training algorithms, varied percolation rates, and different augmentation scales to ensure comprehensive coverage.
>
> Your suggestion of introducing a **continuous augmentation gradient** — examining performance as augmentation levels increase (while fixing sparsification) and then reversing the process — offers a highly actionable and valuable direction. We will incorporate this approach in the revised version to assess the robustness and sensitivity of all five augmentation techniques.
>
> > Weakness: I also highly recommend emphasizing performance in more realistic settings (augmentation will usually be within the same order of magnitude as the number of existing edges to maintain a reasonable amount of signal to noise).
>
> Thank you for this insightful comment. We will incorporate comprehensive experiments using realistic augmentation budgets (e.g., 1×, 2×, 5×, 10×) rather than the current 100× setting.
>
> Our choice of 100× augmentation in the initial study was driven by the need to match the original graph’s edge count after applying an extreme sparsification rate (q = 0.01), ensuring a fair comparison with the upper bound. However, we agree that this setting is unrealistic for practical applications. In the revised version, we will include experiments with more practical augmentation scales to better reflect real-world scenarios and maintain a reasonable signal-to-noise ratio.
>
> > Weakness: The paper lacks meaningful baselines to compare against its own approaches, and doesn't seem to bring any of the works from the related work section into its experimental analysis. More generally, the paper offers limited experimental insights and needs a substantial amount of additional work (including the above suggestions) with other baselines to be ready for publication.
>
> Thank you for raising this important point. We would like to clarify our current baseline choices and acknowledge the need for further expansion.
>
> In the current version, the baseline for benchmark datasets (Amazon and MovieLens) is the sparse graph without any augmentation applied. This serves as a reference point for evaluating our proposed AEGIS approaches alongside other augmentation strategies (Random ER-Like, Perturbation-based Synthetic, and Semantic-KNN) as well as the original graph (upper bound).
>
> Regarding related work, we discussed how prior studies in graph data augmentation and homophily informed our design choices. However, we did not directly implement methods such as GraphSMOTE, FG-SMOTE, CFLP, or CLBR because these approaches primarily target node classification rather than link prediction in bipartite graphs. Our five augmentation strategies were adapted from these concepts to suit the link prediction task. That said, we recognize the importance of including additional baselines and will incorporate relevant methods from the literature where feasible. If you have specific references addressing link prediction in sparse bipartite knowledge graphs, we would greatly appreciate your suggestions.
>
> We fully agree that the study requires more extensive experimentation and comparative analysis to make a substantial academic contribution. We will address this in the revision by adding more baselines, expanding datasets, and extending the experimental settings.

---

### Official Review · Reviewer_uN2E · 2025-11-01

**Soundness:** 3
**Presentation:** 3
**Contribution:** 3
**Rating:** 4
**Confidence:** 3

**Summary:**

The paper introduces AEGIS, an edge-only augmentation framework targeting extreme edge sparsity in single-relation bipartite graphs. AEGIS duplicates existing training edges (either uniformly — simple — or inverse-degree biased — degree-aware) to densify training supervision without creating new nodes. The authors compare AEGIS to three other augmentation families : **random ER-like, perturbation-synthetic (SMOTE-style), and semantic-KNN**.

Evaluation uses ROC-AUC and Brier score; significance is assessed with paired two-tailed t-tests and Cohen’s d. Main empirical claim: copy-style AEGIS matches sparse baselines, while semantic-KNN recovers AUC and calibration when node descriptions are informative

**Strengths:**

1. **Clear, reproducible algorithmic description**: The AEGIS variants and competing augmentation algorithms are specified precisely (pseudo-code, sampling weights, semantic-KNN algorithm).

2. **Insightful empirical observation**: Resampling (authentic) augmentation is a stable baseline; semantic similarity–based augmentation is required to meaningfully recover performance when textual metadata are informative.

3. **Paper well organized**

**Weaknesses:**

1. **GAT model tested**: performance could differ with other GNNs or matrix-factorization baselines.

2. **AEGIS shows weak gains**:
- Often statistically indistinguishable from baseline on MovieLens & Amazon
- Semantic KNN provides most improvements, overshadowing AEGIS resampling.

3. **Ambiguous effect of degree-aware resampling**: The inverse-degree (degree-aware) resampling is intended to help cold-start nodes, but results are mixed; e.g., GDP AUC decreases significantly for degree-aware (−0.028). The paper should investigate why degree-aware can hurt ranking while improving calibration.

**Questions:**

1. What would be results on big datasets like ogbl-wikikg2 or ogbl-biokg?
2. The experiments currently focus on one percolation case (q=0.01). How would they change for other values such as q = 0.05 or 0.1?
3. How will the results improve if you use other models like GraphSAGE, SEAL/GNN subgraph method, or KGE models (e.g., ComplEx or RotatE)?
4. Can you give some explanations for **Weaknesses** 3 and 4?

---

> ### Author Response · Authors · 2025-11-19
>
> Dear Reviewer uN2E,
>
> Thank you for reviewing our submission and providing valuable feedback.
>
> We appreciate your recognition that the algorithms are clearly described for reproducibility, the paper is well organized, and that the empirical observations offer meaningful insights. Your comments are encouraging and affirm the relevance of our work.
>
> We have provided a comprehensive global response outlining the motivation and context of this study. Please refer to that section for additional background.
>
> Below, we present detailed and structured responses to your specific comments, focusing on the weaknesses and questions you highlighted.
>
> > Weakness 1: GAT model tested: performance could differ with other GNNs or matrix-factorization baselines.
>
> > Question 1: What would be results on big datasets like ogbl-wikikg2 or ogbl-biokg?
>
> > Question 2: The experiments currently focus on one percolation case (q=0.01). How would they change for other values such as q = 0.05 or 0.1?
>
> > Question 3: How will the results improve if you use other models like GraphSAGE, SEAL/GNN subgraph method, or KGE models (e.g., ComplEx or RotatE)?
>
> Thank you for these detailed and actionable comments. They highlight an important overarching question: how generalizable are our findings under different scenarios?
>
> In the current version, we aimed to ensure some level of generalizability by evaluating two benchmark datasets alongside a domain-specific case study. However, we fully agree that additional dimensions—such as alternative models, larger datasets, varying percolation rates, and different augmentation scales—must be explored to rigorously validate our claims.
>
> We will address this by expanding our experimental scope in the revision. Specifically, we plan to:
>
> - Incorporate diverse GNN architectures (e.g., GraphSAGE, SEAL) and matrix-factorization baselines.
> - Evaluate performance on large-scale datasets such as ogbl-wikikg2 and ogbl-biokg.
> - Conduct sensitivity analyses across multiple sparsity levels (e.g., q = 0.05, 0.1) and augmentation factors.
> - Include knowledge graph embedding models (e.g., ComplEx, RotatE) for a broader comparison.
>
> These steps will strengthen the robustness and generalizability of our findings and provide a more comprehensive understanding of augmentation strategies under varying conditions.
>
> (To be continued in next comment)

---

> ### Author Response · Authors · 2025-11-19
>
> (Continued from previous comment)
>
> > Weakness 2: AEGIS shows weak gains
>
> > - Often statistically indistinguishable from baseline on MovieLens & Amazon
>
> > - Semantic KNN provides most improvements, overshadowing AEGIS resampling
>
> We acknowledge your observation regarding the limited gains of AEGIS and the dominance of Semantic-KNN improvements. Allow us to provide context and reframe our contribution.
>
> Background and motivation:
>
> This work stems from a domain-specific challenge—game design pattern discovery—rather than a purely methodological question. We encountered a sparse bipartite knowledge graph (GDP: 715 edges) where existing augmentation methods either did not apply (e.g., GraphSMOTE targets node classification) or violated semantic constraints (e.g., random edges distort expert-verified relationships).
> AEGIS was designed to test a key hypothesis: Can authenticity-constrained resampling improve link prediction in sparse knowledge graphs?
>
> On the observed weak gains:
>
> Our empirical results challenge this hypothesis. AEGIS-simple and AEGIS-degree show no significant improvement over sparse baselines on Amazon and MovieLens. While initially disappointing, these findings are highly informative:
>
> What does not work under extreme sparsity (q = 0.01):
>
> - Uniform edge resampling (AEGIS-simple): no measurable benefit
> - Inverse-degree bias (AEGIS-degree): mixed results (hurts AUC on GDP, slight Brier improvement)
> - Random edges: statistically significant degradation
> - Synthetic perturbation: also harmful
>
> What does work:
>
> - Semantic-KNN (leveraging textual features): consistent, significant gains across datasets
>
> Revised contribution:
>
> Rather than claiming an effective augmentation method, our contribution is better framed as:
>
> - Empirical insight: In edge-sparse bipartite knowledge graphs, semantic similarity (when textual features are available) dominates structural augmentation strategies. Authenticity alone is insufficient.
> - Baseline establishment: AEGIS provides reproducible, simple baselines for future research in sparse graph augmentation. Negative results are valuable for guiding future work and avoiding redundant exploration.
> - Domain guidance: For expert-curated knowledge graphs like GDP, investing in richer textual descriptions yields greater benefits than structural augmentation alone.
>
> On incremental contribution:
>
> - We agree AEGIS is conceptually simple and may not meet the novelty threshold for top-tier venues from a methodological perspective. However, for domain researchers facing similar challenges—sparse, expert-curated, bipartite knowledge graphs—these empirical findings offer practical guidance absent from current literature.
>
> We appreciate the reviewers’ feedback in clarifying the gap between domain-driven exploration and methodological expectations at ICLR, and we will incorporate this perspective into the revised manuscript.
>
> > Weakness 3: Ambiguous effect of degree-aware resampling
>
> Thank you for highlighting this important observation. We have two working hypotheses to explain the mixed outcomes:
>
> Hypothesis 1: Cold-start assumptions do not transfer well to knowledge graphs
>
> Degree-aware resampling is designed for recommendation systems, where low degree typically signals insufficient training data. In knowledge graphs like GDP, however, low degree often reflects genuine rarity—the relationship truly does not apply broadly. Oversampling these rare edges introduces distributional shift, effectively signaling to the model that these rare patterns are common. This leads to miscalibration (Brier score +0.036) and degraded ranking performance (AUC −0.028).
>
> In contrast, for artificially sparsified graphs such as Amazon and MovieLens, edge removal is random, so degree-aware resampling has a more neutral effect.
>
> Hypothesis 2: Semantic features dominate structural signals in GDP
>
> GDP contains rich textual descriptions (200–500 words per node), making semantic similarity the primary predictive signal (as evidenced by Semantic-KNN’s +0.014 AUC gain). Degree-aware resampling prioritizes structural considerations over semantics, potentially down-weighting semantically central, high-degree patterns.
>
> For Amazon and MovieLens, where textual features are limited, structural and semantic signals align more closely, reducing this conflict.
>
> We will investigate these hypotheses systematically in the revised work by testing degree-aware resampling across datasets with varying levels of textual richness. Thank you for pointing this out—it opens a promising direction for future research.
>
> > Question 4: Can you give some explanations for Weaknesses 3 and 4?
>
> Thank you for raising this question. We have addressed Weakness 3 (“Ambiguous effect of degree-aware resampling”) in detail above. However, we could not locate Weakness 4 in the posted review. Could you kindly clarify or point us to the specific comment? This will help us provide a precise and thorough response.

---

### Official Review · Reviewer_Gfnf · 2025-11-01

**Soundness:** 2
**Presentation:** 2
**Contribution:** 2
**Rating:** 2
**Confidence:** 3

**Summary:**

This paper proposes AEGIS, a simple  framework for link prediction in edge-sparse bipartite graphs. Instead of generating synthetic edges, AEGIS augments training data by resampling authentic existing edges, with variants like degree-aware and semantic KNN sampling. Experiments show that it has some improvements in some cases.

**Strengths:**

1. The paper tackles an ealistic problem — link prediction under extreme edge sparsity in bipartite knowledge graphs, which is interesting.

2. The proposed idea of authentic edge augmentation is intuitive and easy to implement, avoiding noisy or unrealistic synthetic edges.

**Weaknesses:**

1. The paper is not well organized, and the motivation behind the proposed authentic edge–based augmentation is not clearly articulated.

2. The method description is limited to pseudo-code without sufficient explanation of design details or implementation.

3. The proposed AEGIS variants perform very close to the baseline, raising questions about the method’s effectiveness. What's the definition of "baseline" in the experiment?

4. The contribution is relatively incremental — resampling existing edges is conceptually simple.

**Questions:**

please refer to the weakness

---

> ### Author Response · Authors · 2025-11-19
>
> Dear Reviewer Gfnf,
>
> Thank you for reviewing our submission and providing feedback.
>
> We appreciate your recognition that our research addresses an important and realistic problem, and that the proposed ideas are intuitive and straightforward to implement. Your acknowledgment reinforces the relevance of our work to relevant challenges in sparse bipartite knowledge graphs.
>
> We have provided a comprehensive global response outlining the motivation and context of this study. Please refer to that section for additional background.
>
> Below, we present detailed and structured responses to your specific comments, focusing on the weaknesses and questions you highlighted.
>
> > Weakness 1: The paper is not well organized, and the motivation behind the proposed authentic edge–based augmentation is not clearly articulated.
>
> We acknowledge that the motivation for authenticity-constrained augmentation was not sufficiently emphasized in the initial submission. Our focus on authenticity is deliberate and stems from the characteristics of our data: the original graphs are inherently sparse, and downstream tasks such as link prediction perform poorly under these conditions. To address this, we explored augmentation strategies that preserve semantic integrity. Because the relationships in our knowledge graphs are expert-verified, introducing synthetic or random edges would compromise authenticity and distort domain semantics. Therefore, we designed augmentation methods that operate exclusively on existing edges, aligning with the core task of link prediction while maintaining structural and semantic fidelity.
>
> In the original version, we prioritized methodological and technical contributions, as these are central to ICLR’s review criteria. Given the page limit, we condensed the motivation section. In the revision, we will expand this discussion to clearly articulate the rationale for authenticity-constrained augmentation and its critical role in sparse bipartite knowledge graphs.
>
> Could you kindly elaborate on the specific aspects of the paper’s organization that you found unclear? This will help us make targeted improvements.
>
> > Weakness 2: The method description is limited to pseudo-code without sufficient explanation of design details or implementation.
>
> Thank you for highlighting this point. We intentionally used pseudo-code to present the algorithms in a generalized and implementation-agnostic manner, ensuring clarity regardless of programming language. However, we recognize that additional design details and implementation specifics would strengthen the paper. As noted in the supplementary material, our implementation is in Python using PyTorch-based libraries, but these details were omitted due to space constraints.
>
> In the revised version, we will provide a more comprehensive explanation of the design rationale, implementation choices, and configuration details to enhance transparency and reproducibility.
>
> (To be continued in next comment)

---

> ### Author Response · Authors · 2025-11-19
>
> (Continued from previous comment)
>
> > Weakness 3: The proposed AEGIS variants perform very close to the baseline, raising questions about the method’s effectiveness. What's the definition of "baseline" in the experiment?
>
> > Weakness 4: The contribution is relatively incremental - resampling existing edges is conceptually simple
>
> For the benchmark datasets (Amazon and MovieLens), baseline refers to the original dataset after edge-dropping without any augmentation applied. For the domain case study (GDP), baseline denotes the naturally sparse original graph without augmentation. We considered labeling GDP as “baseline/original,” but ultimately chose “baseline” to emphasize that no augmentation was applied. In contrast, “original” in our study specifically means the graph before edge-dropping.
>
> We acknowledge both critiques and would like to provide context and reframe our contribution.
>
> Background and motivation:
> This work originates from a domain-driven challenge (game design pattern discovery) rather than a purely methodological question. We encountered a sparse bipartite knowledge graph (GDP: 715 edges) where existing augmentation methods either did not apply (e.g., GraphSMOTE targets node classification) or violated semantic constraints (random edges distort expert-verified relationships).
> AEGIS was developed to test a key hypothesis: Can authenticity-constrained resampling improve link prediction in sparse knowledge graphs?
>
> On the observed weak gains:
> Our empirical results challenge this hypothesis. AEGIS-simple and AEGIS-degree show no significant improvement over sparse baselines on Amazon and MovieLens. While initially disappointing, these findings are highly informative:
>
> What does not work under extreme sparsity (q = 0.01):
> - Uniform edge resampling (AEGIS-simple): no measurable benefit
> - Inverse-degree bias (AEGIS-degree): mixed results (hurts AUC on GDP, slight Brier improvement)
> - Random edges: statistically significant degradation
> - Synthetic perturbation: also harmful
>
> What does work:
> - Semantic-KNN (leveraging textual features): consistent, significant gains across datasets
>
> Revised contribution:
> Rather than claiming an effective augmentation method, our contribution is better framed as:
> 1. Empirical insight: In edge-sparse bipartite knowledge graphs, semantic similarity (when textual features are available) dominates structural augmentation strategies. Authenticity alone is insufficient.
> 2. Baseline establishment: AEGIS provides reproducible, simple baselines for future research in sparse graph augmentation. Negative results are valuable for avoiding redundant exploration.
> 3. Domain guidance: For expert-curated knowledge graphs like GDP, investing in richer textual descriptions yields greater benefits than structural augmentation alone.
>
> On incremental contribution:
> We agree AEGIS is conceptually simple and may not meet the novelty threshold for top-tier venues from a methodological perspective. However, for domain researchers facing similar challenges (sparse, expert-curated, bipartite knowledge graphs), these empirical findings offer subject-field specific guidance absent from current literature.
> We recognize the gap between domain-driven exploration and methodological expectations at ICLR and appreciate the reviewers’ feedback in clarifying this distinction.

---

### Author Response · Authors · 2025-11-19

We deeply appreciate the reviewers for their thoughtful and constructive feedback. This work tackles the critical challenge of link prediction in edge-sparse bipartite graphs—a problem that frequently arises in real-world applications such as knowledge representation and discovery within the game design domain. Our motivating dataset, Game Design Pattern (GDP), exemplifies this scenario with 208 games, 296 patterns, and 715 expert-verified links, which makes it inherently sparse and significantly challenging for downstream tasks like link prediction to achieve practical performance.

Unlike typical bipartite graphs (e.g., user–item networks), knowledge graphs impose strict authenticity requirements because they rely on expert-verified relationships. This constraint fundamentally limits the applicability of synthetic or random augmentation strategies commonly used in other domains. To address this, we propose authenticity-constrained edge augmentation as a rigorous and principled approach to improve link prediction performance while preserving semantic integrity. Our study introduces AEGIS as an essential first step toward robust and scalable solutions for sparse bipartite knowledge graphs, balancing structural fidelity with improved predictive reliability.

We thank you once again for your comments and the references provided. Please find below our detailed and structured responses to each reviewer’s comment.

---

### Author Response · Authors · 2025-11-27

Dear Reviewers,

Thank you once again for the thoughtful and constructive feedback. We have now uploaded a revised manuscript, with all changes highlighted in red for ease of reference. Given the rebuttal window, we prioritized the additions that you collectively emphasized: (i) a concise summary of the sensitivity analysis in the main text, (ii) expanded implementation details to clarify the design choices, and (iii) a comprehensive set of sensitivity analyses—covering multiple sparsity levels, augmentation budgets, model architectures, and accompanying degree-distribution diagnostics and runtime analysis—in the Appendix.

Other than these targeted additions, the overall structure and presentation of the manuscript remain unchanged. We hope the revisions address your concerns and make the contributions and methodology clearer.

Thank you.

---

### Meta-Review · Area_Chair_qu78 · 2026-01-01

**Summary:**

In this paper, the authors study edge-only data augmentation for link prediction in extremely edge-sparse bipartite knowledge graphs. In particular, they propose AEGIS, which resamples existing edges under different policies. While the problem setting is relevant, the reviewers raised serious concerns about the novelty, experimental design, and overall contribution. In particular, the main empirical findings show that the proposed AEGIS variants provide little or no improvement over sparse baselines. Although the authors provided rebuttal responses and promised additional experiments, the revised manuscript still does not convincingly demonstrate a strong methodological or empirical advance suitable for ICLR.

**Reviewer Concerns:**

Concerns partially addressed:

1. The reviewers noted missing details for the Hetero-GAT model and experimental setup. The authors clarified these points in the rebuttal and added implementation details and diagnostics in the revised version.

2. The reviewers requested analysis across different sparsity levels, augmentation budgets, and architectures. The authors added sensitivity analyses and additional results.

3. The rebuttal clarified the domain motivation (expert-curated bipartite KGs such as GDP) and explained why synthetic or random edges may be undesirable in this setting.

Concerns still outstanding:

1. Multiple reviewers explicitly stated that resampling existing edges (uniform or degree-based) is conceptually simple and closely related to existing ideas. Even after revision, the contribution remains largely incremental.

2. A core concern remains that AEGIS-simple and AEGIS-degree are often statistically indistinguishable from the sparse baseline, and sometimes worse.

3. The reviewers noted missing comparisons with stronger or more standard baselines (e.g., KGE methods, alternative link prediction models, larger benchmarks), and these gaps are not fully closed in the revision.

4. Several reviewers concluded that, despite careful experimentation, the paper mainly reports expected or negative results and does not provide sufficiently deep new insights to justify publication at ICLR.

**Reviewer Scores:**

Reviewer Gfnf: Gave a 2 due to limited novelty, unclear motivation, and weak improvements over baseline. These concerns remain after rebuttal. Hence, the score would likely remain 2.

Reviewer uN2E: The reviewer gave 4. They acknowledge careful experiments but noting weak gains and limited generalization. Their score would likely remain 4 after reuttal.

Reviewer xXuc: The reviewer gave 2 with high confidence. They emphasize the concerns of lack of substantial contribution, unrealistic experimental setting, and insufficient baselines. The rebuttal version does not fundamentally change this assessment. Hence, the score would likely remain 2.

Reviewer f1XV: The reviewer gave 2. They raised cocerns regarding benchmark choice, and contribution level. After the rebuttal, the reviewer explicitly requested more evidence before reconsidering. Hence, it is likely they will remain the score of 2.

---

### Decision · Program_Chairs · 2026-01-26

Reject